# Drosophila Alms1 proteins regulate centriolar cartwheel assembly by enabling Plk4-Ana2 amplification loop

Marine Brunet [ID] [1,2,3], Joëlle Thomas [ID] [1,2,3], Jean-André Lapart [ID] [1,2,3], Léo Krüttli[1,2,3], Marine H Laporte[1,2,3], Maria Giovanna Riparbelli[4], Giuliano Callaini[4], Bénédicte Durand [ID] [1,2,3,5✉] & Véronique Morel [ID] [1,2,3,5✉]

## Abstract

Centrioles play a central role in cell division by recruiting peri-centriolar material (PCM) to form the centrosome. Alterations in centriole number or function lead to various diseases including cancer or microcephaly. Centriole duplication is a highly conserved mechanism in eukaryotes. Here, we show that the two *Drosophila* orthologs of the Alström syndrome protein 1 (Alms1a and Alms1b) are unexpected novel players of centriole duplication in fly. Using Ultrastructure Expansion Microscopy, we reveal that Alms1a is a PCM protein that is loaded proximally on centrioles at the onset of procentriole formation, whereas Alms1b caps the base of mature centrioles. We demonstrate that chronic loss of Alms1 proteins (with RNA null alleles) affects PCM maturation, whereas their acute loss (in RNAi KD) completely disrupts procentriole formation before Sas-6 cartwheel assembly. We establish that Alms1 proteins are required for the amplification of the Plk4-Ana2 pool at the duplication site and the subsequent Sas-6 recruitment. Thus, Alms1 proteins are novel critical but highly buffered regulators of PCM and cartwheel assembly in flies.

**Keywords** Centriole Duplication; Ana2; Plk4; Alms1; Alström Syndrome
**Subject Categories** Cell Adhesion, Polarity & Cytoskeleton; Cell Cycle

## Introduction

Centrioles are highly conserved, small, cylindrical organelles composed of nine triplets of microtubules. They are surrounded by a complex array of proteins, the pericentriolar material (PCM). Together they form the centrosomes that serve as microtubule-organising centres (MTOC), playing a central role in cell division. Alterations of centriole number are associated with numerous pathologies, including cancers (Levine et al, 2017). Centriole duplication must therefore be tightly regulated and coordinated with the cell cycle (Nigg and Holland, 2018).

Non-dividing cells normally contain a pair of connected centrioles, an older, the mother, and a younger, the daughter, arranged perpendicularly. As the cell enters G1-phase, centrioles disconnect in a process called disengagement and initiate their duplication. A procentriole forms on the side of each centriole, grows to become a daughter centriole and recruits PCM. The two duplicated centrosomes hence formed organise the mitotic spindle and, ultimately as cells divide, each daughter cell inherits a single centrosome made of a mother and a daughter centriole.

Centriole duplication relies on a few sets of proteins that are highly conserved across species and which functions are highly hierarchised. It begins with the recruitment of the Plk4 kinase (respectively Sak in *Drosophila* and PLK4 in mammals) by the PCM proteins Asl (CEP152) and Spd2 (CEP192) at the proximal side of the centriole (Dzhindzhev et al, 2010; Kim et al, 2013; Sonnen et al, 2013). Plk4 initially forms a cylinder around the centriole and next resolves to a single dot which corresponds to the future site of procentriole assembly (Dzhindzhev et al, 2017; Yamamoto and Kitagawa, 2019). This reorganisation relies on the stabilisation of Plk4 by Ana2 (STIL) (Moyer et al, 2015; Ohta et al, 2014, 2018) and a complex cascade of trans autophosphorylations (Sillibourne and Bornens, 2010; Lopes et al, 2015; Park et al, 2019) leading to Plk4 activation. This initiates the sequential phosphorylation on multiple sites of Ana2 which can then recruit Sas-6 (SAS-6) (Dzhindzhev et al, 2017; McLamarrah et al, 2018; Moyer and Holland, 2019) to form the cylindrical ninefold symmetry array of the cartwheel. The latter serves as a template for Sas-4 (CPAP)-controlled deposition of microtubule triplets, initiating procentriole elongation (Kitagawa et al, 2011; van Breugel et al, 2011).

ALMS1 is a centrosome and cilium-associated protein characterised by a C-terminal ALMS domain involved in the localisation of the protein at centrioles and basal bodies (Hearn et al, 2005; Knorz et al, 2010). Mutations in *ALMS1* are responsible for a rare ciliopathy, the Alström syndrome (Collin et al, 2002; Hearn et al, 2002), but how ALMS1 controls ciliary functions is still enigmatic. Two *alms1* genes have been identified in *Drosophila*, *alms1a* and *alms1b*, which share more than 80% sequence homology. Alms1a was shown to localise at the centrosome in the *Drosophila* male germline and to be involved in centriole duplication in asymmetrically dividing male germline stem cells

[1]Universite Claude BERNARD Lyon 1, Lyon, France. [2]MeLiS—CNRS-UMR5284, Lyon, France. [3]INSERM-U1314, Lyon, France. [4]Università degli Studi di Siena, Siena, Italy. [5]These authors contributed equally: Bénédicte Durand, Véronique Morel. ✉E-mail: benedicte.durand@univ-lyon1.fr; veronique.morel@univ-lyon1.fr

(GSCs), at the exclusion of other cell or division types (Chen and Yamashita, 2020).

Using Ultrastructure Expansion Microscopy (U-ExM), we characterised Alms1a and Alms1b localisation with unprecedented resolution. We show that Alms1a and Alms1b are sequentially recruited at the proximal end of centrioles and that Alms1a is also a component of the PCM. We observe that sudden and severe, hereafter referred to as acute, depletion of both Alms1a and b by RNAi is associated with complete centriole duplication failure in several *Drosophila* tissues, while chronic loss of *alms1a* (using an RNA null *alms1a* mutant) leads to reduced PCM recruitment and chronic loss of both *alms1a* and *1b* (RNA null alleles) leads to premature centriole disengagement. Finally, we place Alms1a,b in the molecular hierarchy of centriole duplication. We show that they are involved in the efficient recruitment/stabilisation of Plk4 and Ana2 at the centriole duplication site and that the loss of Alms1a,b leads to the complete failure of Sas-6 recruitment. Collectively, our work demonstrates that Alms1a and b are key but highly buffered players in the initiation of cartwheel formation and PCM assembly in *Drosophila*.

# Results

## Alms1a and Alms1b present different spatiotemporal localisations at centrioles

To characterise the centriolar localisation of Alms1a and b proteins and their dynamics of recruitment during centriole duplication in *Drosophila* (Fig. 1), we adapted the U-ExM protocol (Gambarotto et al, 2019) to several *Drosophila* tissues (testes, larval brains and early embryos) expressing either Tomato-tagged Alms1a or GFP-tagged Alms1b (Appendix Fig. S1A).

*Drosophila* spermatogenesis starts with the asymmetric division of a germline stem cells (GSC) which gives rise to a GSC and a goniablast that initiates differentiation into a spermatogonium (SG) (Appendix Fig. S2A). SG undergo four symmetric divisions to generate a cyst of 16 cells, each containing two extremely small (minute) centrioles. As SGs complete a pre-meiotic S phase and duplicate their centrioles, they differentiate into spermatocytes (SCs) (Appendix Fig. S2A). In SCs, the 4 centrioles grow unusually long, reaching about 1.3 µm in length, and all centrioles mature into basal bodies nucleating a primary-like-cilium (Jana et al, 2016). SCs finally undergo two meiotic divisions, forming 4 spermatids each (resulting in 64 spermatids for each initial goniablast) with one basal body from which the sperm flagellum emanates (Demarco et al, 2014).

We observed that Alms1b-GFP is not detected at centrosomes in spermatogonia (Fig. 1A, SG) and is first recruited to the mother centriole (MC) in early spermatocytes, forming a ring which caps the proximal end of centriolar microtubules, while no Alms1b-GFP is observed on the daughter centriole (DC, Fig. 1A, early SC: SC#1). This strong asymmetry within the centrosome is maintained until the end of the SC stage (Fig. 1A, late SC: SC#2). As SC undergo the second meiotic division, both centrioles exhibit similar amounts of proximal Alms1b-GFP (Fig. EV1A), which suggests that its recruitment at daughter centrioles occurs rapidly at the very end of the SC stage or during meiotic divisions (Fig. 1A, late SC: SC#2, white arrow). In contrast, Alms1a-Tomato localises at both mother

and daughter centrioles from the GSC stage onwards and forms a ring at the base of the centriole wall and a sleeve surrounding it (Fig. 1B). It is loaded on the nascent procentriole during each centriole duplication event (Fig. 1B, SG#1 and #2). Alms1a-Tomato then accumulates on the daughter centriole and at mid-late SC stages both mother and daughter centrioles present similar amounts of Alms1a-Tomato at their proximal end (Fig. 1B, SC). Together, these results indicate that Alms1a is associated with each centriolar duplication event in spermatogenesis, from asymmetrically dividing stem cells (GSCs) (Chen and Yamashita, 2020) to symmetrically dividing spermatogonia, whereas Alms1b is only associated with post-duplication mature centrioles. Similar dynamics of localisation were observed for endogenous Alms1a and Alms1b (Fig. EV1B) using previously published antibodies (Chen and Yamashita, 2020).

Neuroblasts (NBs) are somatic stem cells that generate the central nervous system of flies through asymmetric divisions (Januschke et al, 2011). In larval brain NBs, Alms1a-Tomato localises at both mother and daughter centrioles, labelled with the PCM protein Asterless (Asl, Fig. 1C), with a higher concentration of Alms1a-Tomato on the mother compared to the daughter centriole in early interphase (Fig. 1C, left panel NB#1, MC at the top, DC at the bottom). This asymmetry is still observed in pre-mitosis, after migration to the opposite pole of the NB of the centrioles, the mother centriole being inherited by the differentiating cell (ganglion mother cell, GMC) (Januschke et al, 2011) while the daughter centriole is maintained in the NB (Fig. 1C, right panel NB#2, DC: #1, MC: #2). In contrast to Alms1a, we were unable to detect Alms1b-GFP in the larval brain neither in NBs nor GMCs (Fig. EV1C), in agreement with RNA-seq data showing low expression of *alms1b* in neurons and glial cells (Li et al, 2022; Berger et al, 2012).

In syncytial embryos, which undergo 13 symmetric and synchronous mitosis (Foe and Alberts, 1983), Alms1a-Tomato is observed at both centrioles from the first mitosis onwards. As in the male germline, we detected a faint Alms1a-Tomato staining at the onset of procentriole assembly in prophase (Fig. 1D, embryo #1, white arrow), the signal becoming stronger on daughter centrioles at the beginning of the centriole-to-centrosome conversion in metaphase (Fig. 1D, embryo #2). In comparison, Alms1b-GFP was not detected in syncytial embryos and only observed at centrioles after cellularisation (Fig. EV1D).

Thus, despite their high degree of identity (>80%), Alms1a and Alms1b display very different spatiotemporal dynamics of centriolar recruitment in all tissues observed. Alms1a is observed on all centrioles from the onset of procentriole assembly, while Alms1b is only detected on mature centrioles.

As only one *ALMS1* gene has been identified in humans, we wondered whether Alms1a or b was more similar to human ALMS1 by comparing their localisation profiles. Using U-ExM on cycling human retinal pigment epithelial cells (RPE-1), we observed that ALMS1 forms a ring capping the proximal end of both mother and daughter centrioles and slightly overlaps with the proximal segment of the centriolar wall (Fig. 2A,B). Careful examination of the dynamics of ALMS1 recruitment further reveals that ALMS1 is recruited after the onset of procentriole formation, as the procentriole reaches a length of 120 nm (Fig. 2C–E), which corresponds to the beginning of the procentriole elongation phase (Laporte et al, 2024). Thus, the localisation dynamics of human

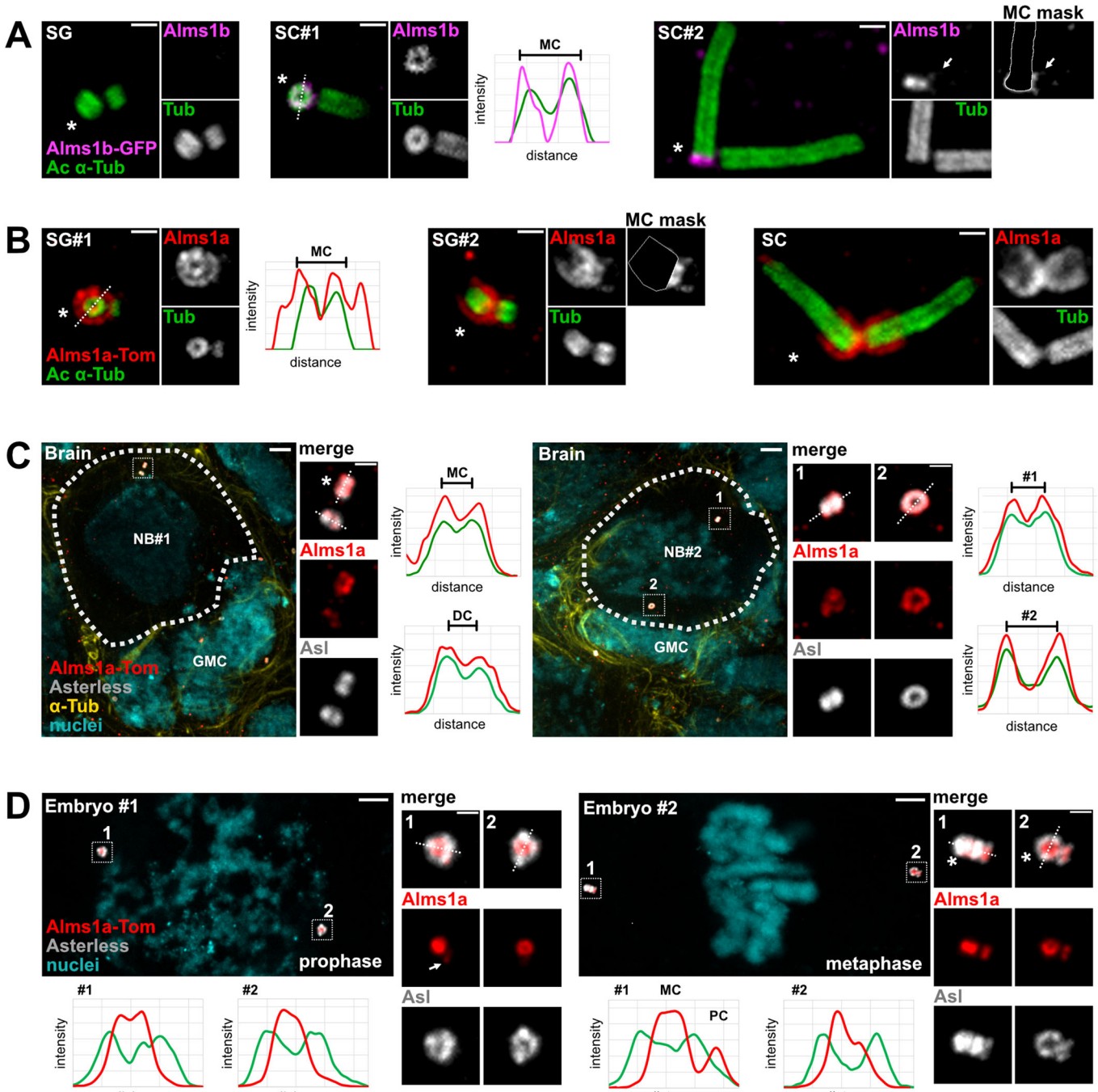

ALMS1 is more closely related to Alms1a profile, even though the fly Alms1a is recruited earlier during procentriole formation in the spermatic lineage than is human ALMS1 during procentriole elongation in RPE-1 cells.

## Alms1a is an inner PCM protein

Spatial analysis of Alms1a in U-ExM shows that Alms1a-Tomato colocalises with Asterless (Asl, CEP152) and Pericentrin-like protein (Plp, PCNT) (Fig. 3A,B), which are proposed to link centrioles and PCM and are hence called inner or bridge PCM proteins (Lattao et al, 2017; Varadarajan and Rusan, 2018). While

Alms1a-Tomato fully surrounds the minute centrioles of SG (Fig. 1B), it remains restricted to the proximal third of the growing centriole of SC, a behaviour similar to Plp localisation (Figs. 1B, SC and 3B).

To confirm a possible function of Alms1a in PCM assembly, we generated fly lines with an *alms1a* deletion by CRISPR/Cas9 (*alms1a*[del1], Appendix Fig. S1) and quantified the fluorescence intensity of Asl and Plp at centrioles in control and *alms1a*[del1] spermatogonia, early and late spermatocytes. We observed a significant decrease of both Asl and Plp fluorescence intensity in *alms1a*[del1] cells at all stages (Figs. 3C,D and EV2A). Expression of Alms1a-Tom in *alms1a*[del1] restored Asl intensity to normal, hence

**Figure 1.  Alms1a and Alms1b show different spatial-temporal localisations.**

U-ExM images of (A, B) the centriolar localisation of (A) Alms1b, as shown with the expression of the fusion protein Alms1b-GFP under *alms1b* native promotor in *alms1b* deletion background (magenta; *alms1b*^del2^, Alms1b-GFP) and (B) Alms1a, as shown with the expression of the fusion protein Alms1a-Tom under *alms1a* native promotor in *alms1a* deletion background (red; *alms1a*^del1^, Alms1a-Tom) at early stages of spermatogenesis (spermatogonia SG; spermatocytes SC). (A) SC#1: early SC, bottom view exposing the proximal ring of Alms1b on mother centrioles (MC). SC#2: late SC, Alms1b is also detected at the proximal end of daughter centrioles (DC) (white arrow). (B) SG#1, bottom view with the focus plane on the proximal end of the MC, revealing the two concentric localisations of Alms1a. SG#2, Alms1a is also detected around the DC. Centriolar walls are revealed with acetylated α-tubulin (green, ac α-tub). MC are shown on the left, DC on the right. MC-masked insets enable better visualisation of faint localisations. (C) Alms1a-Tomato (red) localisation at centrioles in larval brain neuroblasts (NBs) and ganglion mother cells (GMCs). NB#1, shows a NB in interphase. Insets: close-up of the centriolar pair, MC at the top and DC at the bottom. Alms1a (red) forms a ring superposed to the ring of Asterless (Asl, grey) and is less abundant at the DC compared to the MC. NB#2, shows a dividing NB. The two centrioles have separated and migrated on both sides of the cell. DC in inset 1, MC in 2. (D) Alms1a-Tomato (red) localisation at centrioles in early embryos. Left panel: prophase. In the inset, the two centrioles have initiated their duplication as evidenced by the faint Alms1a concentration on the side of the centriole (white arrows) preceding the recruitment of Asl. Right panel: metaphase with advanced centriole duplication as shown in the insets by the accumulation of both Alms1a and Asl on the side of the MCs. In these two stages, the cells are dividing, forming both asters and mitotic spindle. This explains the wide and intense Asl staining with respect to Alms1a. This contrasts with NBs, GSCs and SCs where cells are not in the process of forming a spindle and where Alms1a localisation is slightly wider than that of Asl. Asl (grey) labels the centriole, α-tubulin (yellow) the cytoskeleton and Hoechst (cyan) the nuclei. In all images, mother centrioles are marked with an asterisk. In all panels, the fluorescence intensities along the white dotted line are plotted as a function of distance to further document the width of Alms1a or b localisation with respect to centriolar makers acetylated α-tubulin or Asl (A–D) Brackets show the centriolar wall width of Mother (MC) or Daughter (DC) Centrioles. Scale bars (corrected from expansion factor): (A, B): 250 nm, (C, D): 1 µm, insets: 250 nm.

showing the specificity of the phenotype (Fig. EV2B). Altogether, these observations indicate that Alms1a is a novel inner PCM protein required to either efficiently recruit or stabilise the inner PCM proteins Asl and Plp at centrioles.

## Compensatory mechanisms are triggered upon chronic *alms1a, alms1b* loss-of-function

A previous study demonstrated that depletion of both *alms1a* and *alms1b* by RNAi in asymmetrically dividing GSCs results in a major loss of centrosomes and proposed, based on the absence of Alms1b in GSC, that Alms1a alone was involved (Chen and Yamashita, 2020). Here, despite the full deletion of *alms1a* locus in *alms1a*^del1^ mutant (an mRNA null allele, Appendix Fig. S1C), we failed to detect any centriole duplication loss (Fig. EV3A, middle). To exclude a possible compensation of *alms1a* loss by *alms1b*, we generated flies carrying a genomic deletion of both *alms1a,b* genes (*alms1*^del3^, mRNA null allele, see map and characterisation in Appendix Fig. S1). While centriole duplication in *alms1*^del3^ flies is largely normal (Fig. EV3B), we observed a premature disjunction of centrioles in 26% of centriole pairs of *alms1*^del3^ testes (Fig. EV3B middle and focus b, EV3C). This phenotype was fully rescued by expression of Alms1a-Tom (*alms1*^del3^, Alms1a-Tom), together validating the specificity of the *alms1*^del3^ CRISPR deletion and the functionality of our Alms1a-Tom transgene (Fig. EV3B, right, EV3C). In contrast, we confirmed that *alms1a,b* depletion by RNAi in the whole germline, using a *nanos*-Gal4 driver expressed in the GSCs and an RNAi targeting both *alms1a* and *alms1b*, results in the complete loss of centriole duplication (Fig. 4A, middle, only one or two single centrioles observed per testis) (Chen and Yamashita, 2020). In this condition as well, the introduction of extra copies of Alms1a with an Alms1a-Tomato transgene was sufficient to restore centriole duplication in 59% of observed testes (Fig. EV3D). The discrepancy between the two extreme phenotypes resulting from the acute *alms1* depletion by RNAi or the chronic *alms1* loss in *alms1*^del3^ flies could be due to either an off-target effect of the *alms1*^RNAi^ used or to compensation mechanisms triggered by the chronic loss of *alms1a,b*. To discriminate between these two possibilities, we performed *alms1* depletion by RNAi in *alms1*^del3^ mutants (Fig. 4A). If off-target effect was to occur, *alms1*^RNAi^ should induce identical loss of centrioles in WT and *alms1*^del3^ backgrounds. However, expressing *alms1*^RNAi^ in *alms1*^del3^ flies leads to the same phenotype as *alms1*^del3^ flies, *i.e.* no centriolar loss (Fig. 4A, a compared to c), whereas expressing *alms1*^RNAi^ in a control background leads to complete centriolar loss (Fig. 4Ab). *alms1*^del3^ flies have thus developed compensatory mechanisms which alleviate the loss of Alms1 proteins, indicating that Alms1 proteins play a critical role in centriole biogenesis during male spermatogenesis which can be highly buffered during development.

## Alms1 proteins are general regulators of centriole duplication

To determine whether this function in centriole duplication is a general feature of Alms1 proteins, we extended our study to later stages of centriole duplication in the male germline and to the two other tissues studied above (Fig. 4B–F). We depleted both *alms1a* and *b* in these tissues by expressing the *alms1*^RNAi^ with the appropriate tissue-specific Gal4 drivers. We used *bam*-Gal4 (*bag of marbles*) (Chen and McKearin, 2003) to induce *alms1*^RNAi^ expression in symmetrically dividing spermatogonial cysts, between 4 and 16-cells stages (Demarco et al, 2014) (Fig. 4B–D; Appendix Fig. S2A). While control SC (*bam*-Gal4>*lacZ*^RNAi^) contain two centrosomes with one centriole pair each, for a total of 4 centrioles per cell (Fig. 4B: 4C, 100% of cells), *bam*-Gal4>*alms1*^RNAi^ cells contain only one centriole (Fig. 4B: 1C, 92.1% of cells) or no centriole (Fig. 4B: 0C, 6.7% of cells). This distribution is consistent with a centriole duplication failure event in: (i) all cells within 8-cells spermatogonial cysts (each giving 2 cells with one centriole after the 4th round of mitosis); and (ii) in some cells at the 4-cells stage (giving two cells without centrioles and two with one centriole after the last round of mitosis, Appendix Fig. S2B). More, using U-ExM, we confirmed that most of the centrioles still present in *bam*-Gal4>*alms1*^RNAi^ are mother centrioles that fail to assemble procentrioles as visualised with acetylated α-tubulin (94.1% of MC) (Fig. 4C). The remaining centriole pairs (5.9% of centrioles) likely arise from a duplication event at the SG stage, in cells where the *alms1*^RNAi^ was not activated yet. Together, these findings demonstrate that Alms1 proteins are crucial for centriole duplication in *Drosophila* male germline and that this role is not restricted to GSCs as previously proposed (Chen and Yamashita, 2020) but applies to all dividing germline cells. Strikingly, we observe that all the unduplicated centrioles present a normal distribution of Alms1a (Fig. EV4A,B), indicating that centriolar duplication events

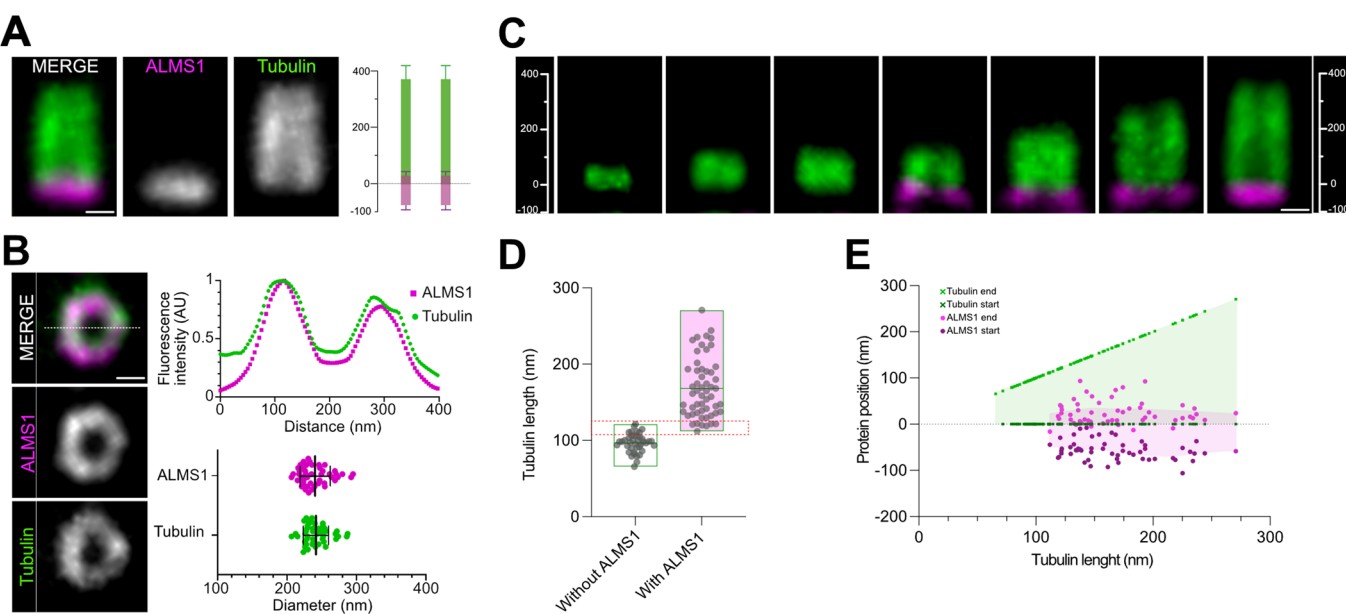

**Figure 2. Localisation of human ALMS1 during procentriole formation and at mature centriole.**

U-ExM images of hTERT-RPE-1 cells showing (**A**) side and (**B**) bottom views of a mature centriole (green) revealing ALMS1 (detected with an antibody directed against ALMS1, magenta) localisation as a ring capping the proximal extremity of the centriolar wall. (**A, B**) Graphs showing the position of ALMS1 with respect to the centriolar wall in longitudinal (**A**) and radial (**B**) dimensions. Data from $N = 3$ independent experiments, $n = 72$ centrioles (**A**) and $n = 46$ (**B**) ALMS1 length: 107.6 nm, Tubulin length: 389.1 nm (**A**). ALMS1 diameter: 240.1 nm, Tubulin diameter: 241.7 nm. Error bars correspond to s.d. (**B**) The fluorescence intensity along the white dotted line is plotted as a function of distance. Error bars correspond to s.d. (**C**) Dynamics of ALMS1 recruitment at forming procentrioles. (**D, E**) Quantifications of ALMS1 localisation with respect to the centriolar wall revealing the onset of ALMS1 recruitment as procentrioles reach 120 nm in length. (**D**) Size of procentrioles without ($n = 39$) or with ($n = 54$) ALMS1 capping the proximal side. (**E**) Evolution of the start and end position of ALMS1 signal relative to tubulin during procentriole growth. The light green and pink regions depict the centriole length (green) and region of the centriole which is covered by ALMS1 signal (pink). Scale bars (after expansion factor correction): 100 nm. Source data are available online for this figure.

do not depend on the pool of Alms1a surrounding the mother centriole but more likely rely on a new pool of Alms1a recruited at procentrioles (Fig. 1B, SG). In agreement with these observations, unduplicated centrioles also show a normal ultrastructure by EM (Fig. 4D), while no procentriolar structures could be detected. Together, these observations indicate that Alms1 proteins are involved in the initiation of all centriole duplication events in the male germline.

We then used the *worniu*-Gal4 driver (*wor*-Gal4) (Lai et al, 2012) to express the *alms1*[RNAi] in the larval brain. Control larval brains (*wor*-Gal4>*lacZ*[RNAi]) display centrosomes in GMCs and in NBs while *wor*-Gal4>*alms1*[RNAi] brains show a strong reduction of GMCs with centrosomes and a total loss of centrosomes in NBs (Fig. 4E), consistent with a role of Alms1 proteins in centriole duplication during NBs asymmetric division. We also generated embryos depleted in maternally provided *alms1a,b* transcripts (from females expressing *alms1*[RNAi] under the control of *nanos*-Gal4) (Van Doren et al, 1998) to investigate the contribution of Alms1 proteins in symmetric divisions in somatic tissues. While control syncytial embryos (from *nanos*-Gal4>*lacZ*[RNAi] females) undergo symmetrical and synchronous nuclear division cycles, *alms1*[RNAi]-derived embryos exhibit strong mitosis defects including multipolar spindles characteristic of centriole duplication failure (Fig. 4F).

Collectively, our results thus demonstrate that Alms1 proteins are required for centriole duplication in both symmetrically and asymmetrically dividing cells of the soma or germline.

## Alms1 proteins are required for cartwheel formation

The formation of the procentriole is the result of a complex cascade of phosphorylations initiated by the recruitment, concentration and transactivation of Plk4 as a single dot on the side of the mother centriole (Lopes et al, 2015; Sonnen et al, 2012). Activated Plk4 phosphorylates Ana2 in two steps: first on serine 38 (localised in the ANST motif at Ana2 N-terminus), allowing the efficient loading of Ana2 and the reinforcement of Plk4 localisation and activation at the duplication site (Dzhindzhev et al, 2017; McLamarrah et al, 2018; Moyer and Holland, 2019; Moyer et al, 2015; Ohta et al, 2014, 2018), then on the STAN motif, triggering the recruitment of Sas-6 by Ana2 (Dzhindzhev et al, 2014; McLamarrah et al, 2018; Moyer et al, 2015) and the formation of the cartwheel onto which the procentriole assembles (Fig. 5A).

In control testes, a dot of Sas-6-GFP is observed on the side of most mother centrioles initiating centriole duplication and subsequently in the proximal lumen of all growing procentrioles (Fig. 5B, U-ExM in *bam*-Gal4> *lacZ*[RNAi]). Sas-6-GFP is detected at the proximal end of the mother centriole in both *alms1*[RNAi] and control conditions (U-ExM, Fig. 5C), in agreement with the conservation of the cartwheel after centriole elongation in *Drosophila* (Nigg and Holland, 2018). In contrast, we never observed Sas-6-GFP affixed to mother centrioles in late spermatogonia nor in spermatocytes in *bam*-Gal4>*alms1*[RNAi] (Fig. 5C). Hence, in the absence of Alms1 proteins, the cartwheel of Sas-6 fails to assemble at the duplication site.

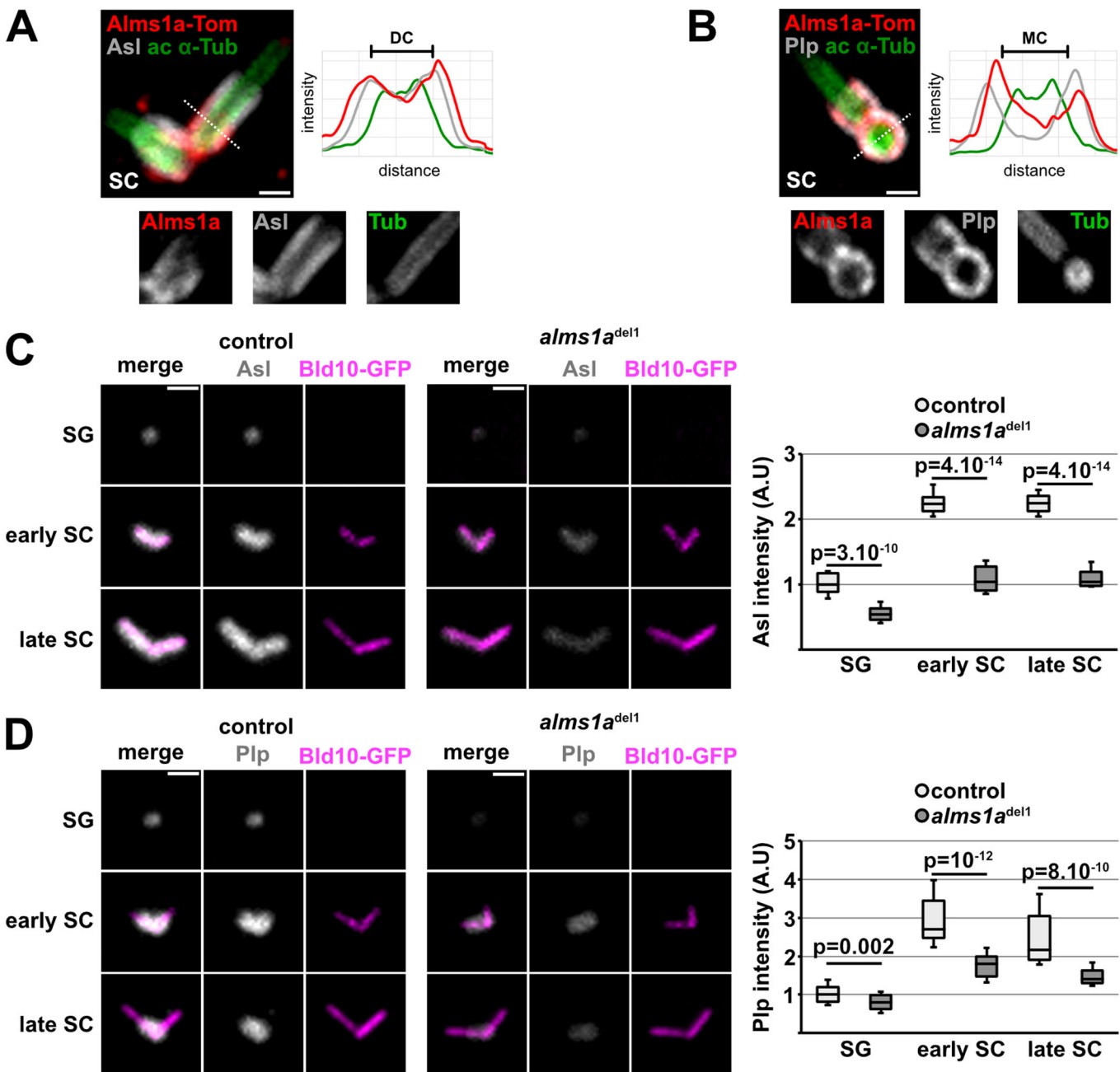

**Figure 3. Alms1a is an inner PCM protein.**

(A, B) U-ExM images of Alms1a-Tomato (red, in *alms1a*<sup>del1</sup>;;Alms1a-Tom) relative localisation with (A) Asl (grey) and (B) Pericentrin-like protein (Plp, grey) in SC. The fluorescence intensity along the white dotted line is plotted as a function of distance. Bracket: centriolar wall width. Standard images of (C) Asl and (D) Plp in *w*<sup>1118</sup> (control) or *alms1a*<sup>del1</sup> in SG and SC and related fluorescence intensity quantification. In both conditions, Bld10-GFP (magenta) is expressed and used as an internal control for the quantification. Ten centrioles were quantified per stage and per testis. Here, we show two independent experiments that were quantified blind and pooled. For Asl: control: $n = 70$ centrioles per stage, 7 testes, *alms1a*<sup>del1</sup>: $n = 80$ centrioles per stage, 8 testes. For Plp: control $n = 80$ centrioles per stage, 8 testes, *alms1a*<sup>del1</sup>: $n = 80$ centrioles per stage, 8 testes. The box plots show the interquartile range (IQR), with the median (50th percentile) indicated by the horizontal line inside the box and the whiskers extending to the 10th and 90th percentiles. A two-sided unpaired Wilcoxon test was performed. (A, B) Scale bars (corrected from expansion factor): 250 nm; (C, D) scale bars: 1 µm. Source data are available online for this figure.

## Alms1 proteins are required for Ana2 stabilisation at the centriole duplication site

The lack of cartwheel in *alms1*<sup>RNAi</sup> might reflect a role of Alms1 proteins either in the recruitment of Ana2 at the site of centriole duplication or in the recruitment of Sas-6 by Ana2.

We thus characterised Ana2 localisation in control and *alms1*<sup>RNAi</sup> conditions by U-ExM using a Ana2-mNeonGreen knock-in line (Ana2-eNG) (Steinacker et al, 2022). In control SG,

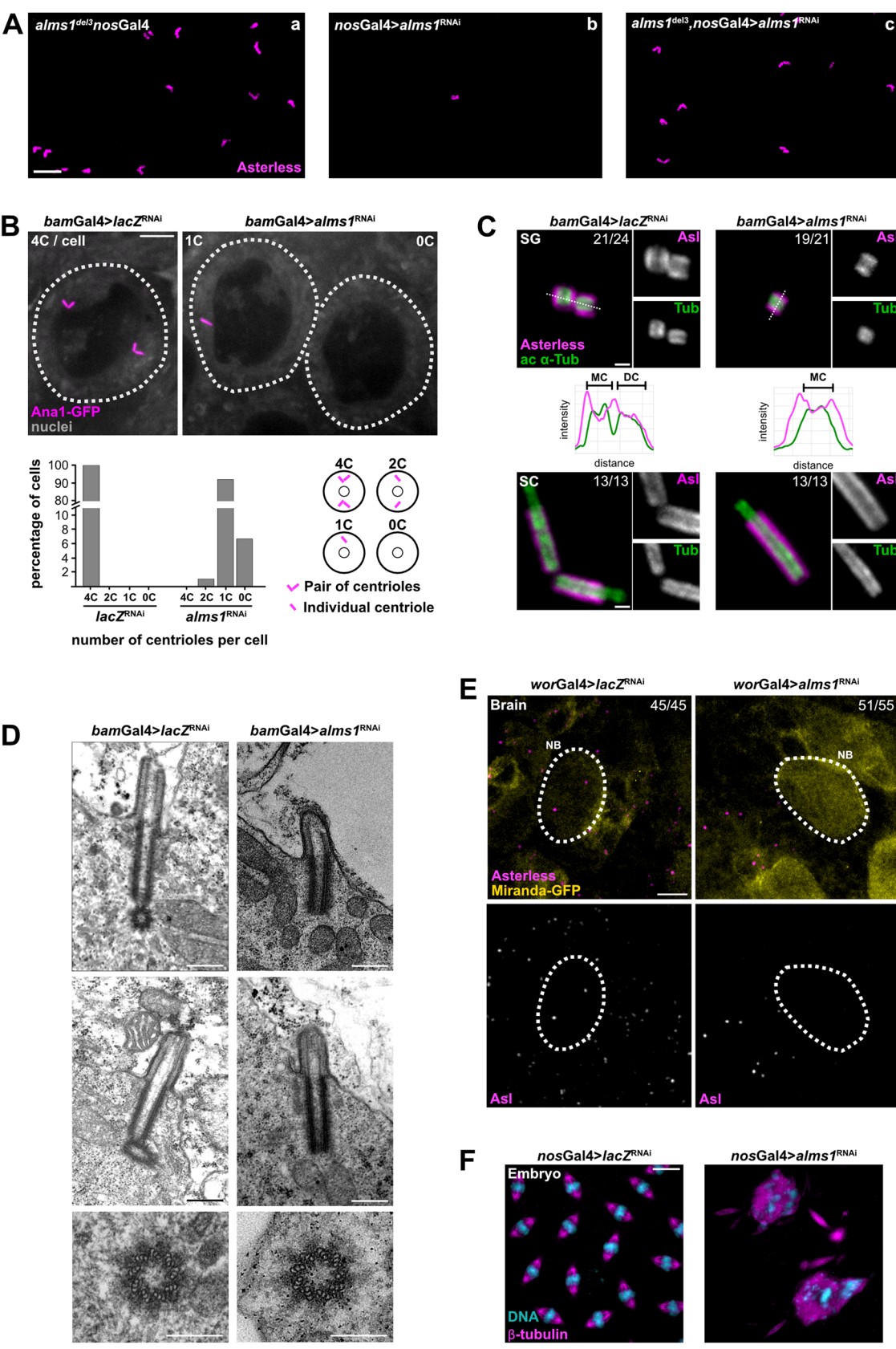

◀ **Figure 4. The RNAi depletion of Alms1 proteins leads to centriole duplication failure in various cell lineages.**

(A) Unexpanded conventionally fixed testis expressing *nos*-Gal4>*alms1*[RNAi] in *alms1*[del3] background (c) characterised by numerous centrioles per testis, a phenotype identical to the one displayed by *alms1*[del3] testis (a). In contrast, *nos*-Gal4>*alms1*[RNAi] driven in *w*[1118] (b) shows a sharp decrease in the centriole number, with only few single centrioles observed per testis. (B) Images of conventionally fixed *bam*-Gal4>*lacZ*[RNAi] or *alms1*[RNAi] SC (outlined with white dotted lines) expressing Ana1-GFP (magenta) as a centriolar marker. Nuclei (grey). The observed distribution of the number of centrioles per spermatocytes (4C, 2C, 1C or 0 C) in the indicated genotypes (*lacZ*[RNAi]: $n = 36$ cells, 3 testes; *alms1*[RNAi]: $n = 89$ cells, 5 testes). *bam*-Gal4>*lacZ*[RNAi] and *bam*-Gal4>*alms1*[RNAi] distributions are significantly different, two-sided unpaired Fisher exact test: $P = 2 \times 10^{-16}$. (C) U-ExM images of *bam*-Gal4>*lacZ*[RNAi] or *alms1*[RNAi] centrioles in SG and SC. Centriolar walls are revealed with acetylated α-tubulin (green, tub). Asl (magenta) as a marker of PCM. On each image, numbers indicate the occurrence of the phenotype (*lacZ*[RNAi]: $n = 37$ centrioles, 2 testes; *alms1*[RNAi]: $n = 34$ centrioles, 2 testes). *alms1*[RNAi] KD results in a significant loss of centriole duplication compared to *lacZ*[RNAi], two-sided unpaired Fisher exact test: $P = 2 \times 10^{-14}$. The fluorescence intensity along the white dotted line is plotted as a function of distance. (D) TEM images of *bam*-Gal4>*lacZ*[RNAi] or *bam*-Gal4>*alms1*[RNAi] mature primary spermatocytes confirming the absence of procentriole formation in *alms1*[RNAi]. Unduplicated centrioles show no structural defects in the centriole wall (bottom right image) and induce cilia formation (bottom left and middle images). (E) Unexpanded images of *wor*-Gal4>*lacZ*[RNAi] or *wor*-Gal4>*alms1*[RNAi] larval brain (*lacZ*[RNAi]: 45/45 observed NBs with centrosomes, 3 brains; *alms1*[RNAi]: 51/55 observed NBs lack centrosomes, 5 brains). *alms1* RNAi KD results in a significant loss of centriole duplication compared to *lacZ*[RNAi], two-sided unpaired Fisher exact test: $P = 2 \times 10^{-16}$. NBs (outlined with white dotted lines) are identified based on Miranda-GFP expression (yellow). Asl as a centriolar marker (magenta). (F) Unexpanded images of *nos*-Gal4>*lacZ*[RNAi] or *nos*-Gal4>*alms1*[RNAi] early embryos. β-tubulin for the mitotic spindle (magenta) and DNA (cyan). Scale bars: (A, B, E, F) 5 μm, (C) (after expansion factor correction) 250 nm, (D) 5 μm on top and middle images and 200 nm on bottom images. Source data are available online for this figure.

Ana2-eNG localises at the proximal inner end of the mother centriole and forms a dot at the base of the forming procentriole (Fig. 5Da,b) (Dzhindzhev et al, 2017; McLamarrah et al, 2020). As daughter centrioles elongate, we observed a shift in Ana2 abundance from the mother to the daughter centriole. Whereas in early duplicating centrioles Ana2 is more abundant at the mother centriole than at the nascent procentriole (Fig. 5Da, 55% of SG short centrioles), it becomes more abundant at daughter centrioles as they grow (all centrioles in SG long, Fig. 5Dc), to finally fully disappear from some mother centrioles ($n = 8/32$ SC, Fig. EV5A) (McLamarrah et al, 2018, 2020).

In *alms1*[RNAi] condition, Ana2-eNG is also nested in the proximal end of all remaining mother centrioles in SG and SC. However, we failed to detect any Ana2-eNG on the side of 58% of the unduplicated centrioles (Fig. 5Dd,f for SG, SC see Fig. EV5B), whereas for the remaining 42% we observed a very faint Ana2-eNG staining at the centriole duplication site, almost always less abundant than the concentration at the base of the unduplicated centriole (Fig. 5De,g for SG, SC see Fig. EV5B). Altogether our observations thus suggest that Alms1 proteins are involved in the efficient recruitment or stabilisation of Ana2 at the duplication site and that this reduced Ana2 pool is responsible for the lack of Sas-6 recruitment.

## Alms1 proteins are required for Plk4 function in centriole duplication

Ana2 recruitment at the procentriole is under the direct control of Plk4 (Dzhindzhev et al, 2017). In wild-type conditions, Plk4 activity is tightly buffered to avoid the formation of supernumerary duplication sites and hence centriole overduplication. This control can be overcome by Plk4 overexpression (Dzhindzhev et al, 2017; Habedanck et al, 2005; Park et al, 2019). Hence, in *Drosophila* spermatogonia, increased expression of Plk4 leads to extra-centrioles organised in rosettes and de novo centriole formation (Lopes et al, 2015). In agreement with this latter result, Plk4 overexpression (Plk4[OE]) in our control background (*bam*-Gal4 > {*lacZ*[RNAi], Plk4}) induces rosettes containing more than three centrioles (Fig. 6A, >3C). In contrast, when Plk4 is overexpressed in the absence of Alms1 proteins (Fig. 6A, *bam*-Gal4 > {*alms1*[RNAi], Plk4}), the centrioles remain unduplicated except for an extremely low percentage of centriole pairs (less than 1%,). Thus, Alms1 proteins are required for Plk4-driven centriole

overduplication. More, while overexpression of Alms1a alone doesn't induce centriole overduplication, co-expression of Alms1a together with Plk4 enhances centriole overduplication compared to overexpression of Plk4 alone (respectively 76% and 57% of centrosomes with overduplicated centrioles, Fig. 6B,C). Altogether these results suggest a role of Alms1 proteins in either Plk4 activity or recruitment at duplication sites.

In spermatogonia depleted for Alms1 proteins, the endogenous spot of Plk4 (Plk4-eGFP) (Nabais et al, 2021) is still detected on the side of 91.7% of the unduplicated centrioles but with reduced concentration when compared to control condition (Fig. 5E; Plk4-eGFP staining intensity at ds/daughter centriole, SG: *alms1*[RNAi] = 0.54 A.U., *lacZ*[RNAi] = 1 A.U.; SC: *alms1*[RNAi] = 0.67 A.U., *lacZ*[RNAi] = 1.92 A.U.). Together, these results indicate that Alms1 proteins are not strictly required for the recruitment of Plk4 at the duplication site, but are necessary to reach a concentration of Plk4 sufficient to elicit duplication, either by enhancing its recruitment or by stabilising it once it is recruited.

Plk4 is a suicidal kinase: above a local threshold concentration, Plk4 trans-autophosphorylates two residues of its degron motif (Ser293 and Thr297), thus eliciting its own degradation by the SCF-Slimb/βTrCP-E3 ubiquitin ligase complex and the proteasome machinery (Cunha-Ferreira et al, 2013, 2009). We reasoned that if Alms1 proteins are involved in stabilising Plk4 protein, expression of a non-degradable form of Plk4 (with the point mutations S293A and T297A) (Cunha-Ferreira et al, 2009) could rescue the centriole duplication defects associated with *alms1*[RNAi]. In the control condition, overexpression of a non-degradable Plk4 (*bam*-Gal4 > {*lacZ*[RNAi], Plk4[ND-GFP]}) induces numerous ectopic centrioles in SG, either organised in clusters or isolated, suggesting that we can achieve overduplication of centrioles (Fig. 6D,E) (Cunha-Ferreira et al, 2009). In contrast, when Plk4[ND-GFP] is overexpressed in conditions where Alms1 is depleted (*bam*-Gal4 > {*alms1*[RNAi], Plk4[ND-GFP]}), centrioles remain unduplicated (Fig. 6D,E). Interestingly, overexpression of ND-Plk4 in *alms1a,b* deleted flies (*alms1*[del3]) leads to less centrosomes with overduplication (three centrioles) than overexpression in control conditions in which numerous rosettes are observed (*alms1*[del3], *nos*-Gal4 > ND-Plk4-GFP compared to *nos*-Gal4 > ND-Plk4-GFP, Fig. 6G). This indicates that despite the compensation of Alms1 loss operating in the chronic (RNA null) mutants, Alms1 remains required for over-amplification of centrioles in these overexpression assays.

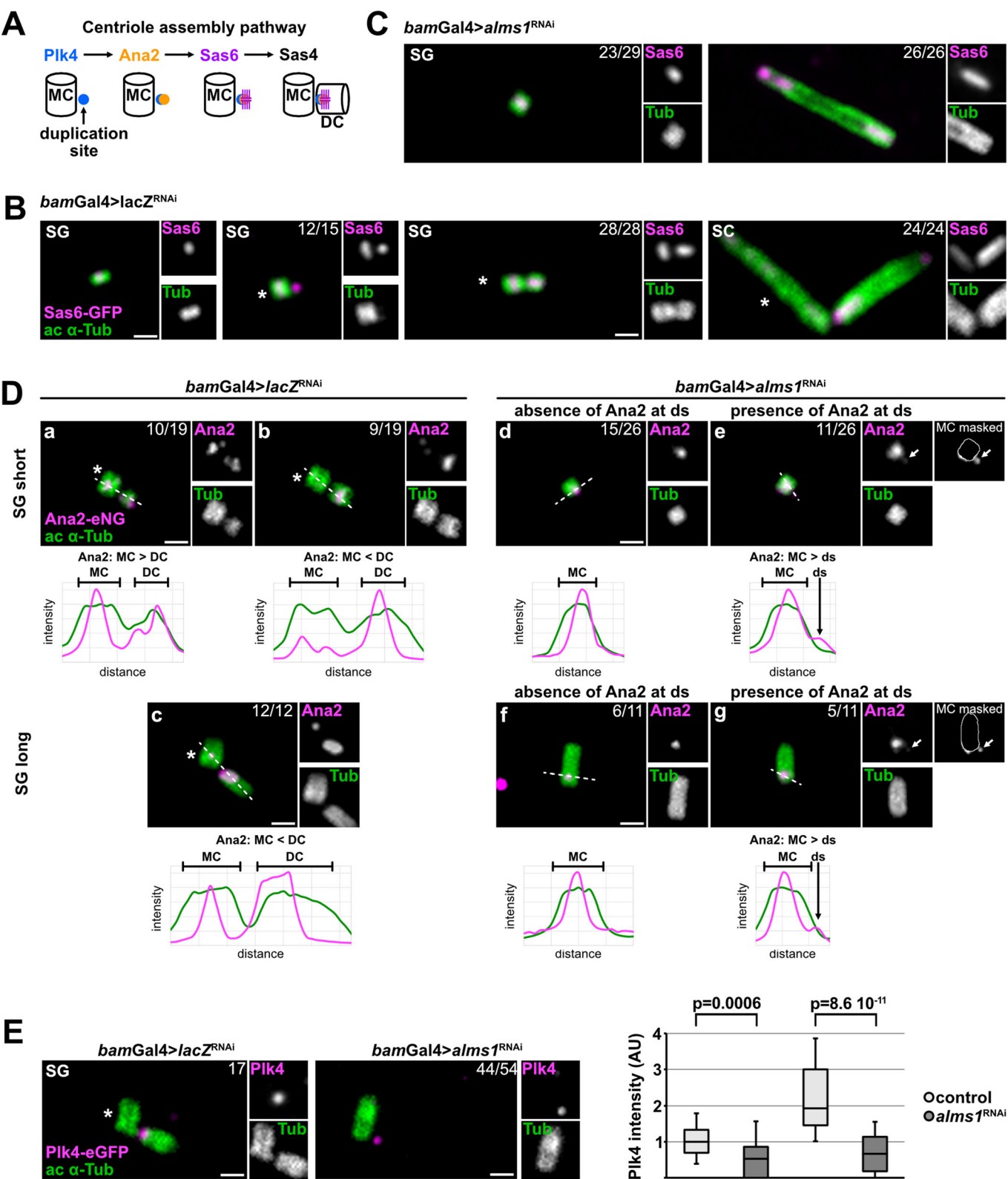

**Figure 5.  Alms1 proteins are essential for Sas-6 recruitment and required to stabilise Ana2 and Plk4.**

(A) Scheme representing part of the known protein hierarchy responsible for centriole duplication, leading to procentriole formation. (B) U-ExM images of Sas-6-GFP (in magenta) in *bam*-Gal4>*lacZ*[RNAi] in SG (a) before Sas-6 recruitment at the duplication site, (b) when Sas-6 is at the duplication site, (c) when procentriole has formed and in SC. (C) U-ExM images of Sas-6-GFP (in magenta) in *bam*-Gal4>*alms1*[RNAi] in SG and SC. (B, C) Centriolar walls are revealed with acetylated α-tubulin (green, ac α-tub), (*lacZ*[RNAi]: 4 testes; *alms1*[RNAi]: 6 testes). (D) U-ExM images of Ana2-mNeonGreen knock-in (Ana2-eNG, in magenta) in *bam*-Gal4>*lacZ*[RNAi] or *bam*-Gal4>*alms1*[RNAi]. Centrioles at the SG stage are divided into two categories: SG short, when the length of the centrioles does not exceed their width and SG long, when their length exceeds their width (*lacZ*[RNAi]: 3 testes; *alms1*[RNAi]: 5 testes). The fluorescence intensity along the dotted line is plotted as a function of distance. Mother centriole (MC, on the left of all images) and daughter centriole (DC, on the right) are delineated by brackets, duplication site (ds) by an arrow. (E) U-ExM images of Plk4-meGFP knock-in (in magenta) in *bam*-Gal4>*lacZ*[RNAi] or *bam*-Gal4>*alms1*[RNAi] in SG (*lacZ*[RNAi]: 3 testes; *alms1*[RNAi]: 6 testes). Quantification of Plk4-meGFP fluorescence intensity performed on unexpended testis (*lacZ*[RNAi]: $n = 4$ testes; *alms1*[RNAi]: $n = 5$ testes) is shown on the right. The box plots show the interquartile range (IQR), with the median (50th percentile) indicated by the horizontal line inside the box and the whiskers extending to the 10th and 90th percentiles. Two-sided unpaired Wilcoxon test. In all images, the MC is on the left and marked with an asterisk. Scale bars (after expansion factor correction): 250 nm. Source data are available online for this figure.

In addition, in the control condition, Plk4[ND-GFP] localises at the base of most centrioles contained within a rosette (Fig. 6F) and forms large and very bright aggregates in SG undergoing the 3rd and 4th mitosis and early SC as previously described (Cunha-Ferreira et al, 2013) (Fig. EV6A). In contrast, we only observed a faint Plk4[ND-GFP] concentration on the side of the unduplicated centrioles in *alms1*[RNAi] (Fig. 6F, white arrow), in agreement with the localisation observed for endogenous Plk4-eGFP (Fig. 5E), and failed to detect Plk4[ND-GFP] cytoplasmic clustering (Fig. EV6A).

Altogether, these results suggest that Alms1 proteins are required to stabilise Plk4 clusters and that this function is independent of the degron-mediated regulation of Plk4 stability.

These experiments thus demonstrate that, in *Drosophila*, Alms1 proteins are required for Plk4 function in centriole duplication, in the regulation pathway that contributes to Plk4 recruitment or stabilisation at mother centrioles.

### Alms1 proteins are likely involved in the positive feedback loop between Plk4 and Ana2

Stabilisation of Plk4 at the duplication site relies on a positive feedback loop between Plk4 and Ana2 which mutually contribute to their concentration and activation (Dzhindzhev et al, 2017; McLamarrah et al, 2018; Moyer and Holland, 2019; Moyer et al, 2015; Ohta et al, 2014, 2018). We thus investigated the possibility that Alms1 could stabilise Ana2 and hence indirectly Plk4. Ana2 overexpression (Ana2[OE]) in WT background (*bam*-Gal4>Ana2) results in mild overduplication of centrioles with 15% of centrosomes containing 3 centrioles. In contrast to what is observed for Plk4, overexpressing Alms1a with Ana2 does not potentialize the overduplication observed, but rather results in a small but significant reduction of centriole overduplication (Fig. 7A), suggesting that other factors are limiting. Considering the mutual reinforcement between Ana2 and Plk4, we reasoned that increasing both Plk4 and Ana2 could be sufficient to by-pass Alms1 requirement. Indeed, while overexpression of ND-Plk4 is not sufficient to rescue the centriole duplication defect associated with *alms1* depletion (Fig. 7B,C, *bam*-Gal4 > {*alms1*[RNAi], ND-Plk4}), and overexpression of Ana2 only rescues centriole duplication in few centrosomes (10%, Fig. 7B,C, *bam*-Gal4 > {*alms1*[RNAi], Ana2}), overexpression of both Ana2 and ND-Plk4 largely rescues the centriole duplication defect associated with *alms1* [RNAi], with 64% of the centrosomes composed of 2 centrioles (Fig. 7B,C, *bam*-Gal4 > {*alms1*[RNAi], ND-Plk4, Ana2}).

We therefore propose that, in *Drosophila*, Alms1 proteins contribute to centriole duplication by potentializing Plk4-Ana2

interaction required for the positive feedback loop (Fig. 7D), thus explaining how Alms1 can be both upstream of Plk4 (required for its localisation/stabilisation at the duplication site) and downstream of Plk4 (required for its activity in centriole duplication). Hence, Alms1 contributes to Plk4 recruitment or stabilisation at mother centrioles, increased Ana2 activation and subsequent cartwheel formation.

## Discussion

Here we show that, in *Drosophila*, Alms1 proteins are regulators of centriole duplication as the acute depletion of both *alms1a* and *alms1b* by RNAi results in complete failure of centriole duplication in all asymmetrically and symmetrically dividing somatic and germline cells analysed. Even though we cannot discriminate between Alms1a and b by RNAi due to the extreme conservation of their transcripts sequence, we anticipate that Alms1a is the one involved in the duplication process since we show during spermatogenesis that Alms1a is recruited at the onset of procentriole formation whereas Alms1b is recruited after the initiation of procentriole formation (Fig. 1), and centriole duplication is lost upon *alms1* RNAi in early embryos or neuroblasts in which Alms1b is not expressed (Fig. 4E,F).

We propose that Alms1 proteins promote centriole duplication by enhancing Plk4-Ana2 functional interactions, thus contributing to the stabilisation of Plk4 at the centriole duplication site. In the absence of Alms1 proteins the concentration of both Plk4 and Ana2 at the duplication site is reduced, leading to dramatic loss of Sas-6 cartwheel assembly and of nascent procentriole formation.

It has been shown that centriole duplication at a single position relies on the functional interaction of Ana2 with the Plk4 ring initially formed around the centriole. The binding at a single position of Ana2/STIL to Plk4 (Dzhindzhev et al, 2017; Ohta et al, 2018) protects it from degradation by the ubiquitin–proteasome pathway. Ana2/STIL binding indeed prevents Plk4 from auto-phosphorylating its degron motif, which is required for its ubiquitination (Rogers et al, 2009), but also prevents Plk4 ubiquitination independently of the degron motif (Ohta et al, 2014). As a consequence, Plk4 is degraded in the whole ring with the exception of the single Plk4/Ana2 interaction site. This initial interaction of Ana2 to Plk4 also releases a positive reinforcement loop whereby Plk4 phosphorylation of Ana2 stabilises it and, in turn, phosphorylated Ana2 stabilises activated Plk4 (Dzhindzhev et al, 2017; McLamarrah et al, 2020; Moyer and Holland, 2019; Ohta et al, 2014). This loop results in the phosphorylation of Ana2 on its STAN domain, making it competent to recruit Sas-6 and trigger the formation of the cartwheel.

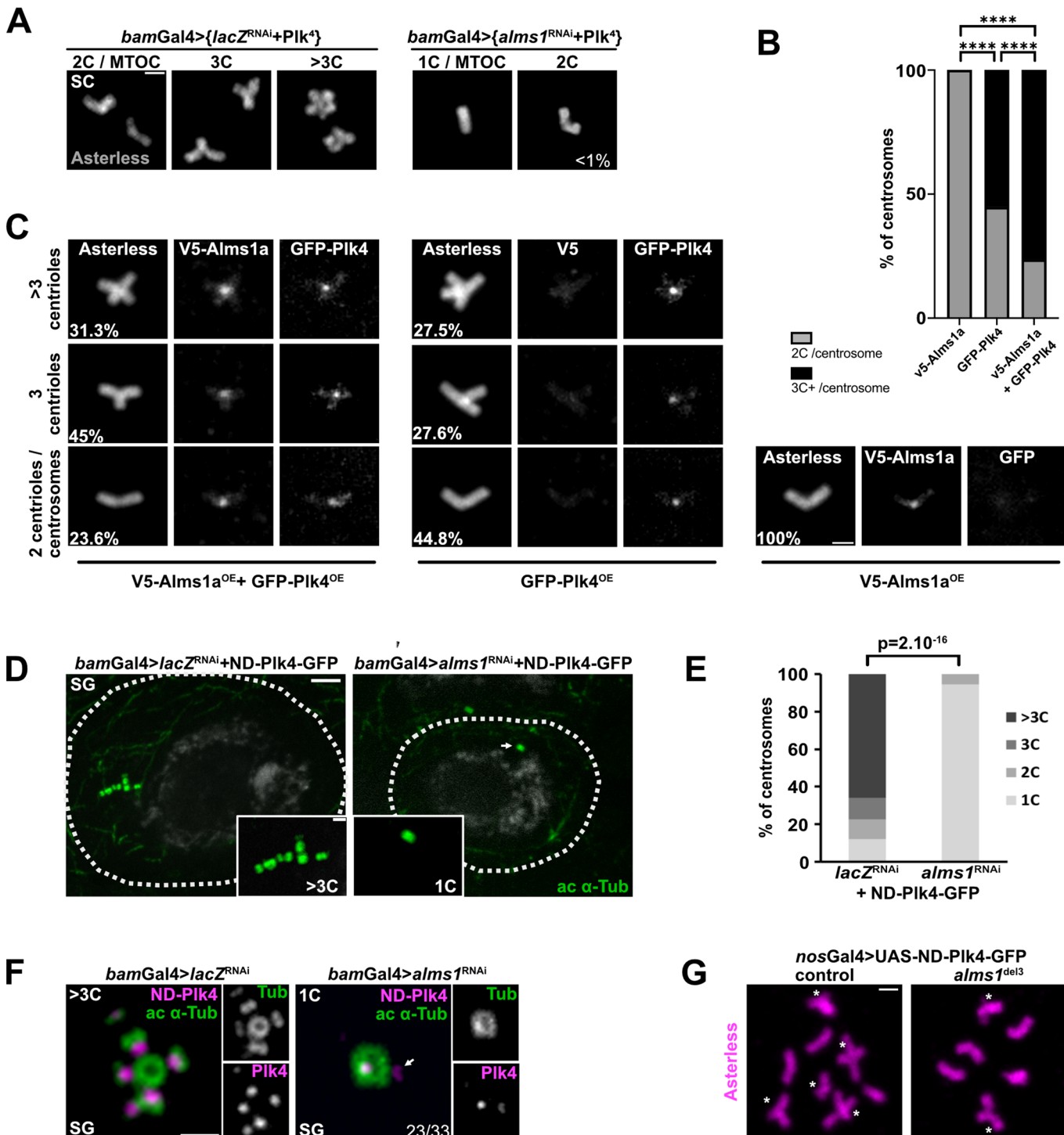

In contrast to isolated mammalian or *Drosophila* cells, we never observed the initial ring of Plk4 around the centriole in *Drosophila* tissues, but directly observed a single dot of Plk4 at the base of the procentriole from the very early steps of centriole duplication (Fig. 5E). After acute depletion of *alms1* by RNAi, we observed that Plk4 is present at a reduced level on centrioles compared to control (Fig. 5E) and Ana2 is either absent in 58% of the centrioles or reduced in the remaining 42% (Fig. 5D), suggesting that the Ana2-Plk4 amplification loop is not set efficiently in absence of Alms1.

Two hypotheses can support this observation: in the absence of Alms1 Plk4 accumulation does not reach a concentration threshold required to switch on the amplification loop, or Alms1 is required to promote Plk4 and Ana2 interactions.

We observed, in agreement with the literature, that overexpression of wild-type or non-degradable (Cunha-Ferreira et al, 2009) Plk4 (with a non-phosphorylable degron motif, ND-Plk4) induces strong centriole overduplication, associated with centriole and cytoplasmic Plk4 accumulation in control flies. It however

**Figure 6. Alms1 proteins are required for Plk4 function in centriole duplication.**

(A) Images of unexpanded *bam*-Gal4>*lacZ*[RNAi] or *bam*-Gal4>*alms1*[RNAi] SC overexpressing Plk4 (Plk4[OE], not tagged). Asl (grey) as a centriolar marker. The different phenotypes observed are shown by genotype: 1, 2, 3 centrioles or more (> 3) per centrosome (MTOC) (*lacZ*[RNAi]: $n = 7$ testes; *alms1*[RNAi]: $n = 9$ testes). (B) Quantification of centrosomes with or without centriole duplication in SC overexpressing Plk4 (Plk4[OE]-GFP), Alms1a (Alms1a[OE]-V5) or both (Alms1a[OE]-V5 +Plk4[OE]-GFP) (two independent experiments are pooled, Plk4[OE]-GFP: $n = 734$ centrioles, 5 testes; Alms1a[OE]-V5: $n = 505$ centrioles, 5 testes; Alms1a[OE]-V5 +Plk4[OE]-GFP: $n = 1054$ centrioles, 5 testes), Fischer's exact test, ****$P = 2 \times 10^{-16}$. (C) Images of unexpanded SC overexpressing Plk4 (Plk4[OE]-GFP), Alms1a (Alms1a[OE]-V5) or both (Alms1a[OE]-V5 +Plk4[OE]-GFP). Percentage of centrosomes with 2, 3 or more (> 3) centrioles (revealed by Asl) is indicated on the images. The overexpressed proteins are revealed with their respective tag (V5 and GFP). (D) U-ExM images of *bam*-Gal4 > {*lacZ*[RNAi], ND-Plk4-GFP} or *bam*-Gal4 > {*alms1*[RNAi], ND-Plk4-GFP}. A close-up of the region with centrosomes (white arrow on the right panel) is presented in the inset and the phenotypes observed are indicated. Acetylated α-tubulin in green as a centriolar marker. (E) Proportion of centrosomes observed with 1 to >3 centrioles per indicated genotypes (*lacZ*[RNAi]: $n = 133$ centrosomes, 3 testes; *alms1*[RNAi]: $n = 55$ centrosomes, 4 testes). Fischer's exact test, $P = 2 \times 10^{-16}$. (F) U-ExM image focusing on a centrosome with the predominant composition per genotype (> 3 centrioles for *lacZ*[RNAi] and 1 centriole for *alms1*[RNAi]). On *alms1*[RNAi], the white arrow points to ND-Plk4-GFP faint concentration on the side of the remaining centriole. ND-Plk4-GFP in magenta, acetylated α-tubulin in green. (G) Images of unexpanded control or *alms1*[del3] SC overexpressing ND-Plk4-GFP (*bam*-Gal4>ND-Plk4-GFP). Overexpression of ND-Plk4-GFP in control background induces a strong overduplication of centrioles with centrosomes often containing more than 3 centrioles while *alms1*[del3] SC overexpressing ND-Plk4-GFP present fewer overduplicated centrioles containing less often more than 3 centrioles. Centrioles labelled with anti Asl (magenta), centrosomes with 3 or more centrioles marked with an asterisk. Scale bars: (A, C, G) 1 µm; after expansion factor correction (D) 1 µm, (D, inset, F): 250 nm. Source data are available online for this figure.

failed to restore Plk4 concentration and centriole duplication in flies depleted for Alms1 proteins (Fig. 6D–F). More, in absence of Alms1 proteins, we did not observe Plk4 accumulation in the cytoplasm as observed in control conditions (Fig. EV6A). These observations indicate that Alms1 proteins play a key role in stabilising PLK4 clusters independently of the phosphorylation of its degron motif. They thus suggest that Alms1 proteins are crucial for the formation of a dot of Plk4 sufficiently concentrated and /or stable to trigger the rest of the duplication pathway.

Stabilisation of Plk4-Ana2 has been proposed to rely on Sas-4/CPAP, which acts as a platform to bring the two proteins in proximity (Dzhindzhev et al, 2017; McLamarrah et al, 2020; Moyer and Holland, 2019). Interestingly, ALMS1 has been identified in BioID screens as a potential interactor of CPAP (Firat-Karalar et al, 2014; Gupta et al, 2015) and is a direct interactor of Plk4 (Chen and Yamashita, 2020). As well, Cep131/AZI1 was recently shown to be a substrate of Plk4 facilitating the Plk4-STIL interaction when phosphorylated (Kim et al, 2019). We speculated that, as proposed for Sas-4/CPAP or for CEP131/AZI1, Alms1 proteins could be involved in stabilising Plk4-Ana2 at the duplication site. In agreement with this hypothesis, while overexpression of Plk4 does not rescue the centriole duplication defect associated with *alms1* depletion and overexpression of Ana2 in *alms1*[RNAi] rescues centriole duplication for a few centrosomes, we observed a strong rescue of centriole duplication upon co-overexpression of Plk4 and Ana2 in *alms1*[RNAi] background. We therefore propose that Alms1 contributes to the amplification loop by promoting Plk4 and Ana2 interactions. Future experiments will be required to position Alms1 proteins relative to the Sas-4-PLK4-Ana2 or the CEP131-PLK4-Ana2 modules.

Our work demonstrates that Alms1a,b are either essential (following acute RNAi depletion) or dispensable (upon chronic loss) intermediate players between Plk4 and Ana2. Such apparent contradictory properties illustrate striking buffering capacities of cells or tissues to maintain centriole numbers during development. This could also explain why Alms1 proteins have remained unidentified so far despite the extensive screens in various models, and in particular in *C. elegans* and *Drosophila*, that led to the identification of the set of conserved proteins at the core of the centriole duplication process (Plk4 (ZYG-1 in *C.e.*), Ana2 (SAS-5), Sas-6 (SAS-6) and Sas-4 (SAS-4)) (Bettencourt-Dias et al, 2005; Delattre et al, 2006; Dzhindzhev et al, 2014; Rodrigues-Martins et al, 2007; Shimanovskaya et al, 2014; Stevens et al, 2010). Several

observations in the literature support the hypothesis that centriole duplication is very sensitive to variations in the expression of core molecular players. In particular, it has been shown that overexpression of centriole duplication factors can induce centrosome amplification and centriole overduplication. Thus, we anticipate that during chronic loss of Alms1a and b, the expression of some core players and/or other components still to be identified are modulated to fine-tune centriolar duplication. Investigations of the compensatory mechanisms involved in *alms1* loss-of-function will likely provide fascinating future knowledge on centriole homoeostasis in cells. Interestingly, such compensatory mechanisms have already been described for other cellular processes in whole organisms (Hall et al, 2013) and thus represent a challenging complexity for their understanding.

The key role of Alms1 in centriole duplication in *Drosophila* that we uncovered in this work raises the question of the conservation of such function in other species and most specifically in humans, as mutations in *ALMS1* are known to lead to Alström syndrome, a severe disorder classified as a ciliopathy (Collin et al, 2002; Hearn et al, 2002). The centriole disjunction observed after chronic loss of *alms1a,b* is also observed in human cells after RNAi depletion of ALMS1 (Knorz et al, 2010), thus indicating that at least this function of Alms1 proteins is likely conserved across evolution. In contrast, detailed examination of ALMS1 dynamics by U-ExM in RPE-1 cells revealed that it is not recruited at the initiation of procentriole assembly but rather appears at the beginning of the procentriole elongation phase, where it forms a ring capping the base of the microtubule wall (Fig. 2B,C). This observation, together with the fact that acute loss (RNAi KD) of ALMS1 is not sufficient to reveal centriolar duplication defects in mammalian cells (Li et al, 2007), suggests that ALMS1 role in centriole duplication in mammals might diverge from the one identified in *Drosophila* or else be shared with other centriolar proteins along which the two other proteins containing an ALMS domain, FATS and CEP295 (Knorz et al, 2010). No centriolar functions have been described for FATS to date. CEP295, on the other hand, is recruited at the onset of procentriole formation and progressively decorates the outer surface of the proximal third of the new centriole (Laporte et al, 2024). This localisation dynamics, together with the role of CEP295 in centriole stability and elongation (Izquierdo et al, 2014; Chang et al, 2016), led to propose that CEP295 is involved in the maintenance of the centriole integrity (Laporte et al, 2024; Meehl et al, 2016). Interestingly, CEP295 dynamics and localisation

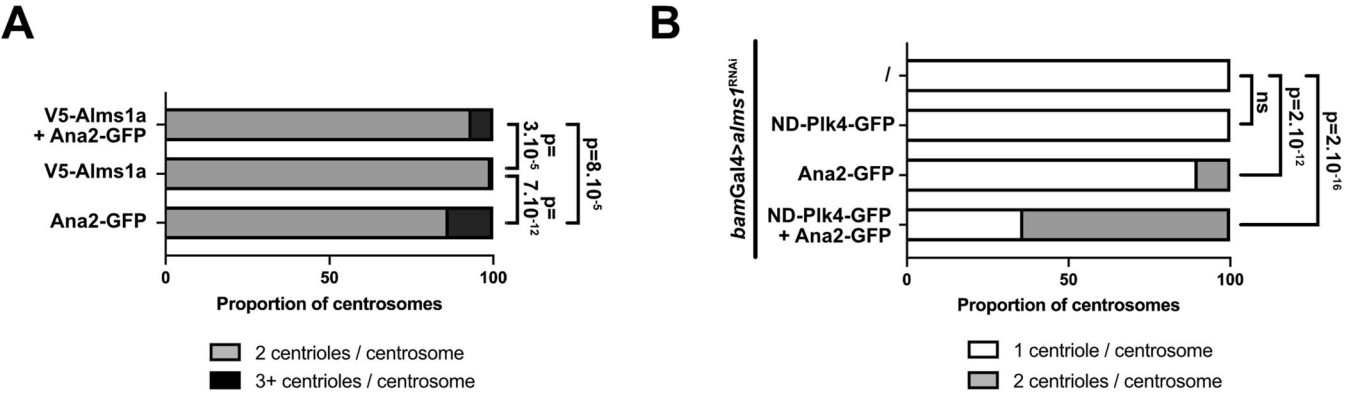

**A**

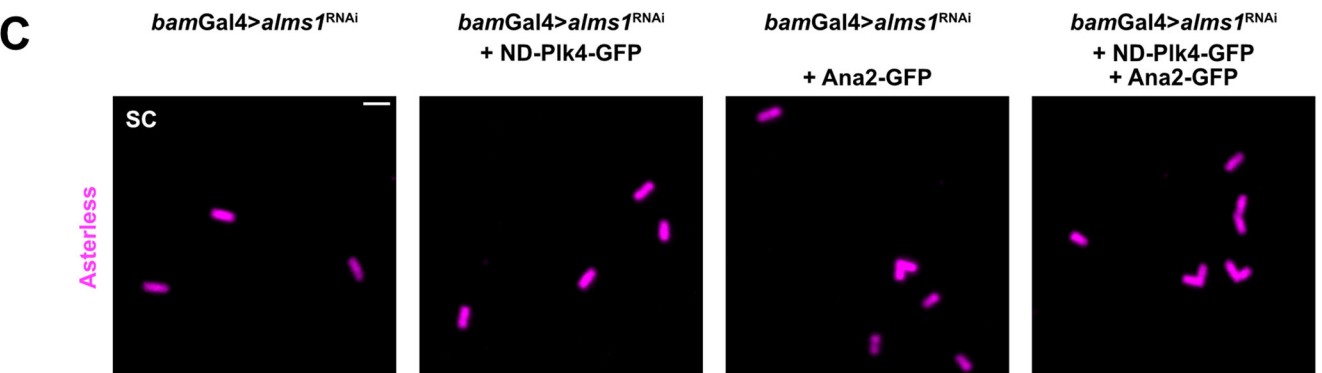

**B**

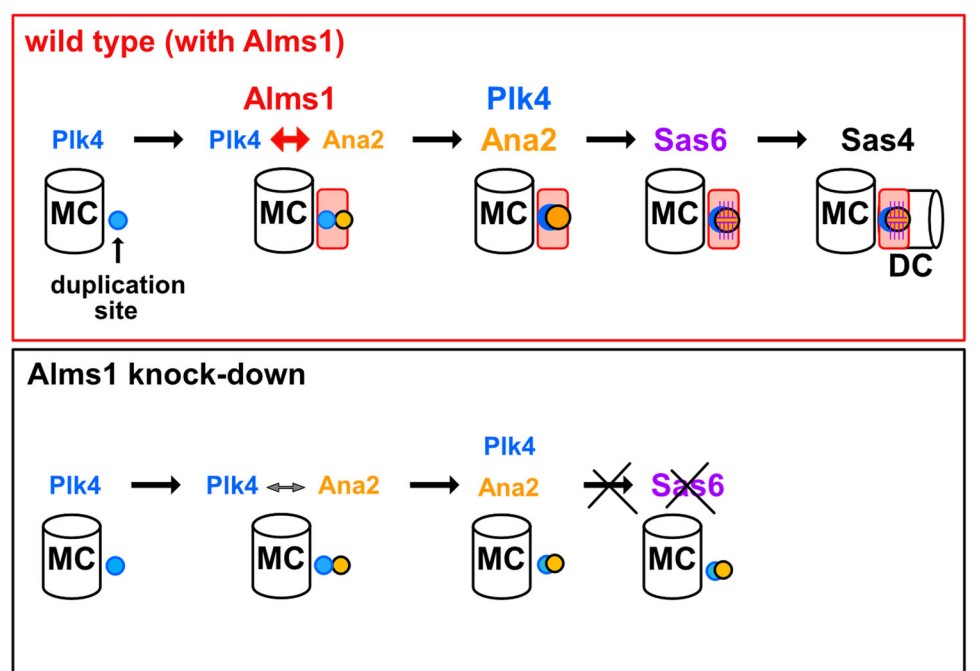

◀ **Figure 7. Alms1 proteins are required for Plk4-Ana2 interaction.**

(A) Quantification of centriole number per centrosome in SC from testis overexpressing Ana2-GFP, Alms1a-V5 or both. Ana2$^{OE}$ results in 13.6% of centrosomes with centriole duplication while 6.6% of centrosomes present centriole duplication upon co-overexpression of Alms1a$^{OE}$-V5 and Ana2$^{OE}$-GFP (Ana2$^{OE}$-GFP: $n = 509$ centrosomes, 7 testes; Alms1a$^{OE}$-V5: $n = 277$ centrosomes, 3 testes; Alms1a$^{OE}$-V5 +Ana2$^{OE}$-GFP: $n = 652$ centrosomes, 3 testes). Two-sided unpaired Fisher exact test. (B) Quantification of centriole number per centrosome in SC from testis bam-Gal4>alms1$^{RNAi}$ or bam-Gal4>alms1$^{RNAi}$ overexpressing ND-Plk4-GFP, Ana2-GFP or both (bam-Gal4>alms1$^{RNAi}$: $n = 303$ centrosomes; +ND-Plk4-GFP: $n = 552$ centrosomes; +Ana2-GFP: $n = 753$ centrosomes; +{ND-Plk4-GFP, Ana2-GFP}: $n = 1348$ centrosomes). Two-sided unpaired Fisher exact test. (C) Images of unexpanded SC bam-Gal4>alms1$^{RNAi}$ or bam-Gal4>alms1$^{RNAi}$ overexpressing ND-Plk4-GFP, Ana2-GFP or both. While only unduplicated centrioles are observed in bam-Gal4>alms1$^{RNAi}$ overexpressing ND-Plk4-GFP, 10% of centrosomes contain two centrioles upon Ana2-GFP overexpression while 64% of centrosomes are composed of doublet in bam-Gal4>alms1$^{RNAi}$ overexpressing both Ana2-GFP and ND-Plk4-GFP. Related controls are shown in Fig. EV6B. Asl (magenta) as a centriolar marker. Scale bar: 1 μm. (D) Model of Alms1 role in centriole duplication. We propose that Alms1 facilitates/potentializes the interaction between Plk4 and Ana2 at the onset of centriole duplication. By promoting the positive feedback loop between Ana2 and Plk4, Alms1 would hence allow for an efficient recruitment of Plk4 at the duplication site (ds), the concentration and activation of Ana2 and the subsequent formation of the cartwheel of Sas-6. In absence of Alms1, the amplification between Plk4 and Ana2 is not sufficient and Ana2 fails to recruit Sas-6. Source data are available online for this figure.

present similarities with the one we observed for Alms1a in *Drosophila* (Fig. 1B), questioning a role for Alms1a in daughter centriole stability and maturation in addition to its role in centriole duplication per se.

In conclusion, our work shows that Alms1 proteins are general regulators of centriole duplication in flies, involved in the complex step of Plk4-Ana2 stabilisation and amplification required to initiate cartwheel formation. It also reveals striking, but still to be understood, buffering capacities of cells and tissues during development to compensate for *alms1* loss-of-function, highlighting their critical role in centriole duplication. Understanding if Alms1 function in centriole duplication is conserved in humans is a future challenge as *ALMS1* is the only gene associated with the extremely rare Alström syndrome in humans (Collin et al, 2002; Hearn et al, 2002).

## Methods

### Fly stock and husbandry

Flies were raised at 18 °C, 25 °C or 29 °C on a standard nutrient medium (cornmeal, yeast, agar, nipagin, ethanol).

All experiments involving flies conform to the relevant regulatory standards (DUO6403). Fly lines generated and stocks used are listed in Tables 1 and 2.

### Plasmids and *Drosophila* gene constructs

All primer sequences are listed in Appendix Table S1. All transgenic constructs were injected by BestGene Inc.

### *Drosophila* Alms1 reporter gene constructs

Alms1a-Tomato was obtained by cloning the PCR product (F-Alms1a-Tom/R-Alms1a-Tom), containing 1.73 kb upstream regulatory sequences and the entire coding sequence (4.23 kb), in frame with Tomato in BglII-NotI sites of pJT108 (Vieillard et al, 2016).

For Alms1b-GFP construct, the PCR product (F-Alms1b-GFP/R-Alms1b-GFP) including 1.5 kb upstream regulatory sequences and the entire coding sequence (3.3 kb) was cloned in frame with GFP in the BglII site of pJT61 (Vieillard et al, 2016) using Gibson Assembly Master Mix (New England Biolabs Inc.).

Alms1a-Tomato was integrated in the 89E11 VK00027 landing site on the third chromosome and Alms1b-GFP on the 53B2

VK00018 landing site on the second chromosome by PhiC31 integrase (BestGene Inc.).

### Generation of *alms1a*$^{del1}$, *alms1b*$^{del2}$ and *alms1*$^{del3}$ by CRISPR/Cas9

*alms1a*$^{del1}$ allele was generated by CRISPR/Cas9-induced deletion (NHJE). Three couples of gRNAs (gRNA1Alms1a+gRNA2Alms1a, gRNA3Alms1a+gRNA4Alms1a and gRNA5Alms1a+gRNA6Alsm1a) were respectively cloned in pBFv-U6.2B vector (Kondo and Ueda, 2013) and injected together in CG12179[NP]/(FM7h);;vasa-Cas9 embryos. Flies were crossed to Hira$^{[-]}$/FM7h females or FM7h/Y males and offspring were selected for white-eyed flies. Deletion in the *alms1a* locus was further characterised by PCR with primers F-Alms1aKO/R-Alms1aKO and confirmed by sequencing.

*alms1b*$^{del2}$ allele was generated by CRISPR/Cas9-induced homologous direct repair (Gratz et al, 2015). The 1.5 kb 5' homology arm and 1.4 kb 3' homology arm were amplified by PCR (F-5'armAlms1b/R-5'armAlms1b and F-3'armAlms1b/R-3'armAlms1b) on genomic DNA from vasa-Cas9 flies and cloned, respectively, into the NheI-KpnI sites and BglII-AvrII sites of pJT38 plasmid (pRK2 plasmid (Huang et al, 2008) containing an attB cassette). Two gRNAs (gRNA1Alms1b and gRNA2Alms1b) were cloned in the pBFv-U6.2B vector (Kondo and Ueda, 2013). The two constructs were injected in vasa-Cas9 embryos. Flies were crossed to Hira$^{[-]}$/FM7h females or FM7h/Y males and the offspring were screened for red-eyed flies. Homologous recombination was checked by PCR (F-5'Alms1bKO/R-5'Alms1bKO and F1-3'Alms1bKO/R-3'Alsm1bKO).

*alms1*$^{del3}$ allele was generated from *alms1a*$^{del1}$ flies by CRISPR/Cas9-induced homologous direct repair on the *alms1b* locus. Template repair vector was constructed as above, with the exception of the 5' arm, which was obtained by PCR with primers F-5'armAlms1a$^{del1}$/R-5'armAlms1b on genomic DNA from *alms1a*$^{del1}$ flies. This new template repair was injected together with the previous pBFv-U6.EB vector expressing gRNA1Alms1b and gRNA2Alms1b in *alms1a*$^{del1}$ embryo. Flies were crossed to Hira$^{[-]}$/FM7h females or FM7h/Y males and the offspring were screened for red-eyed flies. Homologous recombination was checked by PCR (F2-5'Alms1bKO/R-5'Alms1bKO and F2-3'Alms1bKO/R-3'Alms1bKO).

*alms1* deletions were validated by PCR on genomic DNA extracted from single hemizygous male (Appendix Fig. S1B) and on cDNA obtained by RT-PCR on RNA extracted from testes of hemizygous males (Appendix Fig. S1C). Primer couples specific of the control gene *rbp49* (F-RBP49 and R-RBP49, 438 bp), *alms1a* (F2-Alms1a and R2-Alms1a, 345 bp) or *alms1b* (F3-Alms1b and R-3-Alms1b, 354 bp) were used.

**Table 1. *Drosophila melanogaster* stocks.**

| | | |
|---|---|---|
| Alm1a-Tomato in rescue condition | *alms1a*<sup>del1</sup>/FM7h; ; Alms1a-Tomato/TM3, Ser | This study |
| *alms1a*<sup>del1</sup> | *alms1a*<sup>del1</sup>/FM7h | This study |
| Alms1a-Tomato | ; ; Alms1a-Tomato/TM3, Ser | This study |
| Alms1a<sup>R</sup>-Tomato | ; ; Alms1a<sup>R</sup>-Tomato/TM3, Ser | This study |
| Alms1b-GFP in rescue condition | *alms1b*<sup>del2</sup>/FM7; Alms1b-GFP/CyO | This study |
| *alms1b*<sup>del2</sup> | *alms1b*<sup>del2</sup>/FM7 | This study |
| Alms1b-GFP | ; Alms1b-GFP/CyO | This study |
| *alms1*<sup>del3</sup> | *alms1a*<sup>del1</sup>, *alms1b*<sup>del2</sup>/FM7h | This study |
| *lacZ*<sup>RNAi</sup> | w<sup>1118</sup>; P{GD936}v51446 | VDRC 51446 |
| *alms1*<sup>RNAi</sup> | ; UAS-P{TRiP.HMJ30289}/CyO | BL 63721 |
| *bam*-Gal4 | ; If/CyO; *bam*-Gal4, UAS-Dcr2/TM6B | |
| *bam*-Gal4 | *bam*-Gal4, UAS-Dcr2/FM7h | |
| *nos*-Gal4 | w<sup>1118</sup> ; ; P{w[+mC]=GAL4::VP16-*nanos*.UTR}CG6325[MVD1] | BL 4937 |
| *wor*-Gal4, Miranda-GFP | w˙; P{w[+mC]=wor.GAL4.A}2, P{w[+mC]=UAS-mira.GFP}1.2/CyO ; Dr/TM6B | Modified from BL 56555 |
| Ana1-GFP | w˙; Bl/CyO ; P{ana1-GFP.B}/TM6B | |
| Bld10-GFP | ; UASp-endo-Bld10-GFP [142.1]/CyO | From T. Megraw (Mottier-Pavie and Megraw, 2009) |
| Plk4-eGFP | w˙; If/CyO; meGFP-Plk4/TM6B | From M. Bettencourt-Dias (Nabais et al, 2021) |
| Ana2-eNG | w˙; eAna2-mNG 3.72/SM5 | From J. Raff (Steinacker et al, 2022) |
| Sas-6-GFP | w˙; pUbq-GFP-Sas-6/CyO | From R. Basto |
| Plk4<sup>OE</sup> | ; UASp-Plk4/CyO | From M. Bettencourt-Dias |
| Plk4<sup>ND-GFP</sup> | ; UAS-ND-Plk4-GFP/CyO | From M. Bettencourt-Dias |
| Ana2<sup>OE</sup> | ; ; UASp-Ana2-GFP/TM6B | From J. Raff |
| Ana2<sup>OE</sup> | ; UASp-Ana2-GFP/CyO | From J. Raff |
| Alms1a<sup>OE</sup> | ; ; UASp-v5-Alms1a/TM3, Sb | This study |

## Fertility test

Single males emerged from the day were crossed to 3 3-day-old *w*<sup>1118</sup> females. Flies were left together at 25 °C for 5 days before being discarded. Total progeny was scored (Appendix Fig. S1D).

## Classical immunofluorescence preparations

### Testes

Testes from young adults were dissected in PBS and fixed 20 min in PFA 4%. For each fly, a single testis was kept to avoid analysing the same individual twice. After 20 min of permeabilisation in PBS-Triton X-100 0.1% (PBST), testes were blocked 1 h in PBST/BSA 2% (PBST-BSA) and then incubated with primary antibodies (in PBST-BSA, see Table 3) overnight at 4 °C. After four washes, testes were incubated 2 h with appropriate secondary antibodies and Hoechst. After four washes, testes were mounted in Vectashield Antifade Mounting Medium. If immunofluorescence was not required, testes were incubated 2 h in Hoechst in PBST.

### Brains

Larva collection was performed as follows: 20% sucrose solution was added in tubes containing larvae in food. The supernatant containing larvae was collected, rinsed in PBS and larvae were placed on a Petri dish with agar. Third-instar larvae were selected for dissection. Brains were dissected in PBS during sessions not exceeding 20 min and then fixed 25 min in PFA. After two baths of 10 min each in PBST 0.3%, brains were blocked 2 h in PBST-BSA 2% and then incubated with anti α-tubulin overnight at 4 °C. After four washes, brains were incubated with the appropriate secondary antibody and Hoechst overnight at 4 °C. Brains were last washed (4×) and mounted in Vectashield Antifade Mounting Medium.

### Embryos

In all, 30 min–2 h 30 synchronised embryos were collected on a petri dish with agar. Embryos were dechorionated for 5 min in bleach, devitellinised by vigorous shaking in heptane-methanol solution (1:1 ratio) and fixed in methanol. Embryos were blocked in two 30 min baths of PBST 0.1%-BSA 3% and incubated in anti β-tubulin overnight at 4 °C. After four washes, embryos were incubated with the appropriate secondary antibody and Hoechst for two hours at RT. Embryos were last washed 4x and mounted in Vectashield Antifade Mounting Medium.

### PCM proteins fluorescence quantification

*w*<sup>1118</sup>, *alms1a*<sup>del1</sup> and *alms1a*<sup>del1</sup>;;Alms1a-Tom were prepared accordingly to the above protocols and incubated, for classical immunofluorescence with anti-Asterless or anti-Plp antibodies and for

**Table 2.** Genotypes of *Drosophila melanogaster* flies analysed.

| Figure | Flies as in text | Genotype |
|---|---|---|
| Figure 1B–D Figure 3A,B | Alms1a-Tom | *alms1a^del1^/Y; ; *Alms1a-Tom |
| Figure 1A | Alms1b-GFP | *alms1b^del2^/Y;* Alms1b-GFP |
| Figure 3C,D | Control expressing Bld10-GFP | *w^1118^*; UASp-endo-Bld10-GFP [142.1]/+ |
| Figure 3C,D | *alms1a ^del1^* expressing Bld10-GFP | *alms1a^del1^/Y;* UASp-endo-Bld10-GFP [142.1]/+ |
| Figure 4A | *alms1^del3^; nos*-Gal4>*alms1*^RNAi^ | *alms1^del3^* /Y ; UAS-P{TRiP.HMJ30289}/+ P{w[+mC]=GAL4::VP16-*nanos*.UTR}CG6325[MVD1]/+ |
| Figure 4A | *nos*-Gal4>*alms1*^RNAi^ | *w/Y* ; UAS-P{TRiP.HMJ30289}/+ P{w[+mC]=GAL4::VP16-*nanos*.UTR}CG6325[MVD1]/+ |
| Figure 4A | *alms1^del3^; nos*-Gal4 | *alms1^del3^* /Y ; ; P{w[+mC]=GAL4::VP16-*nanos*.UTR}CG6325[MVD1]/+ |
| Figure 4B | males *bam*-Gal4>*lacZ*^RNAi^ expressing Ana1-GFP | *w/Y* ; P{GD936}v51446/+ P{ana1-GFP.B}/*bam*-Gal4, UAS-Dcr2 |
| Figure 4B | males *bam*-Gal4>*alms1*^RNAi^ expressing Ana1-GFP | *w/Y*; UAS-P{TRiP.HMJ30289}/+ P{ana1-GFP.B}/*bam*-Gal4, UAS-Dcr2 |
| Figure 4C,D | *bam*-Gal4>*lacZ*^RNAi^ | *w/Y* ; P{GD936}v51446/+ ;*bam*-Gal4, UAS-Dcr2/+ |
| Figure 4C,D | *bam*-Gal4>*alms1*^RNAi^ | *w/Y* ; UAS-P{TRiP.HMJ30289}/+; *bam*-Gal4, UAS-Dcr2/+ |
| Figure 4E | *wor*-Gal4>*lacZ*^RNAi^ | *w`/Y*; P{w[+mC]=wor.GAL4.A}2, P{w[+mC]=UAS-mira.GFP}1.2/P{GD936}v51446 |
| Figure 4E | *wor*-Gal4>*alms1*^RNAi^ | *w`/Y*; P{w[+mC]=wor.GAL4.A}2, P{w[+mC]=UAS-mira.GFP}1.2/ UAS-P{TRiP.HMJ30289} |
| Figure 4F | *nos*-Gal4>*lacZ*^RNAi^ | *w/Y*; P{GD936}v51446/+ P{w[+mC]=GAL4::VP16-*nanos*.UTR}CG6325[MVD1]/+ |
| Figure 4F | *nos*-Gal4>*alms1*^RNAi^ | *w/Y*; UAS-P{TRiP.HMJ30289}/+ P{w[+mC]=GAL4::VP16-*nanos*.UTR}CG6325[MVD1]/+ |
| Figure 5B | Sas-6-GFP in *bam*-Gal4>*lacZ*^RNAi^ | *w/Y* ; P{GD936}v51446/ pUbq-GFP-Sas-6;*bam*-Gal4, UAS-Dcr2/+ |
| Figure 5C | Sas-6-GFP in *bam*-Gal4>*alms1*^RNAi^ | *w/Y* ; UAS-P{TRiP.HMJ30289}/ pUbq-GFP-Sas-6; *bam*-Gal4, UAS-Dcr2/+ |
| Figure 5D | Ana2-mNeonGreen knock-in *bam*-Gal4>*lacZ*^RNAi^ | *w/Y* ; P{GD936}v51446/ eAna2-mNG 3.72; *bam*-Gal4, UAS-Dcr2/+ |
| Figure 5D | Ana2-mNeonGreen knock-in *bam*-Gal4>*alms1*^RNAi^ | *w/Y* ; UAS-P{TRiP.HMJ30289}/ eAna2-mNG 3.72; *bam*-Gal4, UAS-Dcr2/+ |
| Figure 5E | Plk4-meGFP knock-in in *bam*-Gal4>*lacZ*^RNAi^ | *w/Y* ; P{GD936}v51446/ +; meGFP-Plk4/ *bam*-Gal4, UAS-Dcr2 |
| Figure 5E | Plk4-meGFP knock-in *bam*-Gal4>*alms1*^RNAi^ | *w/Y* ; UAS-P{TRiP.HMJ30289}/ +; meGFP-Plk4/ *bam*-Gal4, UAS-Dcr2 |
| Figure 6A | *bam*-Gal4 > {*lacZ*^RNAi^; Plk4 ^OE^} | *w/Y* ; P{GD936}v51446/ UASp-Plk4; *bam*-Gal4, UAS-Dcr2/+ |
| Figure 6A | *bam*-Gal4 > {*alms1*^RNAi^; Plk4 ^OE^} | *w/Y* ; UAS-P{TRiP.HMJ30289}/ UASp-Plk4; *bam*-Gal4, UAS-Dcr2/+ |
| Figure 6B,C | *bam*-Gal4 > {Plk4 ^OE^-GFP} | *w/Y* ; UASp-GFP-Plk4/(CyO or If); *bam*-Gal4, UAS-Dcr2/+ |
| Figure 6B,C | *bam*-Gal4 > {Alms1a ^OE^-V5} | *w/Y* ; +/(CyO or If); *bam*-Gal4, UAS-Dcr2/UASp-v5-Alms1a |
| Figure 6B,C | *bam*-Gal4 > {Plk4 ^OE^-GFP; Alms1a ^OE^-V5}} | *w/Y* ; UASp-GFP-Plk4/(CyO or If); *bam*-Gal4, UAS-Dcr2/ UASp-v5-Alms1a |
| Figure 6D,F | *bam*-Gal4 > {*lacZ*^RNAi^; ND-Plk4-GFP} | *w/Y* ; P{GD936}v51446/ UAS-ND-Plk4-GFP; *bam*-Gal4, UAS-Dcr2/+ |
| Figure 6D,F | *bam*-Gal4 > {*alms1*^RNAi^, ND-Plk4-GFP} | *w/Y* ; UAS-P{TRiP.HMJ30289}/ UAS-ND-Plk4-GFP; *bam*-Gal4, UAS-Dcr2/+ |
| Figure 6G | *alms1^del3^; nos*-Gal4>ND-Plk4-GFP | *alms1^del3^* /Y ; UAS-ND-Plk4-GFP/+; *nos*-Gal4/+ |
| Figure 6G | *nos*-Gal4> ND-Plk4-GFP | *w/Y* ; UAS-ND-Plk4-GFP/+; *nos*-Gal4,/+ |
| Figure 7A | *bam*-Gal4>Ana2-GFP | *w/Y* ; UASp-Ana2-GFP/(CyO or If ;*bam*-Gal4, UAS-Dcr2/ + |
| Figure 7A | *bam*-Gal4>Alms1a-V5 | *w/Y* ; +/(CyO or If); *bam*-Gal4, UAS-Dcr2/UASp-v5-Alms1a |
| Figure 7A | *bam*-Gal4 > {Ana2-GFP; Alms1a ^OE^-V5}} | *w/Y* ; UASp-Ana2-GFP /(CyO or If); *bam*-Gal4, UAS-Dcr2/ UASp-v5-Alms1a |
| Figure 7B,C | *bam*-Gal4 > {*alms1*^RNAi^} | *w/Y* ; UAS-P{TRiP.HMJ30289}/ (CyO or If); *bam*-Gal4, UAS-Dcr2/+ |
| Figure 7B,C | *bam*-Gal4 > {*alms1*^RNAi^, ND-Plk4-GFP} | *w/Y* ; UAS-P{TRiP.HMJ30289}/ UAS-ND-Plk4-GFP; *bam*-Gal4, UAS-Dcr2/+ |
| Figure 7B,C | *bam*-Gal4 > {*alms1*^RNAi^, Ana2-GFP} | *w/Y* ; UAS-P{TRiP.HMJ30289}/(CyO or If); *bam*-Gal4, UAS-Dcr2/ UASp-Ana2-GFP |
| Figure 7B,C | *bam*-Gal4 > {*alms1*^RNAi^, ND-Plk4-GFP, Ana2-GFP} | *w/Y* ; UAS-P{TRiP.HMJ30289}/ UAS-ND-Plk4-GFP; *bam*-Gal4, UAS-Dcr2/ UASp-Ana2-GFP |

**Table 2.** (continued)

| Figure | Flies as in text | Genotype |
|---|---|---|
| Figure EV1A | Alms1b-GFP | *alms1b^del2*/Y; Alms1b-GFP |
| Figure EV1B | control | *w^1118*/Y |
| Figure EV1C | Alms1b-GFP | *alms1b^del2*/Y; Alms1b-GFP / P{w[+mC]=wor.GAL4.A}2, P{w[+mC]=UAS-mira.cherry}2 |
| Figure EV1D | Alms1b-GFP | *alms1b^del2*/Y; Alms1b-GFP |
| Figure EV2A,B | Control | *w^1118*/Y |
| Figure EV2A,B | *alms1a^del1* | *alms1a^del1*/Y |
| Figure EV2B | *alms1a^del1*::Alms1a-Tom | *alms1a^del1*/Y; ; Alms1a-Tom |
| Figure EV3A | control | *alms1b^del2*/Y; Alms1b-GFP |
| Figure EV3A | *alms1a^del1* | *alms1a^del1*/Y |
| Figure EV3A | *alms1b^del2* | *alms1b^del2*/Y |
| Figure EV3B,C | control | *w^1118*/Y |
| Figure EV3B,C | *alms1^del3* | *alms1^del3*/Y |
| Figure EV3B,C | *alms1^del3*;;,Alms1a-Tom (rescue) | *alms1^del3*/Y ; ; Alms1a-Tom |
| Figure EV3D | nos>*alms1*^RNAi, Alms1a^R-Tom | *w*/Y ; UAS-P{TRiP.HMJ30289}/(CyO or If); nos-Gal4 /Alms1a^R-Tom |
| Figure EV4A,B | bam-Gal4>*alms1*^RNAi | *w*/Y ; UAS-P{TRiP.HMJ30289}/ (CyO or If); bam-Gal4, UAS-Dcr2/+ |
| Figure EV4A,B | bam-Gal4>*lacZ*^RNAi | *w*/Y ; P{GD936}v51446/ (CyO or If); bam-Gal4, UAS-Dcr2/+ |
| Figure EV5A | Ana2-mNeonGreen knock-in bam-Gal4>*lacZ*^RNAi | *w*/Y ; P{GD936}v51446/ eAna2-mNG 3.72; bam-Gal4, UAS-Dcr2/+ |
| Figure EV5B | Ana2-mNeonGreen knock-in bam-Gal4>*alms1*^RNAi | *w*/Y ; UAS-P{TRiP.HMJ30289}/ eAna2-mNG 3.72; bam-Gal4, UAS-Dcr2/+ |
| Figure EV6A | bam-Gal4 > {*lacZ*^RNAi; ND-Plk4-GFP} | *w*/Y ; P{GD936}v51446/ UAS-ND-Plk4-GFP; bam-Gal4, UAS-Dcr2/+ |
| Figure EV6A | bam-Gal4 > {*alms1*^RNAi, ND-Plk4-GFP} | *w*/Y ; UAS-P{TRiP.HMJ30289}/ UAS-ND-Plk4-GFP; bam-Gal4, UAS-Dcr2/+ |
| Figure EV6B | bam-Gal4 > {*lacZ*^RNAi} | *w*/Y ; P{GD936}v51446/ (CyO or If); bam-Gal4, UAS-Dcr2/+ |
| Figure EV6B | bam-Gal4 > {*lacZ*^RNAi, ND-Plk4-GFP} | *w*/Y ; P{GD936}v51446/ UAS-ND-Plk4-GFP; bam-Gal4, UAS-Dcr2/+ |
| Figure EV6B | bam-Gal4 > {*lacZ*^RNAi, Ana2-GFP} | *w*/Y ; P{GD936}v51446/(CyO or If); bam-Gal4, UAS-Dcr2/ UASp-Ana2-GFP |
| Figure EV6B | bam-Gal4 > {*lacZ*^RNAi, ND-Plk4-GFP, Ana2-GFP} | *w*/Y ; P{GD936}v51446/ UAS-ND-Plk4-GFP; bam-Gal4, UAS-Dcr2/ UASp-Ana2-GFP |

These flies result from crosses with stocks listed in Table 1.

U-ExM with anti-acetylated α-tubulin and anti-Asterless antibodies. Samples of the different genotypes were processed identically and simultaneously. Adjustments of acquisition parameters were performed on one testis isolated from each genotype. Acquisitions were then made with the same exposure and laser parameters for all testes. Fluorescence quantification was performed after sample anonymisation by a third party. For each testis, Asl or Plp and Bld10-GFP (IF) or Asl and acetylated α-tubulin (U-ExM) fluorescence was quantified for 10 centrioles per stage (spermatogonia, early spermatocytes and late spermatocytes) using a threshold to delimit the area to be quantified and from which background noise was subtracted (macros available upon request).

*Plk4 fluorescence quantification*

bam-Gal4>lacZ^RNAi and bam-Gal4>alms1^RNAi testes expressing Plk4-meGFP were incubated with anti-Asterless antibody. Samples of the different genotypes were processed identically and simultaneously. Adjustments of acquisition parameters were performed on one testis isolated from each genotype. Acquisitions were then made with the same exposure and laser parameters for all testes. For each testis, Plk4 fluorescence was quantified for 10 centrioles per stage (spermatogonia and early spermatocytes) using a square

surrounding the centriole to delimit the area to be quantified and from which background noise was subtracted (macros available upon request). (*lacZ*^RNAi: $n = 4$ testes; *alms1*^RNAi: $n = 5$ testes).

## U-ExM preparations and immunolabelling (adapted from (Gambarotto et al, 2019))

### Testes

Testes from young adults were dissected in PBS and incubated PBS/1.4% Formaldehyde—0.3% Acrylamide (PBS-FA-AA) overnight at 18 °C. Prior to the gelation, testes were incubated 2 h in the monomer solution (PBS/19% Sodium Acrylate—10% Acrylamide—0.1% Bis-Acrylamide—0.5% TEMED – Hoechst 1/500) to allow reagents to penetrate the inner layers of the tissue. Testes were then deposited in packs of 3–4 on 12 mm coated coverslips (0.2 mg/mL Poly-D-Lysine during 2 h at 37 °C) and all the monomer solution was absorbed with filter paper. Coverslips were deposited on a silicone cushion made in 50-mL Falcon tube and spun down using a swing rotor for 5 min at 5000 rpm at 10 °C. In total, 38 μL of cold monomer solution with 0.5% APS was deposited on coverslips which were immediately put upside down on a parafilm tensed on a microscope slide laid on an ice-cold block. Gelation was started on

**Table 3.  Antibodies used.**

|  |  |  | Classical preparation | U-ExM |
|---|---|---|---|---|
| **Primary antibodies** | | | | |
| Anti-acetylated α-Tubulin | Mouse, IgG2b | clone 6-11B-1, T6793, Sigma-Aldrich, lot 017M4806V | – | 1:500 |
| Anti-RFP | Mouse, IgG2c | 6G6, ChromoTek, 6G6, lot 107272-07-02 | – | 1:100 |
| Anti-GFP | Rabbit | TP401, Chemokine, lot 081211 | – | 1:100 |
| Anti-GFP | Rabbit | 632459, Living Colors, lot1105001 | 1:500 | – |
| Anti-mNeonGreen | Rabbit | Cell Signaling Technology | – | 1:100 |
| Anti-Asterless | Rat/guinea pig | from G. Rogers/ this study | 1:75,000 | 1:5000 |
| Anti-Plp | Rabbit | from G. Rogers | 1:2000 | 1:1000 |
| Anti-β-tubulin | Mouse, IgG1 | E7, Developmental Studies Hybridoma Bank | 1:100 | 1:500 |
| Anti-α-tubulin | Mouse, IgG2a | AA345, ABCD antibodies | – | 1:250 (cells) |
| Anti-β-tubulin | Mouse, IgG2a | AA344, ABCD antibodies | – | 1:250 (cells) |
| Anti-ALMS1 | Rabbit | 27231-1-AP, Proteintech | – | 1:500 (cells) |
| Anti-Alms1a | Guinea pig | From Y. Yamashita (Chen and Yamashita, 2020) | | |
| Anti-Alms1b | Guinea pig | From Y. Yamashita (Chen and Yamashita, 2020) | | |
| **Secondary antibodies** | | | | |
| **Anti-mouse** | | | | |
| Alexa fluor 488 goat anti-mouse IgG (H + L) | | A11001, lot 1810918, Invitrogen | 1:1000 | 1:1000 |
| Alexa fluor 488 F(ab')₂ donkey anti-mouse IgG (H + L) | | 715-546-151, lot 152325, Jackson Lab | 1:1000 | 1:1000 |
| Alexa fluor 594 goat anti-mouse IgG (H + L) | | A11005, lot 1796406, Invitrogen | 1:1000 | 1:10001:625 (cells) |
| Alexa fluor 594 goat anti-mouse IgG2b | | A21135, lot 898249, Invitrogen | – | 1:1000 |
| **Anti_rabbit** | | | | |
| Alexa fluor 488 goat anti-rabbit IgG (H + L) | | A11008, lot 1797971, Invitrogen | – | 1:1000 1:625 (cells) |
| Alexa fluor 488 F(ab')₂ goat anti-rabbit IgG (H + L) | | 111-546-144, Lot 153510, Jackson Lab | – | 1:1000 |
| Alexa fluor 594 F(ab')₂ goat anti-rabbit IgG (H + L) | | 111-586-144, Lot 148526, Jackson Lab | 1:1000 | 1:1000 |
| Alexa fluor 647 goat anti-rabbit | | A21244, lot 1654324, Invitrogen | 1:1000 | 1:1000 1:625 (cell) |
| **Anti-rat** | | | | |
| Alexa fluor 488 goat anti-rat | | A11006, lot 940882, Invitrogen | 1:1000 | 1:1000 |
| Alexa fluor 647 F(ab')₂ donkey anti-rat IgG (H + L) | | 712-606-153, Lot 150868, Jackson Lab | 1:1000 | 1:1000 |

Unless specified, antibodies were used on *Drosophila* tissues.

cold for 5 min before transfer for 1 h at 37 °C. Gels were cut around groups of testes on the coverslip using a biopsy punch (4 mm diameter). Punches were incubated in denaturation buffer (SDS 200 mM—NaCl 200 mM—Tris pH 9 50 mM) for 1.5 h at 95 °C. After three washes in water, gels were preserved in PBS at 4 °C.

Gels were blocked 2 h in PBS/Tween 0.05%—BSA 1%—Sodium azide 0.02% (PBSTw-BSA) and then incubated with primary antibodies (in PBSTw-BSA) in a humid chamber overnight at 4 °C. After four washes, gels were incubated overnight with appropriate secondary antibodies and Hoechst and washed again. Gels were expanded for 2 h in two baths of water. Gels were mounted on 15 mm coated coverslips (0.2 mg/mL Poly-D-Lysine during 2 h at 37 °C).

### Brains
Larvae were collected and selected as described above. Brains were dissected in PBS and immediately incubated in PBS-FA-AA. PBS-FA-AA solution was renewed before incubation overnight at 18 °C. Brains

were then dissected again on 12-mm coated coverslips to remove the two optic lobes. The rest of the protocol is identical as described above.

### Embryos
Embryos were collected, dechorionated and devitellinised as described above. The rest of the protocol is identical to the testes one.

### Human cells (Guennec et al, 2020)
hTERT-RPE-1 cells (gift from A-M. Tassin; ATCC-CRL-4000) were grown on non-coated 12 mm diameter coverslips in DMEM-F12-Glutamax 10% SVF and penicillin/streptomycin 1% at 37 °C in 5% $CO_2$. RPE-1 were seeded at 35,000 cell/cm² for 24 h. Coverslips were rinsed in PBS 1× and incubated in 1 mL of a AA2% -FA1.4% in PBS, in a 12-well plate for 3 h at 37 °C. The coverslip was placed, cells facing down, on a drop of 35 μL monomer solution with APS deposited on a parafilm in a humid chamber placed on ice for 5 min before incubation at 37 °C for 30–60 min.

Gels were cut in 4–5 circles of using a biopsy punch (4 mm diameter) and incubated in denaturation buffer for 15 min at RT under agitation for the gel to detach from the coverslips. Gel pieces were transferred in 1.5-mL tube filled with denaturation buffer at 95 °C for 1.30 h. Gels were then washed two times 15 min in $H_2O$ and transferred in PBS 1×. Primary antibodies diluted in PBS-BSA 2% were incubated from 5 h to O/N at RT with agitation, washed $3 \times 10$ min at RT in PBS-Tween (0.1%) under agitation, incubated with secondary antibodies in PBS-BSA 3 to 5 h at RT and finally washed $3 \times 10$ min at RT in PBS-Tween (0.1%) under agitation. Expansion was performed as above before imaging. For centriole labelling on U-ExM cells, mouse IgG2a anti α-tubulin and anti β-tubulin were mixed.

### Testis centriole identification in U-ExM

Mother and daughter centrioles are identified based on their relative organisation: during centriole duplication, the daughter centriole is templated from the wall of the mother centriole. In U-ExM acquisition, when both centrioles retain their close, orthogonal organisation, the centriole which lumen faces the wall of the other centriole is identified as the daughter.

### U-ExM controls

Expansion factor is calculated in each experiment by measuring the size of the gel disk obtained by standardised biopsy punch (4 mm diameter) after expansion. Centriole expansion isotropy and factor was validated by measuring centriole diameter (electron microscopy reference: 200 nm ± 12 nm). Values presented in graphs and scale bars always correspond to "real" values after the application of the expansion factor.

### Image acquisition

All images (classical and U-ExM) were obtained using an IX 83 inverted microscope from Olympus, equipped with a Yokagawa CSU-X1 Spinning Disk Unit, Borealis technology for homogeneous illumination and Ixon3 888 EM-CCD camera from Andor. The oil immersion Plan Apochromat 60x/1.42 NA objective from Olympus was used for all acquisitions. All images were processed and analysed with FiJi (Schindelin et al, 2012).

### Human centriole representations and quantifications

For each image of human expanded centriole, a small z-projection (stack of 2–4 images maximum) was applied. Images were duplicated and rotated to orientate each centriole (pro or mature) in the proximal to distal orientation. A crop of $50 \times 75$ pixels was made around each centriole and images were sorted by tubulin length For bottom view-oriented centrioles, only single plane images are presented as a crop of $50 \times 50$ pixels.

*Longitudinal and radial measurements from side viewed centrioles*: Measurements of the length, diameter and relative position of ALMS1 were done on dual staining images where tubulin is always used as a proxy for the evaluation of the centriole length. From resized images where the pixel size was artificially decreased by 3 to improve the measurement precision (plugin "CropAndResize", https://github.com/CentrioleLab), the fluorescent signal distribution of tubulin and ALMS1 were measured using the Fiji line scan (with a width covering the centriole for longitudinal measurements) and the plot profile tool. Using the

plugin "PickCentrioleDim" (https://github.com/CentrioleLab) to facilitate the picking of the start and the end of the fluorescent signal (defined as 50% of the peak value at both extremities of the centriole), the user automatically generates an entry table with the coordinates of fluorescent signal extremities in each measured channel. For raw length and diameter measurements, the distance between the fluorescent signal extremities was calculated after application of the gel expansion factor and plotted using GraphPad Prism7. For the relative protein position, the tubulin was defined as the reference protein and its starting coordinate was shifted and set to 0. The same shift was applied to ALMS1. The entry table was read by a second plugin ("CentrioleGraph", https://github.com/CentrioleLab) which generates a plot illustrating the relative average position of ALMS1 to the tubulin.

*Appearance of ALMS1 during procentriole growth*: The appearance ALMS1 was evaluated by measuring the length of the tubulin fluorescent signal in growing procentrioles. Measurements were sorted depending on whether the fluorescent signal of ALMS1 was present or not and plotted as two separate sets of data. The average between the highest centriole length without fluorescent signal and the lowest centriole length with fluorescent signal was considered as the appearance point of ALMS1.

## Transmission electron microscopy

Testes from pupae and 3–4 days old adults were dissected in phosphate-buffered saline (PBS), and fixed in 2.5% glutaraldehyde in PBS overnight at 4 °C. After rinsing for 30 min in PBS, the samples were post-fixed in 1% osmium tetroxide in PBS for 1 h. The samples were dehydrated in a graded series of ethanol and then infiltrated with a mixture of Epon–Araldite resin and polymerised at 60 °C for 48 h. Ultrathin sections (50–70 nm thick) were cut with a Reichert ultramicrotome equipped with a diamond knife. The sections were collected with formvar-coated copper slot grids and stained with 2% aqueous uranyl acetate for 20 min in the dark and then with lead citrate for 2 min. TEM preparations were observed with a Tecnai G2 Spirit EM (FEI Eindhoven, The Netherlands) equipped with a Morada CCD camera (Olympus, Tokyo, Japan).

## Statistical analysis

Either non-parametric, two-sided, unpaired Wilcoxson or Fisher exact tests were performed using R or GraphPad Software. In all figures, the box plot shows the interquartile range (IQR), with the median (50th percentile) indicated by the horizontal line inside the box and the whiskers extending to the 10th and 90th percentiles.

## Data availability

The imaging source data for this study are available in the following database: *BioStudies*, S-BIAD1481 (https://doi.org/10.6019/S-BIAD1481).

The source data of this paper are collected in the following database record: biostudies:S-SCDT-10_1038-S44318-025-00382-8.

# Peer review information

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

## Acknowledgements

This work was supported by an AFM Grant MyoNeurAlp 1 and the Ligue Régionale contre le cancer (CD26 and CD69, 5FI15284XUUK). MB was supported by a doctoral fellowship from the AFM. J-AL was supported by the FRM (DEQ20131029168). Work was also supported by the ANR DIVERCIL (ANR-17-CE13-0023). The authors acknowledge the contribution of SFR Santé Lyon-Est (UAR3453 CNRS, US7 Inserm, UCBL) facility: CIQLE (a LyMIC member). We thank J Raff, M Bettencourt-Dias, R Basto, T Megraw, G Rogers and Y Yamashita for generously sharing fly stocks and reagents. Fly stocks obtained from the Bloomington Drosophila Stock Center (NIH P40OD018537) and the Vienna Drosophila Resource Center (VDRC) were used in this study. We thank J Perrichet for technical assistance and fly husbandry and H Volkmar, S Gomez, A Fagot and R Hocini for help with cell expansion microscopy. The authors thank SC Jana for insightful discussions.

## Author contributions

**Marine Brunet**: Data curation; Formal analysis; Investigation; Methodology; Writing—original draft; Writing—review and editing. **Joëlle Thomas**: Resources. **Jean-André Lapart**: Investigation. **Léo Krüttli**: Investigation. **Marine H Laporte**: Investigation; Methodology. **Maria Giovanna Riparbelli**: Investigation. **Giuliano Callaini**: Investigation. **Bénédicte Durand**: Conceptualisation; Resources; Supervision; Funding acquisition; Methodology; Writing—original draft; Writing—review and editing. **Véronique Morel**: Conceptualisation; Resources; Data curation; Formal analysis; Supervision; Funding acquisition; Investigation; Methodology; Writing—original draft; Writing—review and editing.

Source data underlying figure panels in this paper may have individual authorship assigned. Where available, figure panel/source data authorship is listed in the following database record: biostudies:S-SCDT-10_1038-S44318-025-00382-8.

## Disclosure and competing interests statement

The authors declare no competing interests.

# Expanded View Figures

**Figure EV1.  Alms1a and Alms1b dynamics of localisation.**

(**A**) Unexpended images of Alms1b-GFP (magenta) localisation prior to spermatids formation, during the second meiotic division. The two newly separated centrioles display Alms1b at their proximal end. Ana1-Tomato (green) as a centriolar marker. (**B**) Dynamics of localisation of endogenous Alms1a (top row, red) and endogenous Alms1b (bottom row, magenta) from spermatogonia to round spermatid stages. As observed with Alms1a-Tom transgene, endogenous Alms1a is detected from the spermatogonia until after the round spermatid stages at the proximal side of centrioles (labelled in green with Asl). As for Alms1b-GFP, endogenous Alms1b is not detected in spermatogonia. It is first observed in round spermatids in contrast to Alms1b-GFP which is detected from the late SC stage. This difference in timing could be due to technical difficulties to observe minute amounts of proteins with the antibody compared to the GFP-tagged protein. Endogenous proteins thus show dynamics of localisation at centrioles similar to that described with Alms1a-Tom and Alms1b-GFP. (**C**) Alms1b-GFP (magenta) localisation in neuroblasts (NB, dotted white outline) and ganglion mother cell (GMC, plain white outline). We did not detect Alms1b-GFP in NB, GMC or any other larval brain cell type. Miranda-mCherry (yellow) used as a landmark for GMC and NB. Nuclei in grey. (**D**) Alms1b-GFP (magenta) localisation in syncytial embryo (top) and cellularised embryo (bottom). No Alms1b-GFP is detected during the syncytial stage. Alms1b-GFP starts to localise at spindle pole during the last synchronous mitosis (cycle 14) in cellularised embryos Tubulin in green, DNA in cyan. Scale bars: (**A**, **C**, **D**) 5 μm, (**A**, inset, **B**) 1 μm.

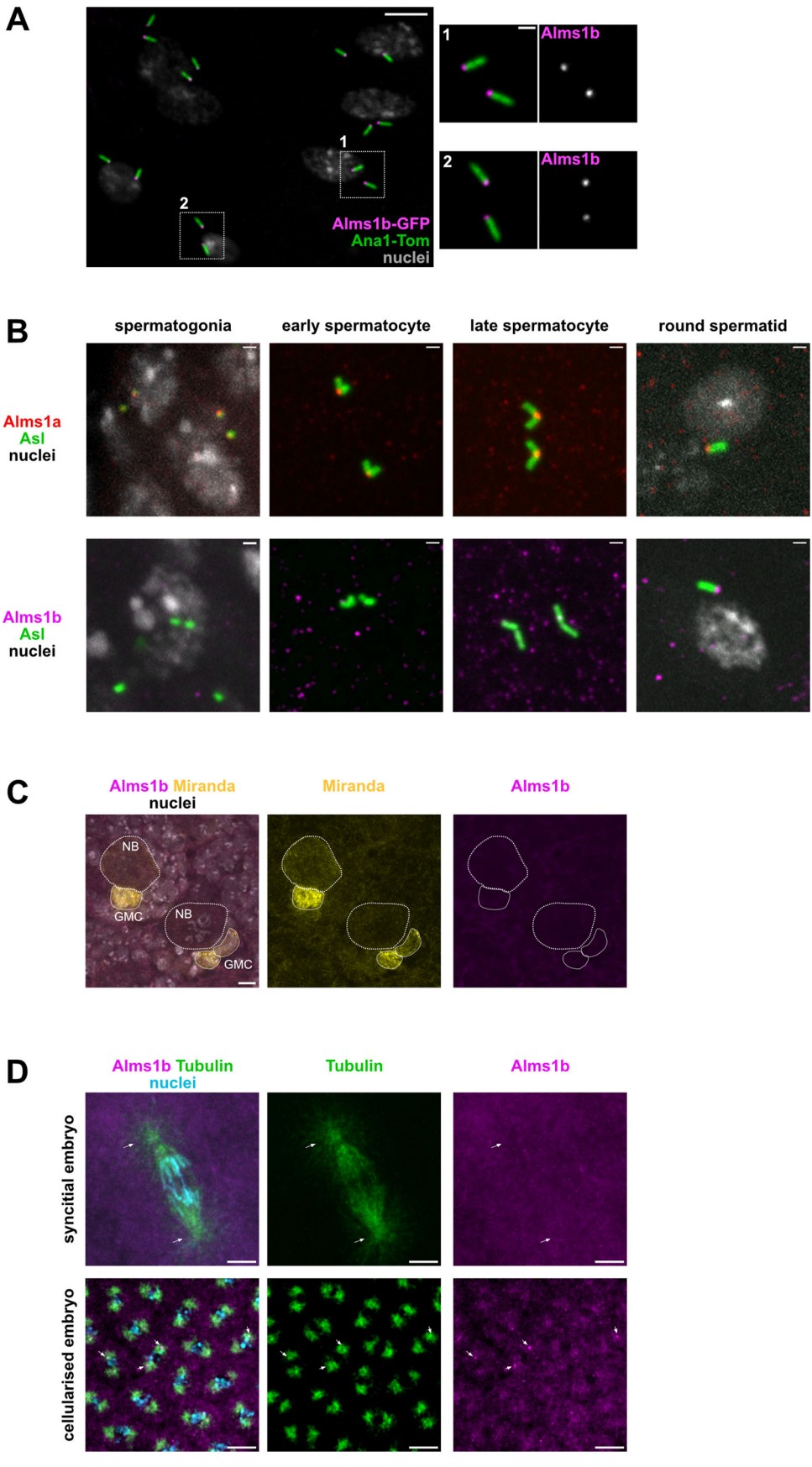

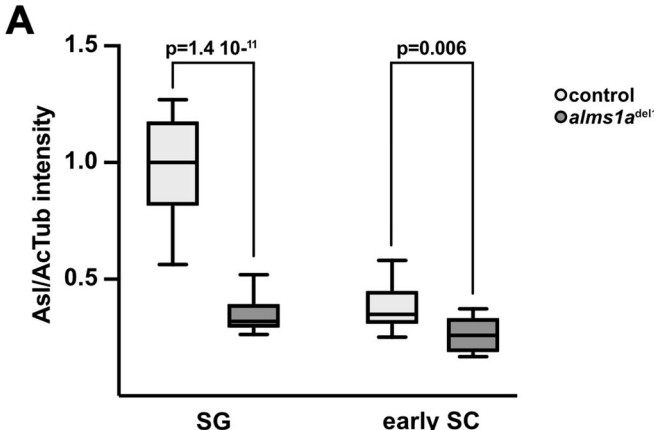

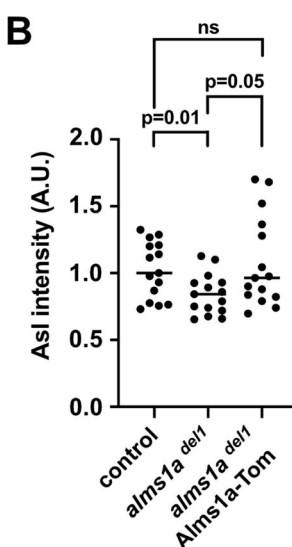

**Figure EV2. Asl recruitment in impaired upon loss of *alms1a*.**

(A) Comparison of Asl intensity at centrosomes normalised to acetylated α-tubulin intensity in U-ExM control and *alms1a*[del1] testis. Acetylated α-tubulin labels the centriole and was used as an internal reference for Asl intensities comparison between samples. Control: $n = 3$ testis, SG $= 20$ centrosomes, early SC $= 21$ centrosomes; *alms1a*[del1]: $n = 3$ testis, SG $= 12$ centrosomes, early SC $= 20$ centrosomes. The Box Plots show the interquartile range (IQR), with the median (50th percentile) indicated by the horizontal line inside the box and the whiskers extending to the 10th and 90th percentiles. (B) Comparison of Asl intensity at centrosomes of early SC of control, *alms1a*[del1] and *alms1a*[del1];;Alms1a-Tomato testes. Reduction of Asl intensity in *alms1a*[del1] is rescued to control levels upon introduction of Alms1a-Tomato transgene. Analysis based on immunofluorescence with anti Asl antibody. control: 3 testes, 15 centrosomes; *alms1a*[del1]: 2 testes, $n = 15$; *alms1a*[del1];;Alms1a-Tomato: 3 testes, $n = 15$. P values were obtained with two-sided unpaired Wilcoxon test. Source data are available online for this figure.

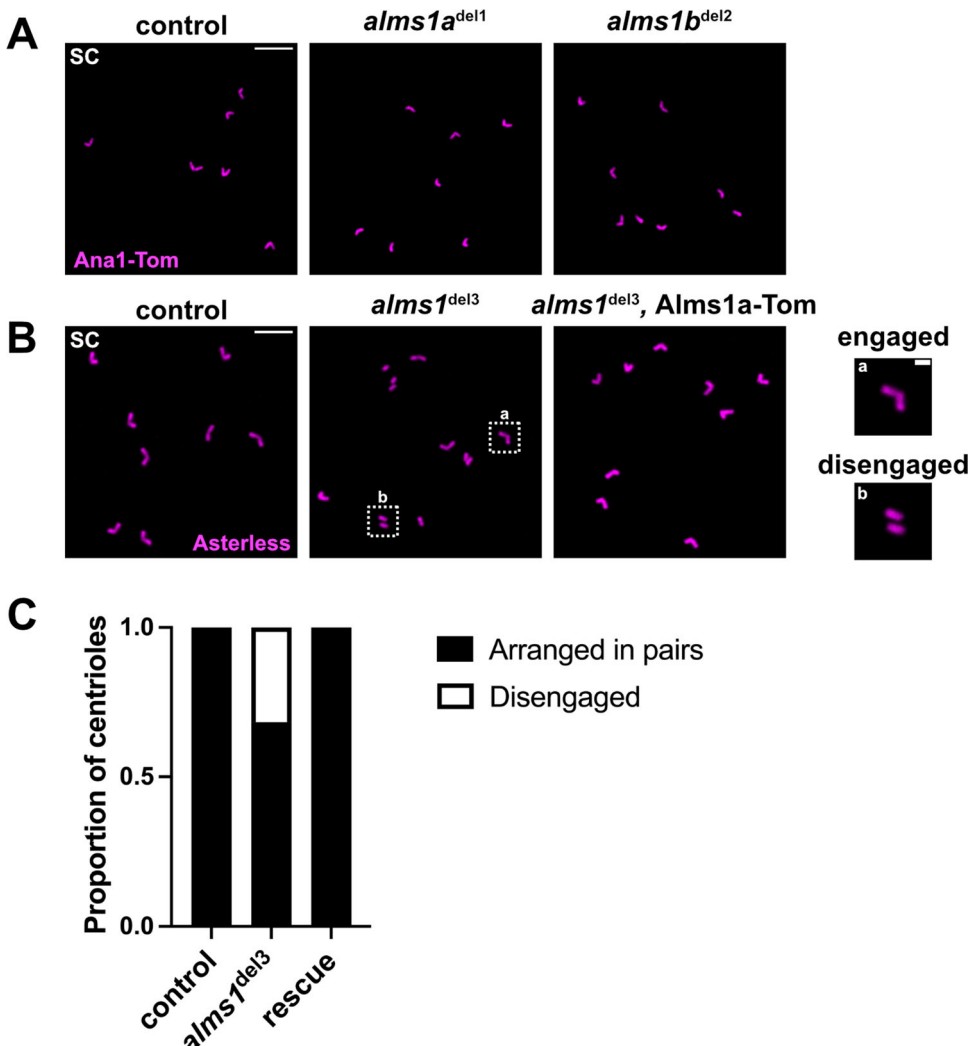

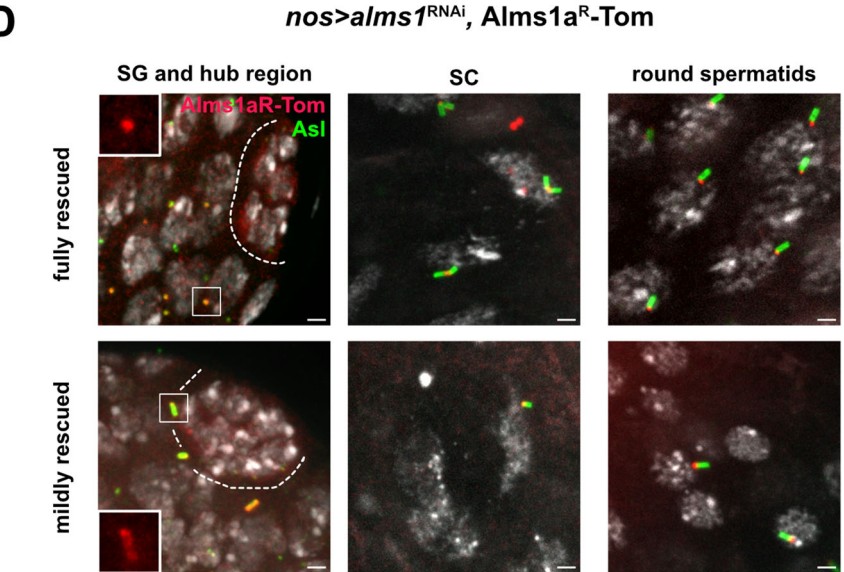

◀

**Figure EV3. Phenotypes of *alms1* mutants.**

Unexpended images of SC showing centrioles in (**A**) *alms1a*^del1^ (middle), *alms1b*^del2^ (right) and in *alms1b*^del2^ rescued by Alms1b-GFP as a control (left, similar behaviour is observed for *alms1a*^del1^ rescued by Alms1a-Tom). Ana1-Tomato (magenta) as a centriolar marker. (**B**) Centrioles in control (left), *alms1*^del3^ (middle) and *alms1*^del3^;;Alms1a-Tom (right). For *alms1*^del3^, a focus on engaged (a) and disengaged (b) centrioles is shown below. Asl (magenta) as a centriolar marker. (**C**) Quantification of centriole disjunction in control, *alms1*^del3^ and *alms1*^del3^;;Alms1a-Tom testes. The double *alms1a,b* deletion (*alms1*^del3^) results in defective centriole cohesion for 26% of centrioles ($n = 865$ centrioles, 4 males). Centriole cohesion defect is fully rescued by expression of Alms1a-Tom (*alms1*^del3^;;Alms1a-Tom) with none of the centriole pair observed presenting centriole cohesion defects ($n = 666$ centrioles, 3 males), as also observed in a wild-type line (100% centrioles in pairs, $n = 454$ centrioles, 3 males). (**D**) Partial rescue of the centriole duplication defect upon introduction in *nos*-Gal4>*alms1*^RNAi^ background of the Alms1a^R^-Tom transgene. The Alms1a^R^-Tom is a modified Alms1a-Tom that carries silent mutations aiming at providing resistance against *alms1*^RNAi^ mediated degradation (this study). 41% (11/27) of *nos*-Gal4>*alms1*^RNAi^;;Alms1a^R^-Tom testes did not show any rescue of the duplication defects. We however observed either mild or almost complete rescue of centriole duplication in respectively 48% (13/27) and 11% (3/27) of the observed testes. Bottom row: in mildly rescued testes few centrioles (marked with Asl, green) were observed in GSCs (cells contacting the hub, delineated with a white dotted line). When centrioles could be observed they were longer than in control conditions and were decorated by Alms1a^R^-Tom (red) on their full length (Alms1a^R^-Tom fluorescence of the boxed centriole is shown at the bottom left corner). SC (middle image) contained either no or only single centrioles and only a subset of round spermatids had a centriole (right). Top row: in fully rescued testes, centrioles co-labelled with Asl and Alms1a^R^-Tom were observed in GSCs (left image, inset shows Alms1a^R^-Tom localisation at the boxed centriole) and SG. SC contained pairs of centrioles (middle row) and most round spermatids contained a centriole (right). In both mildly and fully rescued testes, observed centrioles showed Alms1a^R^-Tom localisation at their proximal end, as observed in control conditions. The partial rescue achieved likely reflects an incomplete resistance of the Alms1a^R^-Tom construct to *alms1*^RNAi^ mediated degradation. Occurrence of rescue events by Alms1a-Tom nevertheless shows the specificity of the *alms1* RNAi construct used. Centrioles labelled with anti Asl (green), Alms1a^R^-Tom (red). Scale bars: (**A, B**) 5 μm, (**B**, insets) 1 μm, (**D**) 2 μm. Source data are available online for this figure.

 

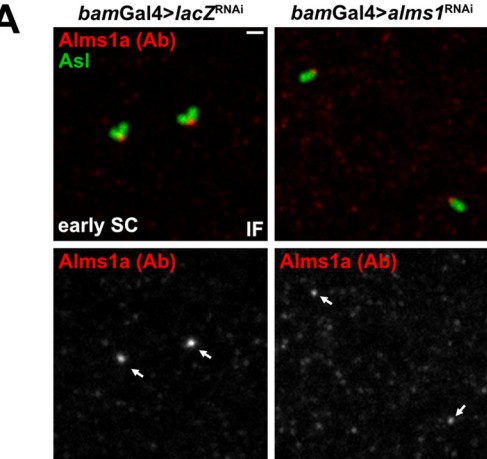

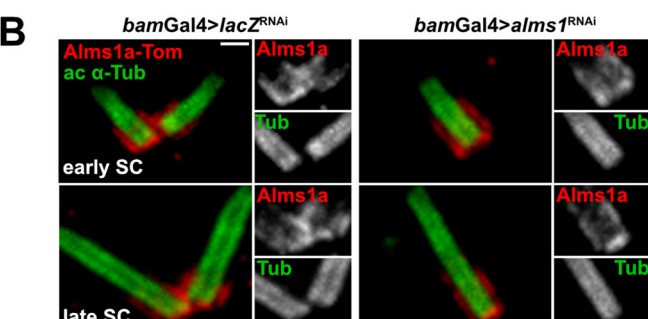

**Figure EV4. Alms1a localises at the proximal side of the remaining centriole in *bam*-Gal4>*alms1*<sup>RNAi</sup>.**

(A) Immunofluorescence images of *bam*-Gal4>*lacZ*<sup>RNAi</sup> and *bam*-Gal4>*alms1*<sup>RNAi</sup> stained with anti-Alms1a (red) and anti Asl (green) antibodies showing endogenous Alms1a localisation at the proximal end of the centrioles. (B) U-ExM images of Alms1a-Tomato (red) in *bam*-Gal4>*lacZ*<sup>RNAi</sup> (*bam*-Gal4>*lacZ*<sup>RNAi</sup>;;Alms1a-Tom) or *bam*-Gal4>*alms1*<sup>RNAi</sup> centrioles (*bam*-Gal4>*alms1*<sup>RNAi</sup>;;Alms1a-Tom). Centriolar walls are revealed with acetylated α-tubulin (green, ac α-tub). Alms1a-Tom forms a cup surrounding the proximal extremity of both mother and daughter centrioles in *bam*-Gal4>*lacZ*<sup>RNAi</sup> as well as the proximal side of the unduplicated centriole in *bam*-Gal4>*alms1*<sup>RNAi</sup>. Scale bars: (A) 1 μm, (B), after expansion factor correction, 250 nm.

**A**

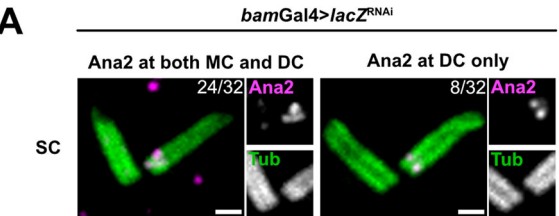

*bam*Gal4>*lacZ*^RNAi

**B**

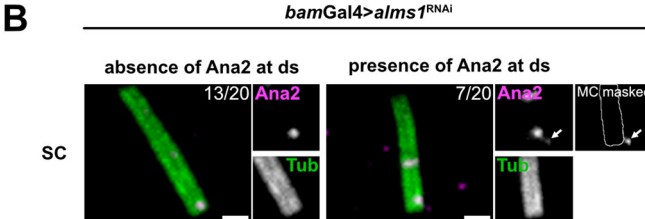

*bam*Gal4>*alms1*^RNAi

**Figure EV5.  Ana2 centriolar localisation in spermatocytes.**

U-ExM images of Ana2-mNeonGreen knock-in (Ana2-eNG, in magenta) in (**A**) *bam*-Gal4>*lacZ*^RNAi or (**B**) *bam*-Gal4>*alms1*^RNAi showing Ana2 localisation at centrioles in spermatocyte stage (SC). Centriolar walls are revealed with acetylated α-tubulin (green, ac α-tub), Mother centriole (MC, on the left of *bam*-Gal4>*lacZ*^RNAi images). Scale bars (after expansion factor correction): 250 nm.

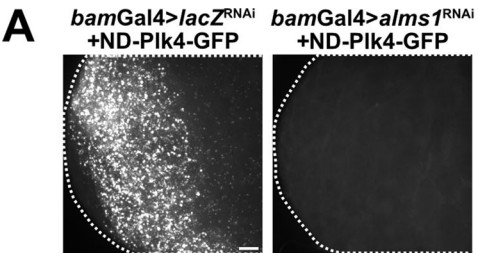

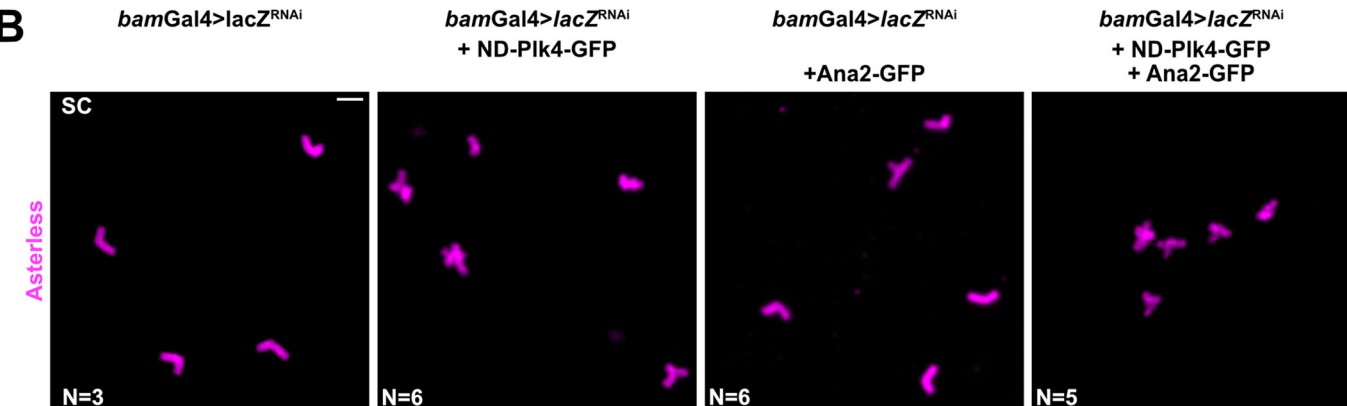

**Figure EV6.** **Overexpression of ND-Plk4-GFP in** *bam*-Gal4>*lacZ*^RNAi^ **and** *bam*-Gal4>*alms1*^RNAi^ **testes.**

(A) Images of unexpanded *bam*-Gal4 > {*lacZ*^RNAi^, ND-Plk4-GFP} or *bam*-Gal4 > {*alms1*^RNAi^, ND-Plk4-GFP testis hub regions. In control condition, ND-Plk4-GFP (grey) forms aggregates in the cytoplasm while we only observe diffuse ND-Plk4-GFP in *alms1*^RNAi^. The hub region is outlined with a dotted line. (B) Representative immunofluorescence images of centrosomes with or without centriole overduplication upon overexpression of ND-Plk4-GFP, Ana2-GFP or both in *bam*-Gal4>*lacZ*^RNAi^ genetic background. Following overexpression of ND-Plk4-GFP most centrosomes presented overduplication of centrioles including numerous events of centrosomes with more than 3 centrioles. In contrast, overexpression of Ana2-GFP induces the formation of few centrosomes with three centrioles while rosettes (> 3C) were never observed. Co-expression of ND-Plk4-GFP and Ana2-GFP leads to massive overduplication of centrioles, to an extent at least as important than observed with ND-Plk4-GFP alone. Centrioles labelled with anti Asl antibody (magenta). Scale bars: (A) 10 μm, (B) 1 μm.

