## [Peer Review File · The EMBO Journal]

Drosophila Alms1 proteins regulate centriolar cartwheel assembly by enabling Plk4-Ana2 amplification loop

Marine Brunet, Joëlle Thomas, Jean-Andre Lapart, Léo Krüttli, Marine Laporte, Maria Riparbelli, Giuliano Callaini, Bénédicte DURAND, and Véronique Morel

Corresponding authors: Véronique Morel (veronique.morel@univ-lyon1.fr) , Bénédicte DURAND (durand-b@univ-lyon1.fr)

Review Timeline:

Transferred from Review Commons:	3rd Jun 24
Editorial Decision:	10th Jun 24
Revision Received:	9th Nov 24
Editorial Decision:	12th Dec 24
Revision Received:	6th Jan 25
Accepted:	24th Jan 25

Editor: Ieva Gailite

Transaction Report:

This manuscript was transferred to The EMBO Journal following peer review at Review Commons.

Review #1**1. Evidence, reproducibility and clarity:****Evidence, reproducibility and clarity (Required)**

Much is known about the centriole duplication machinery and the centriole duplication cycle thanks to key screens performed in *C.elegans* in the past 20-25 years and subsequent work in *Drosophila* or human cells. It has been proposed that the core duplication machinery consists of 5 proteins implicating Cep192-PLK4-SAS6-SAS5 and SAS4, with specific variations according to the model system or cell line studied. Recently, a role for ALM proteins in the duplication of centrioles in the *Drosophila* male germ line has been reported, but in this study the mechanism involved has not been identified or described.

In the Brunet study, the authors show a role for Alm proteins in centriole duplication in different tissues in flies- showing that these proteins have more a ubiquitous role in centriole duplication rather than a restricted role in the male germ line as initially described. This is already accounting for the novelty of this paper. However, this study goes well beyond previous studies as it proposes two very novel concepts. The first one is that Alm proteins can stabilize PLK4, the sole kinase implicated in centriole duplication, and so be responsible for maintaining kinase activity during a time window where daughter centrioles are generated. This is extremely interesting and makes a lot of sense and the data is very convincing. The second novel concept is that loss of Alm can have different consequences according to the- I do not even know how to call it- loss condition- leading to Alm deficiency. While I think this is quite novel, interesting and maybe even real, I think the authors conclude very strongly on the differences between a gene knock out or gene knock down conditions. I think that they may want to tune down their conclusions related to this part, as many more data would be required to conclude in this way.

In other words, I think the paper is of high interest to a broad field of cell biology with interests in centrosome, cilia and the regulation of centriole duplication.

****Major points:****

1. I did not find a rescue experiment of the Alm deletions with the Alm transgenes.
2. If I understood the authors correctly, Alm def compensate for centrosome duplication by a yet unknown process, while Gal4-induced depletions do not. First, calling Gal4- induce RNAi- acute depletion is not correct. This is certainly not acute and so another designation has to be found. Acute is something like a degron such as Auxin or dTag where the protein is

degraded in a very acute manner. RNAi targets the mRNA, so if the protein is already made, it will not suffer from the RNAi treatment. Second, are the authors sure that either their crispr strategy did not generate any other knock out- hence the essentiality of rescuing these mutations. Or alternatively, are they sure about their RNAi conditions? So can they add the RNAi conditions on the background of their deficiency and see that nothing changes? Third if depletion through RNAi indeed leads to a more evident role of Alm proteins, one is expected to see this over time. Can they do a clone-using an FRT site recombined with their Alm mutation, so that the initial cell divisions does not contain Alm, and so it is expected to fail duplication, which may be overcome with time? So a large clone might have progeny with cells with centrosomes (the young ones) and without (the older ones). Can they show that RNAi depletion in cells that will generate sensory cilia- these are not assembled? Because I am assuming that the mutants are not uncoordinated...

3. If they think that Almdef flies compensate for Alm loss - can they analyse levels of PLK4, SAS6, Ana2 and SAS4 in the mother centrioles?

****Minor points****

1. Some figures (the majority) lack scale bars.
2. I do not think that one can consider centrioles that are not in rosettes to be made de novo. They might just have disengaged. The "novo" centriole should be removed. Actually, PLK4 ND generates extra centrosomes, this is sufficient.

2. Significance:

Significance (Required)

The article by Brunet and colleagues investigates the role of ALMs proteins in centriole duplication. Centrioles are the core constituents of centrosomes and as such contribute to microtubule nucleation. Centrioles can behave as basal bodies, providing essential function in cilia assembly and function.

Here the authors have used Drosophila to characterize the role of Alm proteins. They show that 2 isoforms with distinct behaviour are expressed with different localizations. Further, through the assessment of different tissues and loss-of-function conditions, they propose a role for Alm proteins in centriole duplication.

Overall the paper is very well written and easy to follow.

3. How much time do you estimate the authors will need to complete the suggested revisions:

Estimated time to Complete Revisions (Required)

(Decision Recommendation)

Between 1 and 3 months

Yes

Review #2

1. Evidence, reproducibility and clarity:

Evidence, reproducibility and clarity (Required)

The study takes a look at the role of Alms1a and b in centriole formation in *Drosophila* tissues using genetics and microscopy approaches. Alsm1a RNAi was previously shown to lead to loss of centrioles on the male germ line, but the molecular mechanism of how Alsm1a provides this function is not known. This is what this manuscript addresses.

The general proposal of this manuscript is that Alsm1a is a key regulator of centriole formation in *Drosophila* tissues (embryos, male germ line and neuroblasts analysed). Alms1a is an inner PCM protein and functions downstream of PLK4 to drive procentriole formation.

This is an interesting advance in understanding of centriole formation in flies. The authors managed to pull off beautiful expansion microscopy and produced images of centrioles both by immunofluorescence and electron microscopy. The data quality is very high, the quantifications lack a little behind in places. The manuscript would benefit from thinking about structure and presentation. The proposal that chronic loss of Alms1a (mutants) is well tolerated while acute (RNAi) is not is a bit puzzling. It is not impossible, but RNAi is not that acute either. It would be important to clarify this.

To make sure there is no glitch in the tools, the newly generated *alms1* deletions should be better characterised to clarify whether a compensatory mechanism exists, that upon chronic depletion rescues all phenotypes described by RNAi. There are also ways to test whether the *Alsm1a* fluorescent transgene introduce confounding effects, which if tested would be an improvement (see below). The genetic and immunofluorescence analysis got the authors quite far, the manuscript lacks biochemical analysis to strengthen the proposed molecular mechanism and clarify whether key and easy to predict interactions of *Alms1* actually do occur. This would be a big plus but is not limiting. There are also inconsistencies that need clarification, for instance the title of figure 6 is that *Alms1* proteins act downstream of *Plk4*, yet the model in the discussion proposes that *Alsm1* stabilises *Plk4* which does not fit and hinges on quantification of *Plk4* levels that is perhaps currently not robust enough. The statement that *Alms1* is required for *PLK4* activity is too strong based on the data provided or provide data on *PLK4* activity. The attempt to check what is going on in other systems is appreciated, in RPE cells *Alms1* comes in only after 120nm elongation. Some of the quantifications could be done more robustly, using ratiometric analysis rather than directly comparing intensity levels. Referencing in the manuscript and discussion should be improved.

****Specific points****

1. Provide a thorough characterisation of the *alms1* deletions generated using qPCR to measure RNA levels in the flies. Provide whether the alleles are viable and fertile and clarify all genotypes in the manuscript. Can the mutants suppress the *Plk4*-ND overexpression effect?
2. *BamGal4 alms1a* RNAi, SCs loose centrioles, those that keep centrioles, have *Alms1a*, but fail to initiate procentriole formation. To strengthen this view please provide *nos-Gal4 alms1a* RNAi, *Alms1a*-Tomatoe data showing that now *Alms1a*-Tomato is not present accept on the mother centrosomes in GCS, ruling out anything unpredicted happens by introducing the *Alsm1a*-Tomatoe.
3. Quantifications, measure fluorescence ratios (e.g. Figure 3 C,D quantify the ratio of *Asl/PlP* to *Bld10*, similarly for *PLK4*-ND in the relevant figure).
4. A model for *Alms1a* in the style of *Figur5A* would be great.
5. What happens to *Alsm1a* upon *Plk4* inhibition? Experiments like this could strengthen the validity of the hierarchy of events proposed.

****Other comments****

The organisation and presentation should be improved. The figures reorganised to group

what belongs together, I would suggest moving human RPE cell analysis to supplementary data and bring the beautiful EM of BamGal4-Alms1 RNAi, Alsm1a-Tomatoe into the main manuscript.

In the deletions asterless levels are reduced in SC but not in the testis tip, why is that?

Conclusion: "while Alms1b is only detected on mature centrioles as it is absent from rapidly duplicating centrioles of syncytial embryos or from duplicating centrioles of dividing neuroblasts." Perhaps add, Alms1b when expressed.

BamGal4 alms1a RNAi, SCs loose centrioles, those that keep centrioles, have Alms1a, but fail to initiate procentriole formation. To strengthen this view please provide nos-Gal4 alms1a RNAi, Alms1a-Tomatoe data showing that now Alms1a-Tomato is not present except on the mother centrosomes in GCS.

The idea to compare nos-gal4 and bam-gal4 driving Alms1a RNAi is good. Providing a scheme of when these are expressed in the male germ line will help illustrate the experimental strategy. BamGal4 Alsm1a RNAi leads to loss of centrioles in SCs

Alms1b (CG12184) is according to published RNAseq data not expressed in neuroblasts, so it makes sense they do not observe (Knoblich lab data), which could be cited here: "in agreement with RNA-seq data showing low expression of alms1b in neurons and glial cells (Li et al., 2022)."

Fig S5 what cells were analysed by TEM?

Fig S6 monochrome images confirm what is what.

Figure 4 A for clarity it would be helpful to provide landmarks, marker for stem cells (Vasa) or the Hub to be able to understand what we are looking at. Same for Fig 4E, Mira or Dpn? The DAPI staining does not allow in the images provided to identify NBs.

Figure 5

For the fluorescence profile plots it would be nice to see the average plus the standard deviation of the signal in the quantifications.

Figure 6 D is not convincing, how were the dots visible chosen for the quantification?

Figure 1C. Quantification of protein levels is not provided (is this because the expanded ultrastructural approach is not linear precluding quantification?)

Figure 2 B radial dimensions, it would be great to show the sample average and standard deviation.

I don't find figure S2 helpful

2. Significance:

Significance (Required)

Understanding the process of centriole duplication is an important topic that should be relevant to a broader cell biological community especially since the proteins of interest have disease relevance.. This study provides new insights into how Plk4 dependent centriole duplication takes place in *Drosophila*.

The manuscript is strong on the microscopic images both immune fluorescence and TEM. The function of Alms proteins is dissected genetically no biochemical analysis is provided, which is a limitation. Another limitation currently is the uncertainty about the discrepancy between the RNAi and the mutant results. If the mutants are confirmed and technical issues can be ruled out the finding that a compensatory mechanism exists that suppresses chronic loss of Alms1 proteins could be very interesting.

3. How much time do you estimate the authors will need to complete the suggested revisions:

Estimated time to Complete Revisions (Required)

(Decision Recommendation)

Between 3 and 6 months

Yes

Review # 3

1. Evidence, reproducibility and clarity:

Evidence, reproducibility and clarity (Required)

Brunet and colleagues utilized expansion microscopy to identify the distinct localizations of Alms1a and Alms1b at the centrosome in transgenic flies expressing Alms1a-Tomato or Alms1b-GFP-6Myc. Their imaging results are of exceptional quality and include insightful observations and discussion. However, these experiments were exclusively conducted in the transgenic flies, and no evidence has been provided for the expression of endogenous Alms1b proteins. Moreover, the defect in centriole duplication was observed only when Alms1 was depleted via RNA interference, not through gene knockout. Although some discussion is provided to address this discrepancy, the evidence remains unconvincing. More robust evidence is necessary to support their findings.

Major comments

1. One of the primary observations is the defect in centriole duplication in the absence of Alms1. This finding was only observed with RNAi and not with gene knockout. The authors performed the RNAi experiments in Alms1-deleted cells (Alms1del3) and did not observe the defects in centriole duplication. Therefore, authors concluded that acute loss of Alms1 is responsible for centriole duplication in germline stem cells. However, it is still difficult for me to reach that conclusion with only this evidence.

A. Could the authors provide the western blot and staining results of Alms1a and Alms1b proteins after RNAi and gene knockout?

B. The most reliable and widely accepted method to determine the specificity of RNAi (whether it is an off-target effect or not) is through genetic rescue experiments. Could the authors provide rescue data? It would be great if the authors could show the rescue of centriolar signals of Plk4, Sas-6, and Ana2, in addition to centriole duplication.

C. The authors failed to discriminate which gene is crucial for centriole duplication in GSGs due to the sequence similarity. Linked to the above point, could the authors show the rescue results with either or both Alms1a and/or Alms1b genes? If only one gene can rescue the centriole duplication defects, it would clarify the specific functions of Alms1a and Alms1b in this process.

D. (Optional) Considering the results in the manuscript, the authors appear to have good techniques in genetic manipulation in flies. If so, generating transgenic flies of Alms1a and Alms1b tagged with a degron (e.g., dTAG or destabilization domain [DD]) to observe centriole duplication defects upon rapid protein degradation might be feasible to support their observations.

2. There were no results with endogenous Alms1a and Alms1b in the manuscript. Could the authors provide immunostaining results for endogenous Alms1a and Alms1b? If expansion microscopy is not available due to antibody specificity issues, standard confocal imaging would be sufficient.

3. In figure 3C and D, the authors noted measurements from 7 to 8 testes per group. It would be helpful to also know how many centrosomes were measured, as done in Figure 4C.

****Minor comments****

1. In figure S1, the orientation of the gene is not consistent between endogenous and transgenic Alms1. Could the authors make it consistent for readers to intuitively recognize it? Additionally, the promoter used for transgenic gene expression is not specified. Please provide this information as well.

2. On page 5, the statement "whereas Alms1b is associated with post-duplication maturation of the centriole." needs revision. The authors observed strong Alms1b-GFP signals in the centrioles at the end of the SC stage or during meiotic division but did not observe any defects in centriole maturation at the current stages. Furthermore, Alms1b was not detected from rapidly duplicating centrioles in syncytial embryos and dividing neuroblasts (Page 6), making it hard to understand that Alms1b has a role in post-duplication centriole maturation.

3. In figure S2, the centrosome drawings are unclear (initially thought to be 'v' marks). Could a centrosome drawing be added to the legend for clarity?

2. Significance:

Significance (Required)

ALMS1 is a well-known protein which is important for cilia formation in human cells. Recent research by Chen and Yamashita (eLife, 2020) highlighted its significance in centrosome duplication in *Drosophila*'s germ line stem cells (GSGs). Brunet and colleagues' findings align with these previous studies but go further by elucidating the localization of Alms1a as a centriolar and pericentriolar material protein, and Alms1b

solely as a centriolar protein, using expansion microscopy. Moreover, their research suggests that only acute, not chronic, loss of Alms1 leads to defects in centriole duplication, proposing that cells may develop compensatory mechanisms for Alms1 loss in GSGs.

3. How much time do you estimate the authors will need to complete the suggested revisions:

Estimated time to Complete Revisions (Required)

(Decision Recommendation)

Between 3 and 6 months

Yes

Revision Plan

Manuscript number: RC-2024-02443

Corresponding author(s): Véronique Morel, Bénédicte Durand

[The “revision plan” should delineate the revisions that authors intend to carry out in response to the points raised by the referees. It also provides the authors with the opportunity to explain their view of the paper and of the referee reports.]

The document is important for the editors of affiliate journals when they make a first decision on the transferred manuscript. It will also be useful to readers of the reprint and help them to obtain a balanced view of the paper.

*If you wish to submit a full revision, please use our "Full Revision" template. **It is important to use the appropriate template to clearly inform the editors of your intentions.**]*

1. General Statements [optional]

This section is optional. Insert here any general statements you wish to make about the goal of the study or about the reviews.

We thank the referees for their insightful comments. We have already addressed most of their comments as stated in the point-to point answer to each reviewer and our revision plan will permit to strengthen our conclusions and respond to the remaining reviewer concerns not yet addressed.

Referees shared a concern regarding the specificity of the genetic tools used here (*alms1* genes deletion and *alms1*^{RNAi}). These deletions have been sequenced. We could not detect RNA by RT-PCR covering the remaining *alms1a* sequence in the *alms1*^{del1} allele, suggesting that no protein could be produced from this deleted locus. To exclude a potential insertion of the deleted genes elsewhere in the genome we have performed a PCR using primer couples internal to each gene on genomic DNA extracted from *alms1a*^{del1}, *alms1b*^{del2} and *alms1*^{del3} flies. More, we have additional data that allow to already demonstrate the rescue effect in the double deleted line further validating our deletions. We plan to complete the data with the rescue of RNAi conditions with RNAi resistant transgene.

We have also preliminary data to support our model: in particular we have data showing that *alms1*^{RNAi} can be partially rescued by concurrent expression of both non-degradable Plk4 and Ana2, whereas their single expression is not sufficient, supporting our conclusion that Alms1 is required to reinforce the Plk4/Ana2 amplification loop. Second, we show that concurrent overexpression of Alms1a and Plk4 dramatically increases centriole amplification in agreement with our proposed model.

Reply to Referee 1

2. Description of the planned revisions

Insert here a point-by-point reply that explains what revisions, additional experimentations and analyses are planned to address the points raised by the referees.

Referee 1 point 1: I did not find a rescue experiment of the *Alms1* deletions with the *Alms1* transgenes.

Referee 1 point 2: “Second, are the authors sure that either their crispr strategy did not generate any other knock out- hence the essentiality of rescuing these mutations.”

- For *alms1b*^{del2} allele that only removes *Alms1b*, we do not observe any centriolar phenotype nor any other specific phenotype, therefore we cannot evaluate at this point any rescue by the *Alms1b*-GFP transgene.
- Regarding *alms1a*^{del1} allele, the only centriolar phenotype is the reduced accumulation of PCM proteins Asterless and Plp at centrioles (Fig. 3C,D). To exclude a potential off-target effect of the CRISPR, we will repeat these measures comparing i) control, ii) *alms1a*^{del1} and iii) *alms1a*^{del1} rescued by *Alms1a*-Tom conditions.
- Last, for the double *alms1a* and *b* deletion (*alms1*^{del3}), rescue experiments by *Alms1a*-Tom have been included in the manuscript (cf following section).

Referee 1 point 2: “Or alternatively, are they sure about their RNAi conditions? So can they add the RNAi conditions on the background of their deficiency and see that nothing changes?”

Indeed, the difference between the strong phenotype induced by the RNAi and the weaker phenotype associated with the chronic genomic deletion of *alms1a,b* could be due to off-target effect of the RNAi. We have tested this hypothesis as described in the initial version of the manuscript. We indeed have performed *alms1*^{RNAi} depletion in the *alms1a,b* deleted background (*alms1*^{del3}, *nosGal4>alms1*^{RNAi}). This result was presented on Figure 4A where *alms1*^{RNAi} depletion in the *alms1a,b* deletion background (*alms1*^{del3}, *nosGal4>alms1*^{RNAi}) shows the same phenotype as the *alms1*^{del3}, *nos-Gal4* condition with no centriole duplication defect. We can thus conclude that the phenotype observed with *alms1*^{RNAi} is indeed specific to *alms1* depletion.

To further validate the specificity of the *alms1*^{RNAi}, we are generating an RNAi resistant *Alms1a* transgene (*UAS-Alms1a*^R, see answer to Referee 3 points 1BC, planned experiments) to rescue the centriole duplication phenotype of the *alms1*^{RNAi} upon expression of *Alms1a*^R. Results will be obtained in two months.

Referee 1 point 3: “If they think that *Alms* def flies compensate for *Alms* loss - can they analyse levels of PLK4, SAS6, Ana2 and SAS4 in the mother centrioles?”

We are limited by the number of antibodies available to perform quantitative western blots on the endogenous proteins from the testes. Therefore, we need to introduce endogenously tagged Plk4 (*Plk4*-

eGFP) or Ana2 (Ana2-eNG) alleles, already used in the manuscript, as well as Sas6-eNG in the *alms1^{del3}* genetic background. We will quantify by immunofluorescence Plk4, Ana2 or Sas-6 levels at centrioles in control versus *alms1^{del3}* conditions.

3. Description of the revisions that have already been incorporated in the transferred manuscript

Please insert a point-by-point reply describing the revisions that were already carried out and included in the transferred manuscript. If no revisions have been carried out yet, please leave this section empty.

Referee 1 point 1: I did not find a rescue experiment of the *Alms1* deletions with the *Alms1* transgenes.

Referee 1 point 2: “Second, are the authors sure that either their crispr strategy did not generate any other knock out- hence the essentiality of rescuing these mutations.”

We have now included the quantification of the centriole cohesion defects observed in *alms1* double mutant and in rescue conditions (Fig. S4). We can thus conclude that the cohesion defect observed in *alms1^{del3}* is due to the deletion of *alms1a* and *alms1b* genes and not to an off-target effect of the CRISPR strategy. Furthermore, it shows that the re-expression of *Alms1a-Tom* is sufficient to restore centriole cohesion and thus that the transgene is functional. Last, because this phenotype is only present in the double mutant, but not in each single mutant, we can conclude that centriolar cohesion is an *Alms1* specific defect that relies on both *Alms1a* and *Alms1b*.

The manuscript has been modified as follows on p7: “While centriole duplication in *alms1^{del3}* flies is largely normal (Fig. 4A left, S4B), we observed a premature disjunction of centrioles in 26% of centriole pairs of *alms1^{del3}* testes (n=865 centrioles, 4 males) (Fig. S4B middle, focus b). This phenotype was fully rescued by expression of *Alms1a-Tom* (*alms1^{del3}*, *Alms1a-Tom*; n=666 centrioles, 3 males), together validating the specificity of the *alms1^{del3}* CRISPR deletion and the functionality of our *Alms1a-Tom* transgene (Fig. S4B right).” The Figure S4, showing centriole cohesion in *alms1^{del3}* has been modified to include the quantification of the cohesion defects in i) control ii) *alms1^{del3}* and iii) *alms1^{del3}* rescued by *Alms1a-Tom* conditions and representative images.

Referee 1 point 2: “calling Gal4- induce RNAi- acute depletion is not correct. This is certainly not acute and so another designation has to be found.”

We agree with the referee that the RNAi strategy does not remove the pre-existing protein contribution. We used “acute” for our RNAi experiments to convey two ideas. First that the depletion was sudden and not permanently encoded genetically as with the deletions. The point with the RNAi experiment is indeed to alleviate potential genetic compensation mechanisms that can be triggered in chronic genetic contexts and hence partially or fully mask the effect of the protein loss. Second to reflect the strength of the induced depletion.

Revision Plan

This adjective has been long and commonly used in the centriole/cilia field to convey these two ideas, as shown by few examples below.

- In their article published in *J Cell Science* in 2011 and entitled “Spindle positioning in human cells relies on proper centriole formation and on the microcephaly proteins CPAP and STIL”, Kitagawa D, (...) Gönczy P. oppose chronic genetic inactivation and acute RNAi depletion: “This apparent discrepancy might be explained by residual function of the mutant proteins or else compensatory mechanisms upon chronic inactivation that are not operating upon **acute depletion mediated by RNAi.**”
- Hall, E.A., (...) P. Mill. write in “**Acute versus chronic loss** of mammalian Azi1/Cep131 results in distinct ciliary phenotypes” (*PLoS Genet.* 2013, doi:10.1371/journal.pgen.1003928): “we present here detailed analysis of both **acute loss (by siRNA)** and chronic absence (by genetic mutation)”
- In 2020, Aydin ÖZ, Taflan SO, Gurkaslar C, Firat-Karalar in “**Acute** inhibition of centriolar satellite function and positioning reveals their functions at the primary cilium.” (*PLoS Biol* 18(6), DOI: [10.1371/journal.pbio.3000679](https://doi.org/10.1371/journal.pbio.3000679)), write: “Accordingly, **acute or constitutive** loss of satellites through depletion or deletion of PCM1”
- In their review “Centriolar satellite biogenesis and function in vertebrate cells” (*J Cell Sci* (2020), doi: [10.1242/jcs.239566](https://doi.org/10.1242/jcs.239566)), S. Prosser and L. Pelletier also qualify siRNA depletion as acute. “Significantly, **acute (siRNA-mediated)** loss of the satellite protein CEP131”.

In order to remove any ambiguity regarding what is meant by acute, we have defined acute the first time it appears in the text and kept the term in the manuscript with respect to previous usages in the field. The text now appears p4: “We observe that sudden and severe, hereafter referred to as acute, depletion of both Alms1a and b by RNAi is associated with complete centriole duplication failure in several *Drosophila* tissues”

Referee 1 minor point 1: Some figures (the majority) lack scale bars.

For each figure, and unless written otherwise, the scale bar put in one image of a panel applies to the whole panel. We made that choice as adding scale bars in all images made the figures more complicated to read. Some panels were however indeed missing a scale bar. This has been corrected.

Referee 1 minor point 2- I do not think that one can consider centrioles that are not in rosettes to be made *de novo*. They might just have disengaged. The “*novo*” centriole should be removed. Actually, PLK4 ND generates extra centrosomes, this is sufficient.

The text has been modified to remove “*de novo*” (p11).

4. Description of analyses that authors prefer not to carry out

Please include a point-by-point response explaining why some of the requested data or additional analyses might not be necessary or cannot be provided within the scope of a revision. This can be due to time or resource limitations or in case of disagreement about the necessity of such additional data given the scope of the study. Please leave empty if not applicable.

Referee 1 point 2: Third if depletion through RNAi indeed leads to a more evident role of Alms proteins, one is expected to see this over time. Can they do a clone-using an FRT site recombined with their Alms mutation, so that the initial cell divisions do not contain Alms, and so it is expected to fail duplication, which may be overcome with time? So a large clone might have progeny with cells with centrosomes (the young ones) and without (the older ones).

Indeed, generating flip-out clones could mimic the RNAi knock-down of *alms1*. The experiment as suggested can unfortunately not be performed for the following technical reason.

Such Flip-out strategy relies on the recombination between homologous chromosome during mitosis but *alms1a* and *alms1b* are on the X chromosome. In absence of homologous wild-type chromosome in males the flip-out strategy can thus not be performed.

However, a possible genetic strategy would be to induce conditional deletion of the *alms1a* and *alms1b* genes using Lox or FRT sites, but this would require constructing new genetic alleles and finding ways of tracking inactivated cells, which is probably not feasible given that these specific loci have only a very small intron.

Referee 1 point 2: Can they show that RNAi depletion in cells that will generate sensory cilia- these are not assembled? Because I am assuming that the mutants are not uncoordinated...

We decided to focus this manuscript on the, already complex, role of Alms1 in centriole duplication. It is however indeed very interesting to question the role of Alms1 in ciliogenesis and this will be the topic of a future study.

Reply to Referee 2

2. Description of the planned revisions

Insert here a point-by-point reply that explains what revisions, additional experimentations and analyses are planned to address the points raised by the referees.

Referee 2 point 1: “Provide a thorough characterisation of the *alms1* deletions generated using qPCR to measure RNA levels in the flies.”

As already stated in the manuscript, the precise deletion points for each deletion generated by CRISPR are mentioned in the method section and in the legend of Figure S1.

- *alms1a*^{del1} “is a 4270 bp deletion between the position X: 4.635.793, after aa36 in the first exon, and the position X: 4.631.522 in *alms1a* 3’UTR. This deletion removes almost all *alms1a* coding sequence introducing a stop codon at position aa38 of the remaining sequence.”
- For *alms1b*^{del2}, the miniwhite gene has been introduced in place of the gene sequence “between the positions X: 4.627.368 (14nt after *alms1b* 3’UTR) and X: 4.630.821 (after aa21), it generated a deletion of 3542 bp, removing almost all *alms1b* coding sequence”.
- *alms1*^{del3} has been generated by creating the *alms1b*^{del2} in the *alms1a*^{del1} line.

We have performed RT-PCR with primers covering the few remaining 5’ sequences in the putative transcript, but did not detect any transcript compared to control flies. This information is provided here for the reviewer (or data available upon request). More, to demonstrate that no remaining ALMS1 coding sequence is present in the genome of the mutant flies to exclude a potential insertion of the deleted genes elsewhere in the genome, we have performed PCR using primer couples specific to each gene. We did not detect any ALMS1 coding sequence in the genome (gel image added in Fig.S1).

Referee 2 point 1 (suite): Can the mutants suppress the Plk4-ND overexpression effect?

It is indeed an interesting question. We plan to express the ND-Plk4 construct in the *alms1*^{del3} condition and quantify the penetrance of the centriole overduplication phenotype compared to ND-Plk4 expression in a control condition.

Referee 2 point 3: “Quantifications, measure fluorescence ratios (e.g. Figure 3 C,D quantify the ratio of Asl/PIP to Bld10, similarly for PLK4-ND in the relevant figure).”

In our hands, Bld10-GFP is not detected in spermatogonia and could thus not be used as an internal reference for quantification. Neither could Dilatory-GFP, another centriolar protein. Ana1 is commonly used as a centriole marker but it is involved in PCM recruitment. Its overexpression as Ana1-GFP may therefore interfere with *alms1a*^{del1} phenotype. No antibodies specific of other centriolar proteins are available. We hence will repeat this analysis using U-ExM as we will be able to use acetylated α -tubulin staining of the centriole as a reference.

Referee 2 point 4: “A model for Alms1a in the style of Fig5A would be great.”

Revision Plan

This will be included in the final version.

Referee 2 point 5: “What happens to Alsm1a upon Plk4 inhibition? Experiments like this could strengthen the validity of the hierarchy of events proposed.”

We are currently looking at Alms1a-Tom in *plk4*^{RNAi} conditions.

We also have also initiated several experiments to attempt a rescue of the duplication failure phenotype associated with *alms1*^{RNAi}. Hence, while overexpressing either Ana2-GFP or ND-Plk4-GFP (Fig. 6) does not rescue the duplication failure associated with *alms1*^{RNAi}, we have preliminary results showing that co-overexpression of Plk4-ND and Ana2 partially rescues the duplication loss. We are currently repeating this experiment both in IF and U-ExM to strengthen the result.

Hence, Alms1 requirement can be by-passed by increasing simultaneously non-degradable Plk4 and Ana2, further supporting our hypothesis that Alms1 is required for Plk4-Ana2 amplification loop at the duplication site.

Referee 2 Other comments 1

“The organisation and presentation should be improved. The figures reorganised to group what belongs together, I would suggest moving human RPE cell analysis to supplementary data and bring the beautiful EM of BamGal4-Alms1 RNAi, Alsm1a-Tomatoe into the main manuscript.”

We will move the EM data (Figure S5) in the main figures in the final version of the manuscript as figures will certainly evolve with the new experiments on-going.

Referee 2 Other comments 4 “BamGal4 *alms1a* RNAi, SCs loose centrioles, those that keep centrioles, have Alms1a, but fail to initiate procentriole formation. To strengthen this view please provide nos-Gal4 *alms1a* RNAi, Alms1a-Tomatoe data showing that now Alms1a-Tomato is not present except on the mother centrosomes in GCS.”

We will introduce the Alms1a-Tom transgene in the *nanos-Gal4>UAS-alms1*^{RNAi} expressing flies.

3. Description of the revisions that have already been incorporated in the transferred manuscript

Please insert a point-by-point reply describing the revisions that were already carried out and included in the transferred manuscript. If no revisions have been carried out yet, please leave this section empty.

Referee 2 point 1: “Provide whether the alleles are viable and fertile and clarify all genotypes in the manuscript”

We added in Fig. S1 a table clarifying viability and fertility of these lines. We have included a table with the full genotypes and the reference to the related figure in the Method section.

Referee 2 Other comments 3 “Conclusion: “while Alms1b is only detected on mature centrioles as it is absent from rapidly duplicating centrioles of syncytial embryos or from duplicating centrioles of dividing neuroblasts.” Perhaps add, Alms1b when expressed.”

We removed the reference to syncytial embryos and dividing neuroblasts here as it was redundant with the previous paragraph.

Referee 2 Other comments 5 “The idea to compare nos-gal4 and bam-gal4 driving Alms1a RNAi is good. Providing a scheme of when these are expressed in the male germ line will help illustrate the experimental strategy.”

The Fig. S2 has been modified to make the expression profile of *nanos-Gal4* and *bam-Gal4* clearer.

Referee 2 Other comments 6 “Alms1b (CG12184) is according to published RNAseq data not expressed in neuroblasts, so it makes sense they do not observe (Knoblich lab data), which could be cited here: “in agreement with RNA-seq data showing low expression of *alms1b* in neurons and glial cells (Li et al., 2022).””

The reference Berger et al. 2012 has been added.

Referee 2 Other comments 7 “Fig S5 what cells were analysed by TEM?”

Figure S5 shows centrioles from spermatocytes. The figure legend has been modified as follows:

“TEM images of *bam-Gal4>lacZ^{RNAi}* or *bam-Gal4>alms1^{RNAi}* mature primary spermatocytes”

Referee 2 Other comments 8 “Fig S6 monochrome images confirm what is what.”

The figure has been modified to add this information.

Referee 2 Other comments 11 “Figure 6 D is not convincing, how were the dots visible chosen for the quantification?”

Figure 6D shows spermatogonia in U-ExM with multiple (*bamGal4>lacZ^{RNAi}*) or single (*bamGal4>alms1^{RNAi}*) centrioles, easily identified based on acetylated α -tubulin staining. Quantification in Figure 6E was performed on similar U-ExM images. We changed the *bamGal4>lacZ^{RNAi}* image for one

of older spermatogonia with large centriole amplification on which centrioles were more distinguishable.

Referee 2 Other comments 12 “Figure 1C. Quantification of protein levels is not provided (is this because the expanded ultrastructural approach is not linear precluding quantification?)”

The plots shown in Figure 1 aim at providing a better description of the localisation of Alms1a with respect to other proteins. They are not quantification of protein levels which we do not think is important here as we only qualitatively describe the presence and dynamics of the protein. We have added plots for NBs and embryos to provide a better description of the localisation of Alms1a with respect to PCM protein Asterless.

Referee 2 Other comments 13 “Figure 2 B radial dimensions, it would be great to show the sample average and standard deviation.”

These were on the graph but too faint to be seen on the figure. We have improved the representation for it to be more visible.

4. Description of analyses that authors prefer not to carry out

Please include a point-by-point response explaining why some of the requested data or additional analyses might not be necessary or cannot be provided within the scope of a revision. This can be due to time or resource limitations or in case of disagreement about the necessity of such additional data given the scope of the study. Please leave empty if not applicable.

Referee 2 Other comments 2 “In the deletions asterless levels are reduced in SC but not in the testis tip, why is that?”

We observed a significant reduction of Asl not only in SC but also in SG, as quantified on Figure 3C.

Referee 2 Other comments 9: “Figure 4 A for clarity it would be helpful to provide landmarks, marker for stem cells (Vasa) or the Hub to be able to understand what we are looking at. Same for Fig 4E, Mira or Dpn? The DAPI staining does not allow in the images provided to identify NBs.”

For the figure 4A, spermatocyte stage was identified based on nuclei staining. We did not include the labelling of nuclei because on a projection of a z-stack of this stage is complicated to interpret.

For the figure 4E, we used cell size and Miranda-GFP to identify NBs. We modified the panel to show the Miranda-GFP instead of nuclei for cell identification.

Referee 2 Other comments 10 “Figure 5: For the fluorescence profile plots it would be nice to see the average plus the standard deviation of the signal in the quantifications.”

Because not all centrioles are exactly orientated the same way, it is very complicated to superimpose the profile plots to be able to show the mean and standard deviation. We only classified the plots in categories to determine their distribution.

Reply to Referee 3

2. Description of the planned revisions

Insert here a point-by-point reply that explains what revisions, additional experimentations and analyses are planned to address the points raised by the referees.

Referee3 point 1. A. “Could the authors provide the western blot and staining results of Alms1a and Alms1b proteins after RNAi and gene knockout? “

alms1 knock-out lines have been designed to remove most of the coding sequences. These lines have been sequenced (sequences will be made available on request) and the details of the deletion break points are given in the method section and the legend of Figure S1:

- *alms1a*^{del1} “is a 4270 bp deletion between the position X: 4.635.793, after aa36 in the first exon, and the position X: 4.631.522 in *alms1a* 3’UTR. This deletion removes almost all *alms1a* coding sequence introducing a stop codon at position aa38 of the remaining sequence.”
- For *alms1b*^{del2}, the miniwhite gene has been introduced in place of the gene sequence “between the positions X: 4.627.368 (14nt after *alms1b* 3’UTR) and X: 4.630.821 (after aa21), it generated a deletion of 3542 bp, removing almost all *alms1b* coding sequence”.
- *alms1*^{del3} has been generated by creating the *alms1b*^{del2} in the *alms1a*^{del1} line.

Thus, in these lines most of the coding sequence of *alms1a*, *alms1b* or both genes have been deleted. In particular, the antigenic sequences used by Chen and Yamashita (2020) to generate the anti-Alms1a (aa 471-499) and the anti-Alms1b (aa 814-826) antibodies have been removed. We could not detect RNA by RT-PCR covering the remaining Alms1a sequence in the *alms1*^{del1} allele, suggesting that no protein could be produced from this deleted locus. To exclude a potential insertion of the deleted genes elsewhere in the genome we also have performed a PCR using primer couples internal to each gene on genomic DNA extracted from *alms1a*^{del1}, *alms1b*^{del2} and *alms1*^{del3} flies (see Referee 1).

The UAS-*alms1*^{RNAi} line used in our work has been previously validated by Chen and Yamashita (2020) by western-blot. In that paper, they expressed the *alms1*^{RNAi} construct using a line ubiquitously expressing Gal4 (Tub-Gal4) and showed by western-blot a complete loss of both Alms1a and Alms1b proteins in these conditions, thus validating the UAS-*alms1*^{RNAi} line.

Referee3 point 1. B. “The most reliable and widely accepted method to determine the specificity of RNAi (whether it is an off-target effect or not) is through genetic rescue experiments. Could the authors provide rescue data? It would be great if the authors could show the rescue of centriolar signals of Plk4, Sas-6, and Ana2, in addition to centriole duplication.”

Referee3 point 1. C. “The authors failed to discriminate which gene is crucial for centriole duplication in GSGs due to the sequence similarity. Linked to the above point, could the authors show the rescue results with either or both Alms1a and/or Alms1b genes? If only one gene can rescue the centriole duplication defects, it would clarify the specific functions of Alms1a and Alms1b in this process.”

Revision Plan

We are currently generating a *Drosophila* line allowing the expression of and RNAi-resistant Alms1a construct (UAS-Alms1a^R). We hope to be able to provide the rescue experiment of the *alms1^{RNAi}* within two months. Based on the localisation profiles, we expect that the function in centriole duplication is carried by Alms1a as Alms1b is not detected at centrioles in spermatogonia. UAS-Alms1a^R should thus answer point B.

We are also generating RNAi lines specific for either *alms1a* or *alms1b*, which should allow us to answer point C.

Referee 3 point 2. “There were no results with endogenous Alms1a and Alms1b in the manuscript. Could the authors provide immunostaining results for endogenous Alms1a and Alms1b? If expansion microscopy is not available due to antibody specificity issues, standard confocal imaging would be sufficient.”

Anti-Alms1a and anti-Alms1b have been generated by Chen and Yamashita (2020) and we indeed have performed the staining with the anti-Alms1a, kindly provided by Y. Yamashita, and have added it in the new Figure S4 showing that endogenous Alms1a localises as Alms1a-Tom. We have asked for the anti-Alms1b antibody as well and will perform the immunofluorescence showing the endogenous Alms1b once we receive it.

3. Description of the revisions that have already been incorporated in the transferred manuscript

Please insert a point-by-point reply describing the revisions that were already carried out and included in the transferred manuscript. If no revisions have been carried out yet, please leave this section empty.

Referee3 point 3. “In figure 3C and D, the authors noted measurements from 7 to 8 testes per group. It would be helpful to also know how many centrosomes were measured, as done in Figure 4C.”

Numbers are indicated in the method section but were also integrated in the corresponding figure legend as follows:

“10 centrioles were quantified per stage and per testis. For Asl: control n=70 centrioles per stage, 7 testes, *alms1a^{del1}*: n=80 centrioles per stage, 8 testes. For Plp: control n=80 centrioles per stage, 8 testes, *alms1a^{del1}*: n=80 centrioles per stage, 8 testes.”

Minor comments Referee3 point 1 “In figure S1, the orientation of the gene is not consistent between endogenous and transgenic Alms1. Could the authors make it consistent for readers to intuitively recognize it? Additionally, the promoter used for transgenic gene expression is not specified. Please provide this information as well.”

The Fig. S1 has been modified to change the orientation of the genes. Regarding the promoters, we used the *alms1a* or *b* upstream regulatory regions for respectively Alms1a-Tom and Alms1b-GFP transgenes. This information appears both in the Fig. S1 legend and in the Method section.

Minor comments Referee3 point 2. “On page 5, the statement ‘~, whereas Alms1b is associated with post-duplication maturation of the centriole.’ needs revision. The authors observed strong Alms1b-GFP signals in the centrioles at the end of the SC stage or during meiotic division but did not observe any defects in centriole maturation at the current stages. Furthermore, Alms1b was not detected from rapidly duplicating centrioles in syncytial embryos and dividing neuroblasts (Page 6), making it hard to understand that Alms1b has a role in post-duplication centriole maturation.”

Indeed, we meant that Alms1b is associated with post-duplication mature centrioles. The text has been modified accordingly.

Minor comments Referee3 point 3. “In figure S2, the centrosome drawings are unclear (initially thought to be 'v' marks). Could a centrosome drawing be added to the legend for clarity?”

We have modified the Fig. S2 to clarify it.

4. Description of analyses that authors prefer not to carry out

Please include a point-by-point response explaining why some of the requested data or additional analyses might not be necessary or cannot be provided within the scope of a revision. This can be due to time or resource limitations or in case of disagreement about the necessity of such additional data given the scope of the study. Please leave empty if not applicable.

Referee3 point 1. D. (Optional) Considering the results in the manuscript, the authors appear to have good techniques in genetic manipulation in flies. If so, generating transgenic flies of Alms1a and Alms1b tagged with a degron (e.g., dTAG or destabilization domain [DD]) to observe centriole duplication defects upon rapid protein degradation might be feasible to support their observations.

We were indeed very keen to use such approach. We thus generated a knock-in of the AID in *alms1a* gene to induce Alms1a protein degradation. We also tested the DeGradFP approach designed by Caussinus et al. (2012, DOI : 10.1038/nsmb.2180) and using a GFP nanobody to target the same construct (which is also tagged with GFP) to the proteasome. Despite numerous attempts with both approaches, we failed to observe any loss of Alms1a protein or phenotype.

Dear Dr. Morel,

Thank you for submitting your manuscript for consideration by the EMBO Journal. I have now read your manuscript, the reviewer comments and your response to them. Based on our editorial assessment and the referees' positive evaluations, I would like to invite you to submit a revised version of the manuscript along the lines indicated in your revision plan.

We generally allow three months as standard revision time. As a matter of policy, competing manuscripts published during this period will not negatively impact on our assessment of the conceptual advance presented by your study. However, please contact me as soon as possible upon publication of any related work to discuss the appropriate course of action. Should you foresee a problem in meeting this three-month deadline, please let us know in advance in order to arrange an extension.

When preparing your letter of response to the referees' comments, please bear in mind that this will form part of the Review Process File and will therefore be available online to the community. For more details on our Transparent Editorial Process, please visit our website: <https://www.embopress.org/page/journal/14602075/authorguide#transparentprocess>. Please also see the attached instructions for further guidelines on preparation of the revised manuscript.

Please feel free to contact me if you have any further questions regarding the revision. Thank you for the opportunity to consider your work for publication. I look forward to receiving your revised manuscript.

With best regards,

Ieva

Please remember: Digital image enhancement is acceptable practice, as long as it accurately represents the original data and conforms to community standards. If a figure has been subjected to significant electronic manipulation, this must be noted in the

figure legend or in the 'Materials and Methods' section. The editors reserve the right to request original versions of figures and the original images that were used to assemble the figure.

We realize that it is difficult to revise to a specific deadline. In the interest of protecting the conceptual advance provided by the work, we recommend a revision within 3 months (8th Sep 2024). Please discuss the revision progress ahead of this time with the editor if you require more time to complete the revisions. Use the link below to submit your revision:

Link Not Available

Rev_Com_number: RC-2024-02443

New_manu_number: EMBOJ-2024-118083-T

Corr_author: Morel

Title: Alström syndrome proteins regulate centriolar cartwheel assembly by enabling Plk4-Ana2 amplification loop in Drosophila

Corresponding author(s): Véronique Morel, Bénédicte Durand

GENERAL STATEMENTS

We thank the referees for their insightful comments. Referees shared a concern regarding the specificity of the genetic tools used here (deletion of *alms1* genes and *alms1*^{RNAi}). The *alms1* deletions were entirely sequenced as described on Fig. S1. We could also not detect RNA by RT-PCR covering the remaining *alms1a* sequence in the *alms1*^{del1} allele (not shown, available upon request), suggesting that no protein could be produced from the deleted locus. To exclude a potential insertion of the deleted genes elsewhere in the genome we have performed PCR using primer couples internal to each gene on genomic DNA extracted from *alms1a*^{del1}, *alms1b*^{del2} and *alms1*^{del3} flies. As shown on Figure S1B, no *alms1a* gene could be detected in the *alms1a* CRISPR deletion (*alms1a*^{del1}) nor in the line carrying the double *alms1a*, *alms1b* CRISPR deletions (*alms1*^{del3}). Similarly, we did not detect *alms1b* in the lines carrying the *alms1b* CRISPR deletion (*alms1b*^{del2} and *alms1*^{del3}).

To validate the specificity of the *alms1*^{RNAi} tool used, we have generated an Alms1a-Tomato transgene resistant to RNAi (Alms1a^R-Tomato). We show in Figure EV3 that centriole duplication phenotype of the *alms1*^{RNAi} is partially rescued upon expression of Alms1a^R-Tomato, thus showing the specificity of the *alms1*^{RNAi} tool.

We now have also included additional data to support our model: in particular we show that *alms1*^{RNAi} can be partially rescued by concurrent expression of both non-degradable Plk4 and Ana2, whereas their single expression is not sufficient to rescue Alms1 KD, supporting the conclusion that Alms1 is required to reinforce the Plk4/Ana2 amplification loop. Second, we also now show that concurrent overexpression of Alms1a and Plk4 dramatically increases centriole amplification in agreement with our proposed model (Figure 7).

REPLY TO REFEREE 1

Referee 1 point 1: I did not find a rescue experiment of the *Alms1* deletions with the *Alms1* transgenes.

Referee 1 point 2: “Second, are the authors sure that either their crispr strategy did not generate any other knock out- hence the essentiality of rescuing these mutations.”

- For *alms1b*^{del2} allele that only removes *alms1b*, we do not observe any centriolar phenotype nor any other specific phenotype, therefore we cannot evaluate at this point any rescue by the *Alms1b*-GFP transgene.
- Regarding *alms1a*^{del1} allele, the only centriolar phenotype is the reduced accumulation of PCM proteins Asterless and Plp at centrioles (Fig. 3C,D, EV2A). To exclude a potential off-target effect of the CRISPR, we measured Asterless accumulation at centrioles comparing i) control, ii) *alms1a*^{del1} and iii) *alms1a*^{del1} rescued by *Alms1a*-Tom conditions. We observe that Asterless accumulation at the centrioles is rescued upon expression of *Alms1a*-Tom transgene (Figure EV2B), hence showing that this CRISPR strategy did not generate other knock-outs.
- Last, we have now included the quantification of the centriole cohesion defects observed in *alms1* double mutant and in rescue conditions (Fig. EV3B,C). We can thus conclude that the cohesion defect observed in *alms1*^{del3} is due to the deletion of *alms1a* and *alms1b* genes and not to an off-target effect of the CRISPR strategy. Furthermore, it shows that the re-expression of *Alms1a*-Tom is sufficient to restore centriole cohesion.

The manuscript has been modified as follows on p7: “While centriole duplication in *alms1*^{del3} flies is largely normal (Fig. 4A left, Fig. EV3B), we observed a premature disjunction of centrioles in 26% of centriole pairs of *alms1*^{del3} testes (Fig. EV3B middle and focus b, C). This phenotype was fully rescued by expression of *Alms1a*-Tom (*alms1*^{del3}, *Alms1a*-Tom), together validating the specificity of the *alms1*^{del3} CRISPR deletion and the functionality of our *Alms1a*-Tom transgene (Fig. EV3B right, C).” The Figure EV3, showing centriole cohesion in *alms1*^{del3} has been modified to include the quantification of the cohesion defects in i) control ii) *alms1*^{del3} and iii) *alms1*^{del3} rescued by *Alms1a*-Tom conditions and representative images.

Referee 1 point 2: “Or alternatively, are they sure about their RNAi conditions? So can they add the RNAi conditions on the background of their deficiency and see that nothing changes?”

We agree with the reviewer that this point is critical and indeed we have tested this hypothesis as described in the initial version of the manuscript. As described on figure 4, we have performed *alms1*^{RNAi} depletion in the *alms1a,b* deleted background (*alms1*^{del3}, *nosGal4>alms1*^{RNAi}) and we demonstrate that it shows the same phenotype as the *alms1*^{del3}, *nos*-Gal4 condition with no centriole duplication defect. We can thus conclude that the phenotype observed with *alms1*^{RNAi} is indeed specific to *alms1* depletion. To further validate the specificity of the *alms1*^{RNAi}, we have generated an RNAi resistant *Alms1a* transgene (*UAS-Alms1a*^R). We show in Figure EV3D that the centriole duplication phenotype in *alms1*^{RNAi} is partially rescued upon expression of *Alms1a*^R-Tomato thus showing the specificity of the *alms1*^{RNAi} tool.

Referee 1 point 2: “calling Gal4- induce RNAi- acute depletion is not correct. This is certainly not acute and so another designation has to be found.”

We agree with the referee that the RNAi strategy does not remove the pre-existing protein contribution. We used “acute” for our RNAi experiments to convey that the depletion was sudden and not permanently encoded genetically as with the deletions. The point with the RNAi experiment is indeed to alleviate potential genetic compensation mechanisms that can be triggered in chronic genetic contexts and hence partially or fully mask the effect of the protein loss.

This adjective has been long and commonly used in the centriole/cilia field to convey these two ideas, as shown by few examples below.

- In their article published in *J Cell Science* in 2011 and entitled “Spindle positioning in human cells relies on proper centriole formation and on the microcephaly proteins CPAP and STIL”, Kitagawa D, (...) Gönczy P. oppose chronic genetic inactivation and acute RNAi depletion: “This apparent discrepancy might be explained by residual function of the mutant proteins or else compensatory mechanisms upon chronic inactivation that are not operating upon **acute depletion mediated by RNAi.**”
- Hall, E.A., (...) P. Mill. write in “**Acute versus chronic loss** of mammalian Azi1/Cep131 results in distinct ciliary phenotypes” (*PLoS Genet.* 2013, doi:10.1371/journal.pgen.1003928): “we present here detailed analysis of both **acute loss (by siRNA)** and chronic absence (by genetic mutation)”
- In 2020, Aydin ÖZ, Taflan SO, Gurkaslar C, Firat-Karalar in “**Acute** inhibition of centriolar satellite function and positioning reveals their functions at the primary cilium.” (*PLoS Biol* 18(6), DOI: [10.1371/journal.pbio.3000679](https://doi.org/10.1371/journal.pbio.3000679)), write: “Accordingly, **acute or constitutive** loss of satellites through depletion or deletion of PCM1”
- In their review “Centriolar satellite biogenesis and function in vertebrate cells” (*J Cell Sci* (2020), doi: [10.1242/jcs.239566](https://doi.org/10.1242/jcs.239566)), S. Prosser and L. Pelletier also qualify siRNA depletion as acute. “Significantly, **acute (siRNA-mediated)** loss of the satellite protein CEP131”.

In order to remove any ambiguity regarding what is meant by acute, we have defined acute the first time it appears in the text and kept the term in the manuscript with respect to previous usages in the field. The text now appears p4: “We observe that sudden and severe, hereafter referred to as acute, depletion of both Alms1a and b by RNAi is associated with complete centriole duplication failure in several *Drosophila* tissues”

Referee 1 point 2: Third if depletion through RNAi indeed leads to a more evident role of Alms proteins, one is expected to see this over time. Can they do a clone-using an FRT site recombined with their Alms mutation, so that the initial cell divisions do not contain Alms, and so it is expected to fail duplication, which may be overcome with time? So a large clone might have progeny with cells with centrosomes (the young ones) and without (the older ones).

Indeed, generating FRT clones could mimic the RNAi knock-down of *alms1*. The experiment as suggested can unfortunately not be performed for the following technical reason. The FRT clone strategy relies on the recombination between homologous chromosome during mitosis but *alms1a* and *alms1b* are on the X chromosome. In absence of homologous wild-type chromosome in males such FRT clones can thus not be performed.

However, a possible genetic strategy would be to induce conditional deletion of the *alms1a* and *alms1b* genes using Lox or FRT sites, but this would require constructing new genetic alleles and finding ways of tracking inactivated cells, which may not be straightforward given that these specific loci have only a very small intron. We do anticipate to perform such experiments in the future, but we think that investigating the mechanisms of compensation over time are beyond the scope of this manuscript.

Referee 1 point 2: Can they show that RNAi depletion in cells that will generate sensory cilia- these are not assembled? Because I am assuming that the mutants are not uncoordinated...

We decided to focus this manuscript on the, already complex, role of Alms1 in centriole duplication. It is however indeed very interesting to question the role of Alms1 in ciliogenesis and this will be the topic of a future study.

Referee 1 point 3: “If they think that Alms deficiency compensates for Alms loss - can they analyse levels of PLK4, SAS6, Ana2 and SAS4 in the mother centrioles?”

We are limited by the number of antibodies available to perform quantitative western blots on the endogenous proteins from the testes. Therefore, we introduced endogenously tagged Plk4 (Plk4-eGFP) or Ana2 (Ana2-eNG) alleles, already used in the manuscript, as well as Sas6-eNG in the *alms1*^{del3} genetic background. We quantified by immunofluorescence Plk4, Ana2 or Sas-6 levels at centrioles in control versus *alms1*^{del3} conditions (Additional Data 1, for reviewer purpose). We did not observe a significant difference of Plk4 recruitment between control and *alms1*^{del3} condition, in contrast to Ana2 and Sas-6 which are more concentrated at centrioles in *alms1*^{del3} than in control condition. These observations suggest that the compensation mechanism could be an upregulation of key downstream components of centriole duplication. However, many additional experiments would be needed to fully understand the mechanisms of compensation. We therefore chose not to include these data in the manuscript.

Additional Data 1: Fluorescence intensities of Plk4-eGFP, Ana2-eNG and Sas-6-eGFP in control versus *alms1*^{del3} conditions

For each condition, testes were dissected and stained with an anti-Asterless antibody to label the centrosomes and direct GFP or neonGreen signal quantification. Images were taken with the same parameters between control and *alms1*^{del3} using an Andor Spinning Disc microscope. 10 centrosomes were measured per testis. Two independent experiments were pooled after normalization with respect to the median value of the control. (A) Plk4-eGFP. control: 4 testes, n=40; *alms1*^{del3}: 4 testes, n=40. (B) Ana2-eNG. control: 6 testes, n=60; *alms1*^{del3}: 5 testes, n=50. (C) Sas-6-eGFP. control: 5 testes, n=50; *alms1*^{del3}: 5 testes, n=50. Mann Whitney test.

Referee 1 minor point 1: Some figures (the majority) lack scale bars.

For each figure, and unless written otherwise, the scale bar put in one image of a panel applies to the whole panel. We made that choice as adding scale bars in all images made the figures more complicated to read. Some panels were however indeed missing a scale bar. This has been corrected.

Referee 1 minor point 2- I do not think that one can consider centrioles that are not in rosettes to be made de novo. They might just have disengaged. The "novo" centriole should be removed. Actually, PLK4 ND generates extra centrosomes, this is sufficient.

The text has been modified to remove "de novo" (p11).

REPLY TO REFEREE 2

Referee 2 point 1: “Provide a thorough characterisation of the *alms1* deletions generated using qPCR to measure RNA levels in the flies.”

As already stated in the manuscript, the precise deletion points for each deletion generated by CRISPR are mentioned in the method section and in the legend of Figure S1.

- *alms1a*^{del1} “is a 4270 bp deletion between the position X: 4.635.793, after aa36 in the first exon, and the position X: 4.631.522 in *alms1a* 3’UTR. This deletion removes almost all *alms1a* coding sequence introducing a stop codon at position aa38 of the remaining sequence.”
- For *alms1b*^{del2}, the miniwhite gene has been introduced in place of the gene sequence “between the positions X: 4.627.368 (14nt after *alms1b* 3’UTR) and X: 4.630.821 (after aa21), it generated a deletion of 3542 bp, removing almost all *alms1b* coding sequence”.
- *alms1*^{del3} has been generated by creating the *alms1b*^{del2} in the *alms1a*^{del1} line.

We also have performed RT-PCR with primers covering the few remaining 5’ sequences in the putative transcript, but did not detect any transcript compared to control flies (not shown, available upon request). More, to demonstrate that no remaining ALMS1 coding sequence is present in the genome of the mutant flies to exclude a potential insertion of the deleted genes elsewhere in the genome, we have performed PCR using primer couples specific to each gene. We did not detect any *alms1a* coding sequence in the genome of *alms1a*^{del1} or *alms1*^{del3} flies nor *alms1b* coding sequence in *alms1b*^{del2} or *alms1*^{del3} flies (gel image added in Fig.S1B).

Referee 2 point 1: “Provide whether the alleles are viable and fertile and clarify all genotypes in the manuscript”

We added in Fig. S1 a table clarifying viability and fertility of these lines. We have included a table with the full genotypes and the reference to the related figure in the Method section.

Referee 2 point 1 (suite): Can the mutants suppress the Plk4-ND overexpression effect?

It is indeed an interesting question. We expressed the ND-Plk4 construct in the *alms1*^{del3} and control conditions. We observed centrosomes with overduplication either with 3 or 4 centrioles in control condition. Overduplication was also observed in *alms1*^{del3} background although to a lesser extent indicating that indeed *alms1* deletion, like Alms KD by RNAi, reduces the overexpression effect ND-Plk4. This is now added in Fig. 6G and mentioned in the text page 11 as follows:

“Interestingly, overexpression of ND-Plk4 in *alms1a,b* deleted flies (*alms1*^{del3}) leads to few centrosomes with three centrioles compared to the numerous rosettes observed in control conditions (*alms1*^{del3}, *nos-Gal4*>ND-Plk4-GFP compared to *nos-Gal4*>ND-Plk4-GFP, Fig. 6G). This indicates that despite the compensation of Alms1 loss operating in the chronic mutants, Alms1 remains required for over-amplification of centrioles in these overexpression assays.”

Referee 2 point 3: “Quantifications, measure fluorescence ratios (e.g. Figure 3 C,D quantify the ratio of Asl/PIP to Bld10, similarly for PLK4-ND in the relevant figure).”

In our hands, Bld10-GFP is not detected in spermatogonia and could thus not be used as an internal reference for quantification. Neither could Dilatory-GFP, another centriolar protein. Ana1 is commonly used as a centriole marker but it is involved in PCM recruitment. Its overexpression as Ana1-GFP may therefore interfere with *alms1a*^{del1} phenotype. No antibodies specific of other centriolar proteins are available. We hence repeated the analysis of Asterless concentration at centrioles in control versus *alms1a*^{del1} on U-ExM images as this allowed us to use acetylated α -tubulin staining of the centriole as a reference. In agreement with the results shown in Figure 3C, we observed a decrease in the fluorescence ratio Asterless/acetylated α -tubulin in *alms1a*^{del1} when compared to the control (Figure EV2A). We thus confirm our analysis by immunofluorescence and further validate that “Alms1a is a novel inner PCM protein required to either efficiently recruit or stabilise the inner PCM proteins Asl and Plp at centriole”.

Referee 2 point 4: “A model for Alms1a in the style of Figure5A would be great.”

This model has been included in the new version of the manuscript in Figure 7D.

Referee 2 point 5: “What happens to Alms1a upon Plk4 inhibition? Experiments like this could strengthen the validity of the hierarchy of events proposed.”

We attempted to deplete *plk4* by expressing the *plk4*^{RNAi} line (Bloomington stock center stock: ;UAS-Plk4[RNAi, P{TRiP.HMC04604}attP40;) using either bam-Gal4 or nanos-Gal4 drivers. We however failed to observe any impact on centrioles.

Nevertheless, to strengthen our proposed hierarchy, we attempted to rescue the duplication failure phenotype associated with *alms1*^{RNAi} by overexpressing ND-Plk4-GFP, Ana2-GFP or both. While overexpressing ND-Plk4-GFP (Fig. 7) does not rescue the duplication failure associated with *alms1*^{RNAi}, we observed 10% of centrosomes with two centrioles upon overexpression of Ana2-GFP and 64% of centrosomes with 2 centrioles when both ND-Plk4-GFP and Ana2-GFP are overexpressed (Figure 7B,C). Hence, Alms1 requirement can be by-passed by increasing simultaneously non-degradable Plk4 and Ana2, further supporting our hypothesis that Alms1 is required for Plk4-Ana2 amplification loop at the duplication site.

More, overexpressing Plk4-GFP results in more than 57% of centrosomes with over-duplicated centrioles while overexpressing Alms1a has no impact on centriole duplication. However, when co-overexpressing Alms1a and Plk4 we observed 76% of centrosomes with over-duplicated centrioles, thus showing that Alms1a and Plk4 act in synergy to promote centriole duplication (Figure 6 B,C).

Referee 2 Other comments 1

“The organisation and presentation should be improved. The figures reorganised to group what belongs together, I would suggest moving human RPE cell analysis to supplementary data and bring the beautiful EM of BamGal4-Alms1 RNAi, Alms1a-Tomatoe into the main manuscript.”

Referee 2 Other comments 7 “Fig S5 what cells were analysed by TEM?”

We re-organised the figures and moved the EM data (Figure S5) in the Figure 4D in the new version of the manuscript.

EM data (old Fig S5) shows centrioles from spermatocytes. The figure legend has been modified as follows:

“TEM images of *bam-Gal4>lacZ^{RNAi}* or *bam-Gal4>alms1^{RNAi}* mature primary spermatocytes”

Referee 2 Other comments 4 “BamGal4 *alms1a* RNAi, SCs loose centrioles, those that keep centrioles, have *Alms1a*, but fail to initiate procentriole formation. To strengthen this view please provide *nos-Gal4 alms1a* RNAi, *Alms1a*-Tomatoe data showing that now *Alms1a*-Tomato is not present except on the mother centrosomes in GCS.”

In Figure EV3D we present images of *nos-Gal4>alms1^{RNAi}* testes in which an RNAi resistant version of *Alms1a*-Tom (*Alms1a^R*-Tom) has been introduced. 41% of *nos-Gal4>alms1^{RNAi}*; *Alms1a^R*-Tom testes still presented centriole duplication failure and 48% a partial rescue of centriole duplication. In these testes, few extra-long centrioles that fail to duplicate could be observed, some of which remained located at the tip of the testis (usually less than 3). These centrioles are not always in cells contacting the hub but fail to duplicate and are decorated by *Alms1a^R*-Tom. We now show in Fig. EV3D (bottom left image) the hub region of a partially rescued testis in which three extra-long, non-duplicated centrioles are observed in cells touching the hub rosette (GSCs). Inset shows that *Alms1a^R*-Tom localizes along the full length of these centrioles.

Referee 2 Other comments 2 “In the deletions *asterless* levels are reduced in SC but not in the testis tip, why is that?”

We observed a significant reduction of *Asl* not only in SC but also in SG, as quantified on Figure 3C.

Referee 2 Other comments 3 “Conclusion: “while *Alms1b* is only detected on mature centrioles as it is absent from rapidly duplicating centrioles of syncytial embryos or from duplicating centrioles of dividing neuroblasts.” Perhaps add, *Alms1b* when expressed.”

We actually removed these sentence as it was redundant with the previous paragraph.

Referee 2 Other comments 5 “The idea to compare *nos-gal4* and *bam-gal4* driving *Alms1a* RNAi is good. Providing a scheme of when these are expressed in the male germ line will help illustrate the experimental strategy.”

The Fig. S2 has been modified to make the expression profile of *nanos-Gal4* and *bam-Gal4* clearer.

Referee 2 Other comments 6 “*Alms1b* (CG12184) is according to published RNAseq data not expressed in neuroblasts, so it makes sense they do not observe (Knoblich lab data), which could be cited here: “in agreement with RNA-seq data showing low expression of *alms1b* in neurons and glial cells (Li et al., 2022).””

The reference Berger et al. 2012 has been added.

Referee 2 Other comments 8 “Fig S6 monochrome images confirm what is what.”

The figure has been modified to add this information.

Referee 2 Other comments 9: “Figure 4 A for clarity it would be helpful to provide landmarks, marker for stem cells (Vasa) or the Hub to be able to understand what we are looking at. Same for Fig 4E, Mira or Dpn? The DAPI staining does not allow in the images provided to identify NBs.”

For figure 4A, spermatocyte stage was identified based on nuclei staining. We did not include the labelling of nuclei because on a projection of a z-stack of this stage is complicated to interpret.

For figure 4E, we used cell size and Miranda-GFP to identify NBs. We modified the panel to show the Miranda-GFP instead of nuclei for cell identification.

Referee 2 Other comments 10 “Figure 5: For the fluorescence profile plots it would be nice to see the average plus the standard deviation of the signal in the quantifications.”

Because not all centrioles are exactly orientated the same way, it is very complicated to superimpose the profile plots to be able to show the mean and standard deviation. We only classified the plots in categories to determine their distribution.

Referee 2 Other comments 11 “Figure 6 D is not convincing, how were the dots visible chosen for the quantification?”

Figure 6D shows spermatogonia in U-ExM with multiple (*bamGal4>lacZ^{RNAi}*) or single (*bamGal4>alms1^{RNAi}*) centrioles, easily identified based on acetylated α -tubulin staining. Quantification in Figure 6E was performed on similar U-ExM images. We changed the *bamGal4>lacZ^{RNAi}* image for one of older spermatogonia with larger centriole amplification on which centrioles were more distinguishable.

Referee 2 Other comments 12 “Figure 1C. Quantification of protein levels is not provided (is this because the expanded ultrastructural approach is not linear precluding quantification?)”

U-ExM can be used for quantification. However, we do not think that quantifications here would provide any more information on the function of the protein, as we only qualitatively describe the presence and dynamics of the protein.

Referee 2 Other comments 13 “Figure 2 B radial dimensions, it would be great to show the sample average and standard deviation.”

We apologize for this difficulty, these were on the graph but too faint to be seen on the figure. We have improved the representation for it to be more visible.

REPLY TO REFEREE 3 **Referee3 point 1. A.** “Could the authors provide the western blot and staining results of Alms1a and Alms1b proteins after RNAi and gene knockout? “

alms1 knock-out lines have been designed to remove most of the coding sequences. These lines have been sequenced (sequences will be made available on request) and the details of the deletion break points are given in the method section and the legend of Figure S1:

- *alms1a*^{del1} “is a 4270 bp deletion between the position X: 4.635.793, after aa36 in the first exon, and the position X: 4.631.522 in *alms1a* 3’UTR. This deletion removes almost all *alms1a* coding sequence introducing a stop codon at position aa38 of the remaining sequence.”
- For *alms1b*^{del2}, the miniwhite gene has been introduced in place of the gene sequence “between the positions X: 4.627.368 (14nt after *alms1b* 3’UTR) and X: 4.630.821 (after aa21), it generated a deletion of 3542 bp, removing almost all *alms1b* coding sequence”.
- *alms1*^{del3} has been generated by creating the *alms1b*^{del2} in the *alms1a*^{del1} line.

Thus, in these lines most of the coding sequence of *alms1a*, *alms1b* or both genes have been deleted. In particular, the antigenic sequences used by Chen and Yamashita (2020) to generate the anti-Alms1a (aa 471-499) and the anti-Alms1b (aa 814-826) antibodies have been removed. We could not detect RNA by RT-PCR covering the remaining Alms1a sequence in the *alms1*^{del1} allele, suggesting that no protein could be produced from this deleted locus. To exclude a potential insertion of the deleted genes elsewhere in the genome we also have performed a PCR using primer couples internal to each gene on genomic DNA extracted from *alms1a*^{del1}, *alms1b*^{del2} and *alms1*^{del3} flies (see Referee 1).

The UAS-*alms1*^{RNAi} line used in our work has been previously validated by Chen and Yamashita (2020) by western-blot. In that paper, they expressed the *alms1*^{RNAi} construct using a line ubiquitously expressing Gal4 (Tub-Gal4) and showed by western-blot a complete loss of both Alms1a and Alms1b proteins in these conditions, thus validating the UAS-*alms1*^{RNAi} line.

Referee3 point 1. B. “The most reliable and widely accepted method to determine the specificity of RNAi (whether it is an off-target effect or not) is through genetic rescue experiments. Could the authors provide rescue data? It would be great if the authors could show the rescue of centriolar signals of Plk4, Sas-6, and Ana2, in addition to centriole duplication.”

See combined answer with point 1.C.

Referee3 point 1. C. “The authors failed to discriminate which gene is crucial for centriole duplication in GSGs due to the sequence similarity. Linked to the above point, could the authors show the rescue results with either or both Alms1a and/or Alms1b genes? If only one gene can rescue the centriole duplication defects, it would clarify the specific functions of Alms1a and Alms1b in this process.”

To validate the specificity of the *alms1*^{RNAi}, we have generated an RNAi resistant Alms1a transgene (UAS-Alms1a^R-Tomato). We show in Figure EV3D that centriole duplication phenotype of the *alms1*^{RNAi} is partially rescued upon expression of Alms1a^R-Tomato, but also Alms1a-Tom (though to a lesser extent), thus showing the specificity of the *alms1*^{RNAi} tool.

This further shows that *Alms1a* is sufficient to rescue the duplication phenotype associated with depletion of both *alms1a* and *alms1b*. Such result is in agreement with the described localization profiles of *Alms1a* and *Alms1b* at centrioles during their duplication as *Alms1b* is not detected at centrioles in spermatogonia.

As we could observe fully rescued pairs of centrioles, we expect that Plk4, Sas-6 or Ana2 signals are also fully restored. We couldn't however address that point due to genetic constraints as all three proteins are detected using tagged transgenic lines.

Referee3 point 1. D. (Optional) Considering the results in the manuscript, the authors appear to have good techniques in genetic manipulation in flies. If so, generating transgenic flies of *Alms1a* and *Alms1b* tagged with a degron (e.g., dTAG or destabilization domain [DD]) to observe centriole duplication defects upon rapid protein degradation might be feasible to support their observations.

We were indeed very keen to use such approach. We thus generated a knock-in of the AID in *alms1a* gene to induce *Alms1a* protein degradation. We also tested the DeGradFP approach designed by Caussinus et al. (2012, DOI : 10.1038/nsm.2180) and using a GFP nanobody to target the same construct (which is also tagged with GFP) to the proteasome. Despite numerous attempts with both approaches, we failed at this point to observe any loss of *Alms1a* protein.

Referee 3 point 2. "There were no results with endogenous *Alms1a* and *Alms1b* in the manuscript. Could the authors provide immunostaining results for endogenous *Alms1a* and *Alms1b*? If expansion microscopy is not available due to antibody specificity issues, standard confocal imaging would be sufficient."

Anti-*Alms1a* and anti-*Alms1b* have been generated by Chen and Yamashita (2020) and we indeed have performed the staining with the anti-*Alms1a*, kindly provided by Y. Yamashita, and have added it in the new Figure EV1B showing that endogenous *Alms1a* localises as *Alms1a*-Tom. As well, we now also show endogenous *Alms1b* localization (Fig EV1B).

Referee3 point 3. "In figure 3C and D, the authors noted measurements from 7 to 8 testes per group. It would be helpful to also know how many centrosomes were measured, as done in Figure 4C."

Numbers are indicated in the method section but were also integrated in the corresponding figure legend as follows:

"10 centrioles were quantified per stage and per testis. For *Asl*: control n=70 centrioles per stage, 7 testes, *alms1a^{del1}*: n=80 centrioles per stage, 8 testes. For *Plp*: control n=80 centrioles per stage, 8 testes, *alms1a^{del1}*: n=80 centrioles per stage, 8 testes."

Minor comments Referee3 point 1 "In figure S1, the orientation of the gene is not consistent between endogenous and transgenic *Alms1*. Could the authors make it consistent for readers to intuitively recognize it? Additionally, the promoter used for transgenic gene expression is not specified. Please provide this information as well."

The Fig. S1 has been modified to change the orientation of the genes. Regarding the promoters, we used the *alms1a* or *b* upstream regulatory regions for respectively *Alms1a*-Tom and *Alms1b*-GFP transgenes. This information appears both in the Fig. S1 legend and in the Method section.

Minor comments Referee3 point 2. “On page 5, the statement ‘~, whereas *Alms1b* is associated with post-duplication maturation of the centriole.” needs revision. The authors observed strong *Alms1b*-GFP signals in the centrioles at the end of the SC stage or during meiotic division but did not observe any defects in centriole maturation at the current stages. Furthermore, *Alms1b* was not detected from rapidly duplicating centrioles in syncytial embryos and dividing neuroblasts (Page 6), making it hard to understand that *Alms1b* has a role in post-duplication centriole maturation.”

Indeed, we meant that *Alms1b* is associated with post-duplication mature centrioles. The text has been modified accordingly.

Minor comments Referee3 point 3. “In figure S2, the centrosome drawings are unclear (initially thought to be ‘v’ marks). Could a centrosome drawing be added to the legend for clarity?”

We have modified the Fig. S2 to clarify it.

Dear Dr. Morel,

Thank you for submitting a revised version of your manuscript. We have now received input from two of the original reviewers, who find that their main concerns have been addressed satisfactorily and recommend acceptance of the manuscript after a minor revision as outlined by reviewer #2 regarding the statistical analysis and textual presentation.

There also are a few editorial points that need addressing before I can extend official acceptance of the manuscript:

1. Please reduce the number keywords to five.
2. Please make sure that the order of the sections in the manuscript is as follows: abstract, introduction, results, discussion, materials & methods, data availability section, acknowledgments, disclosure statement and competing interests, references, main figure legends, tables, expanded figure legends.
3. In the Author Checklist file, please make sure that an answer option is selected for all rows in the column D, and for the sections marked with "Yes", information on the appropriate manuscript section is added in the column E.
4. In the "Data availability" section, please add the standard sentence "The imaging source data for this study are available in the following database:"
5. CRedit has replaced the traditional author contributions section because it offers a systematic, machine-readable author contributions format that allows for more effective research assessment. Please remove the Authors Contributions from the manuscript and use the free text boxes beneath each contributing author's name in our online submission system to add specific details on the author's contribution. More information is available in our guide to authors.
6. Please rename "Declaration of interests" section into "Disclosure and competing interests statement" (further info: <https://www.embopress.org/page/journal/14602075/authorguide#conflictsofinterest>).
7. Please update references according to The EMBO Journal style - where there are more than 10 authors on a paper, the first 10 should be listed, followed by 'et al.' Please see further information here: <https://www.embopress.org/page/journal/14602075/authorguide#referencesformat>
8. Please check the order of the figure callouts; currently, Fig 3 A,B are called out before Fig 1C,D.
9. There are references to "data not shown" on pages 6 and 11. Since our policy does not permit this, please add this data in the Appendix.
10. Please rename the table of primers into "Appendix Table S1" and update the callouts accordingly.
11. Please remove the legends of the appendix figures from the manuscript text.
12. During our routine image quality check, we noted that in Figure 6C, Plk4-OE section, V5 column, the bottom panel is empty of signal. We realise that this is meant to be control image, however, some level of background signal would be expected. Since the file submitted to BioStudies is rather large, I have not been able to download it. Please submit the source data for this panel in our system so that we can perform standard checks.
13. Our data editors have flagged the following issues in figure legends that need correcting:
 - Please provide the exact p values in the legends of figures 3c-d; 6b; 7a-b; EV 2a.
 - Please indicate the statistical test used for data analysis in the legends of figures 3c-d.
 - Please define the box plots in terms of minima, maxima, centre, bounds of box and whiskers, and percentile in the legends of figures 3c-d; 5e; EV 2a.
 - Please note define the error bars in the legends of figures 2a-b.
 - Please note define the asterisk in the legend of figure 5b-e.
 - Please note define the white arrows in the legend of figure 6d, f.
 - Please note that axis gaps are not labeled appropriately in figure 4b.

With best wishes,

Ieva

Revision to The EMBO Journal should be submitted online within 90 days, unless an extension has been requested and approved by the editor; please click on the link below to submit the revision online before 12th Mar 2025:

Link Not Available

Referee #2:

This study presents a thorough characterisation of the ALMS1 homologs in *Drosophila*. They check the localisation and function in various symmetrically and asymmetrically dividing cells in the fly and provide a in depth characterisation in the cells of the male germ line and in other cells of their function. The overall conclusion is that Alms1 is an inner centriole protein that is required for centriole duplication by supporting the stabilisation of PLK4 via Ana2. Without Alms1 Plk4/Ana2 are unable to trigger the assembly of Sas6 cartwheels leading to the failure to duplicate centrioles. This is an interesting and important observation given the disease relevance of Ana2 and adds to our understanding of the molecular mechanism of centriole duplication. This is strong. The quality of the data is further high, and the experimental design carried out at a high standard. Overall, the experiments presented support the conclusions of the manuscript.

However, the authors could elevate the rigor of the study by backing up some of the data with more robust quantification and statistical analysis to measure the differences observed. This is done in many cases, but still lacking or incomplete in others. I don't doubt the data and find them convincing; it is just that the data representation lacks behind in standard achieved in the data themselves and the experimental design.

An example: "(E) Unexpanded images of wor-Gal4>lacZRNAi or wor-Gal4>alms1RNAi larval brain (lacZRNAi: 3 brains; alms1RNAi: 5 brains). NBs (outlined with white dotted lines) are identified based on Miranda-GFP expression (yellow)." This is not quantifying the observed phenotypes. Please check all experiments in the manuscript that they comply with this standard, they are easy to identify.

I understand why the authors want to make a distinction between the mutant and the RNAi condition given the unknown compensatory pathway they suspect to operate in the alms1 CrispR deletion. For this point I would recommend the authors state that the alms1 deletion is an RNA null allele and provide the data. Acute inhibition is usually associated with an inhibitor or a targeted protein degradation approach, to avoid confusion sticking to mutant and RNAi is probably better.

Perhaps change subheading "Alms1 proteins are required for Plk4 activity" to "required for Plk4 function in centriole duplication"? An effect on activity is likely but not formally shown.

Fig 2 legend state that ALMS1 is detected with an antibody.

Statistical tests used need to be specified somewhere.

Discussion, "Here we show that, in *Drosophila*, Alms1 proteins are general regulators of centriole duplication as their acute depletion by RNAi results in complete failure of centriole duplication in all asymmetrically and symmetrically dividing somatic and germline cells analysed". Maybe clarify in the results that the RNAi tool used targets both Alms1a and b and reword as this is not true in all fly cells, some do not express Alms1b.

Referee #3:

The authors have addressed my concerns and I support publication of this study in its current form.

Rev_Com_number: RC-2024-02443

New_manu_number: EMBOJ-2024-118083R

Corr_author: Morel

Title: *Drosophila* Alms1 proteins regulate centriolar cartwheel assembly by enabling Plk4-Ana2 amplification

The authors addressed the remaining editorial issues.

Dear Dr. Morel,

Thank you for submitting the final revised version of your manuscript. I sincerely apologise for the delay in communicating the decision, which was caused by my absence from the office due to an illness in the family. I am now pleased to inform you that your manuscript has been accepted for publication.

Before we forward your manuscript to our publishers, I would like to propose some minor edits in the manuscript abstract and synopsis (please see below and the attached text file). I have also written a short blurb that will accompany the title of your manuscript in our online system. Please let me know if any corrections or adjustments are needed:

Blurb:

Alms1a and Alms1b are general regulators of centriole duplication that exhibit distinct centriolar localization patterns, as revealed by ultrastructure expansion microscopy.

Synopsis:

Alms1a, a Drosophila homolog of the human ciliopathy gene Alstrom syndrome, has been shown to promote centriole duplication in male germline stem cells. Here, Alms1a/b proteins are both identified as general regulators of centriole duplication, promoting Plk4-Ana interaction.

- Alms1 proteins play a critical role in centriole duplication in various Drosophila cell types.
- Ultrastructure expansion microscopy reveals distinct centriolar localization patterns of Alms1 proteins.
- Alms1 proteins are required for and potentiate the Plk4/Ana2(STIL) interaction, which is crucial for Sas-6 cartwheel formation and centriole duplication.
- More subtle phenotypes upon chronic Alms1a/b depletion compared to acute knockdown suggest the activation of possible compensatory mechanisms.

If you have any questions, please do not hesitate to contact the Editorial Office. Thank you for this contribution to The EMBO Journal and congratulations on a nice study!

With best wishes,

leva

leva Gailite, PhD
Senior Scientific Editor
The EMBO Journal
Meyerhofstrasse 1
D-69117 Heidelberg
Tel: +4962218891309
i.gailite@embojournal.org

Rev_Com_number: RC-2024-02443

New_manu_number: EMBOJ-2024-118083R1

Corr_author: Morel

Title: Drosophila Alms1 proteins regulate centriolar cartwheel assembly by enabling Plk4-Ana2 amplification